# Non Methane Hydrocarbons variability in Athens during wintertime: The role of traffic and heating

Anastasia Panopoulou[1,2,4], Eleni Liakakou[2], Valérie Gros[3], Stéphane Sauvage[4], Nadine Locoge[4], Bernard Bonsang[3], Basil E. Psiloglou[2], Evangelos Gerasopoulos[2], Nikolaos Mihalopoulos[1,2]

[1]Chemistry Department, University of Crete, 71003 Heraklion, Crete, Greece
[2]National Observatory of Athens, Institute for Environmental Research and Sustainable Development, 15236 Palea Penteli, Greece.
[3]LSCE, Laboratoire des Sciences du Climat et de l'Environnement, Unité mixte CNRS-CEA-UVSQ, CEA/Orme des Merisiers, 91191 Gif sur Yvette Cedex, France.
[4]IMT Lille Douai, Univ. Lille, SAGE - Département Sciences de l'Atmosphère et Génie de l'Environnement, 59000 Lille, France

*Correspondence to*: Eleni Liakakou (liakakou@noa.gr)

**Abstract.** Non-methane hydrocarbons (NMHCs) play an important role in atmospheric chemistry, contributing to ozone and secondary organic aerosol formation. They can also serve as tracers for various sources such as traffic, solvents, heating and vegetation. The current work presents, for the first time to our knowledge, time resolved data of NMHCs, from two to six carbon atoms, during a period of five months (mid-October 2015 to mid-February 2016) in the Great Athens Area (GAA), Greece. The measured NMHC levels are among the highest reported in literature for the Mediterranean area during winter months and the majority of the compounds demonstrate a remarkable day to day variability. Their levels increase by up to factor of four from autumn (October-November) to winter (December-February). Microscale meteorological conditions and especially wind speed in combination with the PBL height, seem to contribute significantly to the variability of NMHC levels, with an increase up to a factor of 10 under low wind speed (<3 m s$^{-1}$), reflecting the impact of local sources rather than long range transport. All NMHCs demonstrated a pronounced bimodal, diurnal pattern with a morning peak followed by a second one before midnight. The amplitude of both peaks is gradually increasing towards winter, respectively to autumn, by a factor of 3 to 6 and nicely follow that of carbon monoxide (CO), indicating contribution from sources other than traffic, related to combustion e.g. domestic heating (fuel or wood burning). By comparing the NMHC diurnal variability with that of black carbon (BC), its fractions associated with wood burning (BC$_{wb}$) and fossil fuel combustion (BC$_{ff}$), as well as with source profiles we conclude that the morning peak is attributed to traffic while the night one mainly to heating. For the night peak, the selected tracers and source profiles clearly indicate contribution from both traffic and domestic heating (fossil fuel and wood burning). NMHCs slopes versus BC$_{wb}$ are almost similar when compared to those versus BC$_{ff}$ (slight difference for ethylene), indicating that NMHCs are probably equally produced by wood and oil fossil fuel burning.

# 1 Introduction

Non-methane hydrocarbons (NMHCs) are key atmospheric constituents for atmospheric chemistry. In the presence of NOx, their oxidation leads to formation of tropospheric ozone and other species, such as peroxy radicals ($RO_2$) and peroxy acetyl nitrate (PAN), thus affecting the oxidative capacity of the atmosphere (Atkinson, 2000 and references therein). NMHC oxidation contributes to the formation of secondary organic aerosols (SOA), which in turn affects light scattering, visibility and CCN formation (Tsigaridis and Kanakidou, 2003; Seinfeld and Pandis, 2016 and references therein). In urban areas they mainly originate from anthropogenic sources such as traffic, solvent use, residential heating, natural gas use, industrial activity, but also from natural sources such as vegetation (Guenther et al., 1995; Barletta et al., 2005; Kansal, 2009; Sauvage et al., 2009; Salameh et al., 2015; Baudic et al., 2016; Jaimes-Palomera et al., 2016). Besides their key role as secondary pollutants precursors, NMHCs are of interest because of their association with health issues (EEA report, No 28/2016, 2016). In particular and since 2013, atmospheric substances have been classified by the International Agency for Research on Cancer (WHO-IARC, 2013) in four major groups regarding their carcinogenicity to humans, with benzene and 1,3-butadiene among those NMHCs classified as potential carcinogens (IARC, 2012).

Athens, the capital of Greece with almost five million of inhabitants, is frequently subjected to intense air-pollution episodes, leading to exceedance of the EU air quality limits. The driving processes and atmospheric dynamics of these episodes have been scrutinized during the last decades (Cvitas et al., 1985; Lalas et al., 1982, 1983, 1987; Mantis et al., 1992; Nester, 1995; Melas et al., 1998; Ziomas et al., 1995; Kanakidou et al., 2011). However, the measurements of pollution precursors are mostly limited to ozone and nitrogen oxides. The few existing and non-continuous NMHC measurements in Athens by means of canisters or sorbent tubes have been performed for short period (days) during summertime or autumn (Moschonas and Glavas, 1996; Klemm et al., 1998; Moschonas et al., 2001; Giakoumi et al., 2009). Continuous measurements of NMHCs in Athens for a period of one month have been conducted during summer 20 years ago at three locations, two suburban and one urban, reporting almost 50 C4 – C12 compounds (Rappenglück et al., 1998, 1999), and recently by Kaltsonoudis et al. (2016), for 1 month in winter 2013 at an urban location (Thissio) and one in summer 2012 at a suburban one (A. Paraskevi), reporting 11 oxygenated organic gaseous compounds and C5 – C8 NMHC. Meanwhile, significant changes in pollutant sources occurred in Athens the last 20 years, which lead to significant decreases in the annual concentrations of major pollutants such as CO, $SO_2$, NOx (Gratsea et al., 2017; Kalabokas et al., 1999). As this trend was attributed to the car fleet renewal, fuel improvement, metro line extension and industrial emission controls, a decrease in NMHC levels originating from traffic and industrial emissions is also expected. However, after 2012, a new winter-time source of pollution emerged in Greece, due to uncontrolled wood burning for domestic heating (Saffari et al., 2013; Paraskevopoulou et al., 2015; Kaltsonoudis et al., 2016; Fourtziou et al., 2017; Gratsea et al., 2017). This is an important source of various pollutants such as particulate matter (PM), polycyclic aromatic hydrocarbons (PAHs), black carbon (BC) and CO (Gratsea et al., 2017; Hellén et al., 2008; Paraskevopoulou et al., 2015; Schauer et al., 2001 et references therein), while it can represent up to 50% of the mass of Volatile Organic Compounds (VOCs) during winter (case of Paris; Baudic et

al., 2016). Studies for the characterization of the VOC emissions from domestic wood burning based on emissions close to sources, in ambient air or in chambers are published, however, differences are observed in the emission rates or the emission profiles, that are attributed to the type of wood, stove, lightening material and the burning stages (Barrefors and Petersson, 1995; Baudic et al., 2016; Evtyugina et al., 2014; Gaeggeler et al., 2008; Gustafson et al., 2007; Hellén et al., 2008; Liu et al., 2008; Schauer et al., 2001 and references therein). Moreover, the studies reporting light NMHC measurements from domestic wood burning are very few (Barrefors and Petersson, 1995; Baudic et al., 2016; Liu et al., 2008; Schauer et al., 2001) and present significant discrepancies. For example, the higher contribution of benzene relatively to acetylene in the residential wood burning profile reported by Baudic et al. (2016) was different from the profile of Liu et al. (2008). In addition, in their recent work, Kaltsonoudis et al. (2016) reported important contribution of wood burning to the wintertime night concentrations of aromatics and oxygenated VOCs. The above clearly demonstrates the increasing need for intensive measurements of NMHCs in Athens, which in turn will allow assessing the impact of future changes (fuel composition changes or other control strategies) on atmospheric composition. In other words, there is a need to establish a "current baseline" for Athens atmospheric composition in terms of NMHC levels.

The current study presents, time-resolved data of 11 selected from 15 determined NMHCs with two to six carbon atoms, during a time span of several months (October 2015 to mid-February 2016) in the Great Athens Area (GAA). In addition, time-resolved data of toluene, ethylbenzene, m- /p- xylenes and o-xylene are used, which were monitored simultaneously from mid-January to mid-February 2016. The emphasis of this work is on: (1) the determination of the ambient levels of C2 – C6 NMHCs during autumn and winter, twenty years after their first summer-time measurements. Especially for C2 – C3, these are the first ever continuous measurements of NMHCs in Athens; (2) the study of the NMHC temporal characteristics and the determination of the factors controlling their variability, and (3) the investigation of traffic and residential heating impact on NMHC levels which are among the most important sources of air pollution in Athens, especially during the "crisis" period characterized by an important decline of industrial activity (Vrekoussis et al., 2013).

## 2 Experimental

### 2.1 Sampling site

Measurements were conducted from 16 October 2015 to 15 February 2016, at the urban background station of the National Observatory of Athens (NOA, 37.97° N, 23.72° E, 105 m a.s.l and about 50 m above the mean city level) at Thissio, considered as receptor of pollution plumes of different origins (Paraskevopoulou et al, 2015). The station is located in the historical center of Athens, on top of a hill (Lofos Nimfon), surrounded by a pedestrian zone, a residential area and by the Filopappou (108 m a.s.l) and Acropolis Hills (150 m a.s.l), which are located 500 m and 800 m away respectively (Fig. 1). More information about Athens' morphology, meteorology and dominant transport patterns can be found in Kanakidou et al. (2011), Melas et al. (1998) and references therein.

## 2.2 On line NMHC measurements

Two portable gas chromatographs equipped with a flame ionization detector (GC – FID), Chromatotec, Saint Antoine, France) were used for the measurement of NMHCs in Athens. Specifically, the "airmoVOC C2 – C6" (during the whole period, from October 2015 to February 2016) and the "airmoVOC C6 – C12" Chromatrap GC (from mid-January until mid-February 2016) analyzers were used for the determination of C2 – C6 and C6 – C12 NMHCs respectively, collecting ambient air through collocated inlets at the rooftop of the station, 4 m above ground. The C2 – C6 NMHC analyzer was set to sample ambient air on a 10 min basis followed by an analysis time of 20 min, while for the C6 – C12 the respective timing was 20 min and 20 min, with a total cycle of 30min (sampling and analysis). Therefore, the synchronized monitoring was performed with an overall 30 min time resolution, for both analyzers.

For the airmoVOC C2 – C6 analyzer, 189 mL of air was drawn through a 0.315 cm diameter, 6 m-long stainless-steel line with a filter of 4 µm pore size at the sampling inlet, and a flow rate of 18.9 mL min$^{-1}$. Once sampled, ambient air was passed through a Nafion dryer (activated by gas nitrogen) to reduce the water content and then hydrocarbons were preconcentrated at -9 $^0$C (Peltier cooling system), on a 2.25 mm internal diameter, 8 cm-long glass trap containing the following adsorbents: Carboxen 1000 (50 mg), Carbopack B (10 mg) and Carbotrap C (10 mg) all from Supelco Analytical, Bellefonte, PA, USA. The trap was then heated rapidly to 220 $^0$C for 4 min and the pre-concentrated VOCs were thermally desorbed onto a Plot Column (Restek Corp., Bellefonte, PA, USA, $Al_2O_3/Na_2SO_4$), 25 m x 0.53 mm, 10 mm film thickness). 1 min prior to the analysis, the oven temperature was raised from 36 to 38 $^0$C, followed by a constant heating rate of 15 $^0$C min$^{-1}$ to 200 $^0$C by the end of the analysis. Details about the equipment technique and performances, as well as the estimation of the uncertainty, are provided by Gros et al. (2011). The detection limit is in the range of 0.02 ppb (propene, n-pentane) to 0.05 ppb (propane), while for ethane and ethylene is 0.1 ppb.

The airmoVOC C6 – C12 analyzer was collecting 900 mL of air through a 0.315 cm diameter, 6 m-long stainless-steel line with a filter of 4 µm pore size at the sampling inlet, and a flow rate of 45 mL min$^{-1}$. The hydrocarbons were preconcentrated at ambient temperature on a glass trap containing the adsorbent Carbotrap C. Then the trap was heated to 380 $^0$C over 2 min to desorb the pre-concentrated VOCs into a separation column (MXT30CE, Restek Corp., 30 m x 0.28 mm, 1 mm film thickness). With one minute delay, the oven temperature was raised from 36 to 50 $^0$C at a rate of 2 $^0$C min$^{-1}$, followed by a second heating of 10 $^0$C min$^{-1}$ up to 80 $^0$C. Finally, at a constant heating rate of 15 $^0$C min$^{-1}$ the temperature reached 200 $^0$C and remained there until the end of the analysis. In the present work, toluene, ethylbenzene, m-/p- xylenes and o-xylene (TEX) will be used from the GC C6 – C12 data series. The uncertainty of the instrument is less than 20% and the detection limit of the BTEX is 0.03 ppb.

Simultaneous calibration and identification of the compounds were performed by a certified National Physical Laboratory (NPL) standard of NMHC mixture (~4 ppb) containing: ethane, ethylene, propane, propene, i-butane, n-butane, acetylene, i-pentane, n-pentane, isoprene, benzene and 15 additional hydrocarbons.

**2.3 Auxiliary measurements**

Real time monitoring of carbon monoxide (CO), black carbon (BC) and nitrogen oxides ($NOx = NO$ and $NO_2$) was also conducted during the reported period. For CO and NOx measurements, Horiba 360 Series Gas Analyzers of one-minute resolution were used which were calibrated with certified standards. A seven wavelength Magee Scientific AE33 aethalometer (one minute resolution) was operated for the measurement of BC and its fractions associated with fossil fuel and wood burning ($BC_{ff}$ and $BC_{wb}$, respectively) derived automatically by the instrument software. Meteorological data were provided by NOA's meteorological station at Thissio premises.

**2.4 Street canyon measurements**

To identify the NMHCs fingerprint of traffic emissions, NMHCs measurements were conducted at a monitoring station belonging to the air quality agency of Athens and located at a street canyon downtown Athens with increased traffic and frequent traffic jams (Patission street) on 22 to 24 February 2017 (37.99°N, 23.73°E). Samples were collected every hour during the morning rush hour from 06:55 LT to 10:15 LT (LT = UTC+2), in 6L stainless steel – silonite canisters. The sampling method for ambient air is described in detail elsewhere (Sauvage et al., 2009). Before the analysis, the cylinders were pressurized by adding a known amount of zero-air resulting in a sample dilution by a factor of two. Afterwards each canister was connected to the GC-FID system using a Teflon (PTFE) sampling line and analyzed by the method described in Sect. 2.2. Before sampling, the canisters were cleaned by filling them up with zero air and re-evacuated, at least three times. The content of the cylinders was then analyzed by the GC-FID system to verify the efficiency of the cleaning procedure. The canisters were evacuated a few days prior to the analysis and they were analyzed maximum 1 day after the sampling.

**3 Results and discussion**

**3.1 Temporal variability of NMHCs**

Figure 2 presents the temporal variability of selected NMHCs for five major groups of compounds: ethane and n-butane (for saturated hydrocarbons), propene and ethylene (for alkenes), acetylene (for alkynes), benzene and toluene (for aromatics) and isoprene (for potential biogenic compounds). Other measured NMHCs are presented in Fig. S1. During the reported period, the data availability (in comparison with the maximum potential data availability) for all C2 – C6 NMHCs was higher than 87%. Most of the data for isoprene are below the limit of detection due to the low vegetation activity at this period of the year (Fuentes et al., 2000; Guenther et al., 1995). Moreover, the significant night time levels (above 300 ppt in some cases) could be indicative of non-vegetation sources, like traffic or domestic wood burning (Borbon et al., 2001, 2003; Gaeggeler et al., 2008; Kaltsonoudis et al., 2016). However due to the low data coverage, it is not possible to determine an accurate diurnal variability for this compound.

The majority of the compounds showed a remarkable day to day variability throughout the study period and levels increasing by up to factor of four, from autumn (October-November) towards winter (December-February; Fig. 2 and S1). The highest values which have been observed for ethane and ethylene ranged mostly between 26 and 23 ppb, and were encountered in wintertime. For these compounds, the lower values were above 0.3 ppb for the whole period. During the period of intensive measurements, toluene exceeded 10 ppb, while benzene was below 6 ppb during the four-month monitoring period. Benzene is the only NMHC included in the European air quality standards due to its possible adverse health effects (IARC, 2012).

In Table 1, the mean values of the measurements of this study are compared with those reported in the literature for Athens in the past and other selected areas. The comparison with those already published data for the GAA, indicates an apparent decrease by a factor of 2 to 6 for the majority of the species lying above C4 (taking as reference the case of Ancient agora urban area in the close vicinity of Thissio Station). This decreasing trend is in agreement with a decrease in primary pollutants CO, $SO_2$ already reported by Kalabokas et al. (1999) and Gratsea et al. (2017), due to the air quality measures taken by the Greek government and economic recession (since 2012). Apart changes in emission sources and source strength during the last twenty years, differences in sampling period (summer versus winter), analytical resolution (morning collected samples compared to continuously averaged levels) should be considered, rendering the direct comparison between the present and past measurements quite difficult in the overall evaluation of the NMHCs decrease. However, in order to better investigate the observed decreasing trend and compare these results to past measurements, enhancement ratios (ppb/ppb) are calculated for i – pentane, benzene, toluene, ethylbenzene and o – xylene to NOx (sum of NO and $NO_2$), following the approach of Kourtidis et al., (1999), using the measurements performed in the street canyon (Patission) and presented at Table 2. In a summary, the enhancement ratios are the slopes of the x-y plots of the selected NMHC (in ppb) to NOx (or CO, both in ppb), and for which morning concentrations (07:00 to 10:00 LT) at wind speed lower than 2 m s$^{-1}$ and of SSW to SWW direction (206° to 237°) are used. The NOx and CO data for the Patission site are provided by the Hellenic Ministry of the Environment & Energy, Dept. of Air Quality. Additionally, the same enhancement ratios were calculated for Thissio station, for concentrations associated with wind speed lower than 2 m s$^{-1}$ (no distinction to wind direction), thus maximizing the local influence. Since the enhancement ratios are calculated during the time-window of the traffic rush hours, it is assumed that they are representative of traffic emissions only. Both Thissio station and Patission street canyon demonstrate similar enhancement ratios with differences in the order of 15-30% and 20-35% relatively to NOx and CO respectively and great differences comparatively to the previously reported values. Enhancement ratios for i-pentane, toluene, ethylbenzene and o-xylene to NOx for the same station (Patission) show values which are lower by a factor of 6 ± 1 compared to the ones reported in Kourtidis et al. (1999), whereas a factor of 12 is observed for benzene. The same stands for the present enhancement ratios of the selected NMHCs to CO with a decrease of 2 to 5 times. The lower enhancement ratios reveal the strong impact of the air quality measures to VOC emissions, while the high difference to benzene enhancement ratio is a direct outcome of the Directive n°2000/69/CE (now Directive n°2008/50/EC) of the European Union for the reduction of this compound, especially in fuels.

Beirut, located in the Eastern Mediterranean basin (approximately 200 Km SE of Greece, 230 m above sea level), has a population of 2000000 inhabitants and a typical Mediterranean climate with mild winter and hot summer (Salameh et al., 2015). On the contrary Bilbao is an urban and industrial city with 400000 inhabitants in northern Spain, located along a river delta in SE–NW direction, with two mountain ranges in parallel to the river (Ibarra-Berastegi et al., 2008). Due to their location, both cities experience intense sea breeze cycles. The NMHCs levels observed in Athens are higher by a factor of approximately two for ethylene, propene, acetylene and pentanes compared to these two cities and up to 3.5 for isopentane at Bilbao. Exceptions are propane, butanes and toluene for Beirut and n-butane, benzene and toluene for Bilbao, which are quite comparable to Athens. NMHC levels are also compared with those obtained in Paris, one of the European megacities, with more than 10 million inhabitants with relatively mild winters and warm summers. Again, the observed levels in Athens are significantly higher (almost 2 to 8 times) compared to those reported for Paris (Baudic et al., 2016), with the most important differences concern acetylene and i-pentane (factor of 8.4 and 6.7 higher in Athens respectively, Table 1).

According to Fig. 2, a common pattern for all NMHC concentrations was their gradual increase from October to December, which reflects the transition from the warmer period to the colder one. This is better illustrated in Fig. 3, which depicts the monthly mean concentration for every NMHC presented in Fig. 2. The increase in NMHC levels during the cold period could be explained by the respective increase in their lifetime due to less photochemistry and the contribution from additional sources, such as heating. However, the role of atmospheric dynamics should not be neglected, since the decrease in the height of the planetary boundary layer (PBL) could also trigger the observed winter-time enhancement of the NMHC levels. Nevertheless, according to Alexiou et al. (2018) the mean winter-time decrease of PBL compared to autumn is in the range of 20% for both day and night periods, thus changes in PBL couldn't be the only factor determining the enhancement of NMHCs level observed during wintertime. Furthermore, according to Kassomenos et al. (1995) the day-night difference on PBL is more pronounced during summer. Thus, the night-time accumulation of the pollutants during winter relatively to summer highlights essentially the impact of additional emission sources. Meteorological conditions such as wind speed and direction have to be also considered and their respective role will be discussed thereafter.

## 3.2 Diurnal variability of NMHCs

During the whole monitoring period, all hydrocarbons demonstrated a pronounced bimodal diurnal pattern (Fig. 4 and S2). A morning peak was observed lasting from 07:00LT to 10:00LT, followed by a second one before midnight. The amplitude of both peaks is gradually increasing from October to winter time by a factor of 3 to 6 and nicely follows that of carbon monoxide (CO), BC and its fractions associated with wood burning ($BC_{wb}$) and fossil fuel combustion ($BC_{ff}$) (Fig. 4). As it was noted in Gratsea et al. (2017), the morning maximum of CO is attributed to morning traffic, while the winter night-time increase to additional sources such as domestic heating (fossil fuel or wood burning). Although the amplitude of both CO peaks (morning and night) is almost similar (with the exception of December), the duration of the night peak is at least a factor of 2 larger, which could imply the impact of heating on air quality during wintertime. Moreover, night-time emissions

occur in a shallower boundary layer relatively to the mid-day, resulting into accumulation of pollutants (Alexiou et al., 2018). These observations are indicative of the contribution of traffic and heating to the NMHCs levels. By comparing the NMHC diurnal variability with that of BC, as well as its fractions associated with wood burning ($BC_{wb}$) and fossil fuel combustion ($BC_{ff}$), it is deduced that the morning peak could be mainly attributed to traffic and the late evening to traffic and heating, the latter from the combined use of heavy oil and wood burning.

### 3.3 The role of meteorology on NMHC levels

Once emitted in the atmosphere, NMHCs react mainly with OH and $NO_3$ radicals during day and night-time, respectively, and with ozone throughout the day (Crutzen 1995, Atkinson 2000), whereas the role of Cl could not be omitted, especially for coastal areas (Arsene et al., 2007). Still, in addition to chemistry, many other factors, such as the strength of the emission sources and the atmospheric dynamics (meteorology and boundary layer evolution), determine their abundance and diurnal variability. To investigate the role of wind speed and wind direction, the dependence of n-butane, acetylene and benzene, selected as representative of alkanes, alkynes and aromatics, against wind speed and direction, is depicted in Fig. 5 and 6 respectively (Fig. S3 and S4 include the rest of the compounds). For all studied NMHCs, the highest concentration occurred under low wind speed ($< 3$ m s$^{-1}$) reflecting the critical role of local sources versus long range transport. On a monthly basis, the NMHC dependence on wind speed remains the same for the total examined period (Fig. S5).

To investigate the impact of wind direction on NMHC levels, fig. 6 presents the distribution of wind sectors frequency of occurrence during the sampling period and that of wind speed per sector. In addition, the variability of n-butane, acetylene and benzene levels as a function of wind direction is also depicted. Enhanced levels of NMHCs are found under the influence of air masses from all directions, especially under low wind speed. During the sampling period, the NE sector associated with relatively strong winds ($u > 3$ m s$^{-1}$), was the most frequent one, resulting in moderate levels of NMHCs. Overall, a similar distribution was found for all NMHCs, indicating moderate to higher values under the N-NE-E-SE directions, and lower levels under the NW-W-SW sector, the latter associated with high wind speeds. The influence of the N to SE sector to the enhanced NMHCs levels is probably related to the northern suburbs of GAA, that are characterized by increased number of fireplaces and higher living standards allowing the combined use of heating oil in central heating systems and wood in fireplaces and/or woodstoves. The impact of the N to ESE sector on NMHC levels can be also seen when comparing the concentrations of the morning (07:00 – 09:00) and night (21:00 – 23:00) peaks in October and December (Fig. S6). The wind probability from N to ESE is similar for both months, however significantly higher concentrations are observed during nighttime in December affected by low wind speed ($< 2$ m s$^{-1}$) from the N to NE sector.

The ambient temperature is another parameter which can influence NMHC levels, as high temperatures favor the evaporation of low volatility hydrocarbons and also trigger the production of biogenic compounds, whereas lower temperatures could trigger the emission of NMHCs from increased heating demand, as other tracers as well (Athanasopoulou et al., 2017). The average monthly temperatures varied from 18 $^{0}$C in October and November to 10 - 13 $^{0}$C in December and late winter,

respectively. By examining NMHCs against temperature (Fig. S7), a clear tendency is not evident, although the highest levels are observed at lower temperatures.

### 3.4 Identification of NMHC emission sources with emphasis on traffic and heating

#### 3.4.1 Interspecies correlation

Table 3 shows the interspecies correlation of NMHCs for the total period of measurements. All NMHCs were well correlated ($R^2 > 0.81$), with the exception of isoprene which as seen before, had only few data above the LoD and thus was excluded from Table 3. Note also the excellent correlation of toluene with ethylbenzene, m-/p- xylenes and o-xylene ($R^2$ from 0.92 to 0.93), during the common measured period (from mid-January until mid-February 2016) highlighting their common origin. The strong correlation of NMHCs with combustion tracers, such as CO, NO and BC, could indicate their common emission sources and variability. The deconvolution of BC into its fossil fuel and biomass burning fractions enables further classification of NMHCs into groups that could possibly be emitted by those two distinct sources. The stronger correlation ($R^2 > 0.84$) of the hydrocarbons with $BC_{ff}$ compared to $BC_{wb}$ ($R^2 > 0.64$) could imply stronger emission of NMHCs from fossil fuel combustion processes relatively to wood burning. Finally, no change in the correlation coefficients is observed when data sets are separated between day (6:00-18:00) and nighttime (18:00-6:00) time intervals. However, the above analysis could give only a rough idea on the sources impacting NMHCs levels. A more precise picture could emerge with comparison with source profiles and such discussion follows in the paragraph below.

#### 3.4.2 Impact of various sources on the NMHC levels

To identify periods with differentiated impact from the different pollution sources (with emphasis on traffic and heating), the methodology described by Fourtziou et al. (2017) was applied. The criteria for this separation have been the wind speed not to exceed the threshold value of 3 m s$^{-1}$ (light breeze conditions) and the presence of precipitation (on/off criterion). The role of wind speed was clearly seen at the Sect. 3.3 (Fig. 5). Based on these criteria, the first group (non-shaded in Fig. 7) corresponds to higher wind speeds and thus more efficient dispersion of emitted pollutants (ventilation), as well as the incidents of rain and is characterized as non-smog periods (nSP). The second group (shaded area in Fig. 7) refers to lower wind speeds, favoring accumulation of high pollution loads within the mixing layer and is henceforth referred to as smog periods (SP). The frequency of SP and nSP periods was 65% and 35% respectively. Note that the word "smog" is used as a synonym to highlight cases of relatively high air pollution, as also indicated by the high levels of CO and BC encountered during the SP periods (Fig. 7).

The diurnal variability of all compounds was investigated separately for two distinct months, October and December, representative periods of non-heating and heating activities, respectively (Fig. 8 and S8). Note that SP periods represent 55% of the considered time in October and 73% in December. According to previous findings (Paraskevopoulou et al. 2015; Kaltsonoudis et al. 2016; Fourtziou et al. 2017; Gratsea et al. 2017) wood burning for domestic heating has gained a marking

role as a winter-time emission source in Greece, over the last years. Since wood burning is reported as emission source of specific organic compounds such as ethane, ethylene, acetylene, benzene, methanol, acetaldehyde and acetonitrile (Baudic et al., 2016; Gaeggeler et al., 2008; Gustafson et al., 2007; Hellén et al., 2008; Kaltsonoudis et al., 2016), it can be safely considered as a possible factor contributing to the winter time increase of NMHC levels in GAA. Thus, the two selected months are expected to have different source profiles. October, without or very limited heating demand, was used as a reference period, while December in south-central Greece is traditionally the beginning of the heating period. The low values of $BC_{wb}$ recorded in October, even during the SP periods, support the methodology followed for the separation (Fig. 8).

The levels of all measured NMHCs were significantly higher in December compared to October for the SP periods (Fig. 8 and S8). The most striking difference is related to the night peak, while during mid-day the difference is minimal. For all compounds examined in this work, the night peak in December (SP period) is 2 to 6 times higher compared to October's (SP period) with the highest differences found for ethane, ethylene, propene and acetylene. On the other hand, the December to October ratio during mid-day is ranged between 2.6 (for propene and acetylene) to 0.9 (for benzene). It is worth noting the levels of NMHCs during the traffic related morning peak. Although higher mean levels were observed in December, the amplitude of the morning peak is almost similar in both examined months, denoting no important change in the traffic source between the heating and non-heating periods. In contrast, during the nSP periods in October and December NMHCs levels were equal (Fig. 8 and S8). Furthermore, the concentrations of all compounds during nSP were very low; even lower than the minimum values observed during mid-day during SP periods of the same months. Accordingly, the diurnal variability of all investigated NMHCs was less pronounced compared to the SP periods with a slight increase during night in December, which could be attributed to a background contribution from heating sources. In Sect. 3.4.3 the origin of the morning and nights peaks related to NMHCs will be further investigated.

### 3.4.3 Impact of sources on morning and night peaks of NMHCs

*Morning peak:*

As discussed in Sect. 3.2, the morning peak (07:00 – 10:00 LT) of NMHCs could be mainly attributed to traffic. Fig. 9 presents the profile of this peak (% mass contribution of the measured NMHCs), during January and February SP days when toluene, ethylbenzene, m-/p- xylenes and o-xylene data were also available. Additionally, in the same figure the morning profile obtained during the 2 – days campaign conducted in the street canyon located at the center of Athens (Patission Monitoring Station) is also reported. Details on the calculations for the morning profile for the two sites are provided in Sect. S2. Patission profile reflects all types of traffic-related emissions due to the combination of the high number of vehicles and buses driving on this street, frequent traffic jam conditions, variety of types of fuels (gasoil, diesel, natural gas), vehicles age, maintenance etc.

The two morning profiles, although performed at sites with different impact of traffic, agree quite well ($R^2 > 0.98$). Iso - pentane, toluene and m-/p- xylenes are the three main compounds contributing to the morning profiles accounting by about 50% of the total measured NMHCs at both locations, followed by n- and i-butane and ethylene accounting for almost 21%.

Differences between the two morning profiles regarding these 5 main species are weak (less than a factor of 1.2). Note also that the morning profile at Thissio is the mean of a whole month period compared to the two days campaign at Patission which could explain the small differences between the two profiles. In addition, a comparison with a tunnel study in Athens is made in the supplement (Sect. 2a.), in which similarities are seen for most of the main compounds (i-pentane, m- / p-xylenes, ethylbenzene, o-xylene, benzene, n-pentane, i-butane, propene and ethane), but with the exception of acetylene and toluene that they are lower by a factor of 4 and 1.5 respectively. The similarity of Thissio and Patission morning profiles and their difference from the Athens tunnel profile probably indicates the importance of the type of fuel used. The latter is also observed in recent works (Ait-Helal et al., 2015; Zhang Q. et al., 2018; Zhang Y. et al., 2018), where important differences have been reported between tunnel measurements, and attributed to various typologies of the car-fleets (type of vehicles and fuels). In our case there is a possibility that the car-fleet in the tunnel is not representative for the GAA, since the existing tolls reduce the use of the tunnel due to financial issues. Also, measurements are performed during noon when the traffic density is relatively low compared to the morning peak. In any case, the prevalence of i-pentane and toluene in all profiles, indicates the continuing dominance of gasoline powered cars and evaporative losses. The importance of evaporative losses can be seen in Fig. S11 and S12 where the ratios of butanes and pentanes-to-(C2 – C5)Alkanes (%) versus the temperature are respectively examined. Taking into account the positive dependence of the two ratios, especially that of pentanes, to temperature, we can assume that fuel evaporation losses are also an important source of NMHCs. These observations are in agreement with the general behavior of the temperature dependency reported in Kourtidis et al., (1999) (Fig. S13 and Sect. S3), that performed an investigation of the dependence of the fractionation of NMHCs in evaporative emissions from temperature in Athens. Although the examined periods differ in ambient temperature (winter is colder than autumn), the exponential curve fitting of both datasets was similar. In addition, the above results could indicate why the Athens tunnel results performed in May differ from Patission and Thissio winter morning profiles. Moreover, the higher values of propane and butanes that are depicted in the morning peaks at the urban sites relatively to the tunnel measurement, reflect the increased number of LPG powered vehicles in Athens and natural gas-powered buses (Fameli and Assimakopoulos, 2016). This is further highlighted when the monthly variation of i-butane relatively to n-butane is examined (Fig. S14). The two compounds have linear relationship with no significant temporal differences on the slopes between the various months. Furthermore, the regression is similar to the one derived from the Patission measurements, thus enhancing our assumption that butanes emissions are traffic related. Moreover, the relation between the high levels of C2 – C4 alkanes and the number of LPG-powered cars was highlighted in other tunnel works as well (Ait-Helal et al., 2015; Zhang Q. et al., 2018).

To obtain a better idea on the variability of the traffic source during the studied period, the variability of selected NMHCs (ethylene, i-pentane and benzene) relatively to $BC_{ff}$, the latter used as traffic source tracer, was also plotted for October and December (Fig. 10). Significant correlations are revealed with slopes remaining almost stable (within 30%) during both months, indicating similar emission ratios during the whole studied period, and probably equal contribution from traffic.

*Night-time enhancement period:* During nighttime both $BC_{ff}$ and $BC_{wb}$ were maximized (e.g. Fig. 4 and 8), denoting significant contribution from both fossil fuel and wood burning (the contribution of the latter was more evident during

winter). Figure 11 presents the NMHC profile of the night-time enhancement period for October and December SP nights (details for the calculations are given in S.4). As already discussed, traffic is expected to be the main source of NMHCs during nighttime in October, whereas heating competes traffic during December. When these two profiles are compared (Fig. 11), the statistically significant difference at p<0.01 confidence is obvious, with a smaller contribution from i-pentane

(traffic source contributor) during December. In addition, enhanced contributions from C2 (ethane, ethylene and acetylene) are apparent in December compared to October. These C2 hydrocarbons have been reported as important contributors to the wood burning source profile by Baudic et al. (2016) in Paris. Preliminary data from a fireplace experiment (not part of this work) also confirm these findings; and are in line with our results reported in Fig. 8 indicating impact of wood burning during nighttime in winter months.

Figure 12a (i-iii) presents the relation of ethylene, acetylene and benzene, main contributors of the wood burning profile (Baudic et al., 2016), to BC during the SP night-time periods (18:00 – 05:00 LT), in October and December. During both months, significant correlations were revealed for all examined NMHCs and the slopes remained relatively stable, indicating almost equivalent emission ratios from both traffic and heating sources. To better tackle a possible difference in NMHCs emissions from traffic and residential heating, these NMHCs were also plotted against $BC_{wb}$ and $BC_{ff}$ during the SP periods

in December, from 22:00 to 04:00 LT, i.e. the time frame when traffic is quite limited (Fig. 12b, iv-vi). NMHC slopes versus $BC_{wb}$ are almost similar when compared to those versus $BC_{ff}$ (slight difference for ethylene), with a contribution of $BC_{wb}$ and $BC_{ff}$ to BC of 43% ($\pm$ 10%) and 55% ($\pm$ 11%) respectively, indicating that the studied NMHCs are probably equally produced by wood and fossil fuel burning.

**4 Conclusions**

For the first time to our knowledge, time resolved measurements of 11 Non Methane Hydrocarbons with two to six carbon atoms (C2 – C6 NMHCs) were conducted for several months (mid-October 2015 to mid-February 2016) in the Great Athens Area (GAA) by means of an automatic chromatograph, in parallel with monitoring of major pollutants and meteorological parameters. The temporal variability of NMHCs presented an increasing trend from October to December, due to changes in type and strength of sources, and atmospheric dynamics. In comparison with other works, higher concentrations are reported

for the majority of NMHCs, indicating an air quality issue in Athens. With the exception of isoprene, all NMHCs presented a bimodal diurnal pattern with morning and a broader night-time maxima, whereas the lower concentrations were observed early in the afternoon. Typical indicators of combustion processes such as CO and BC, which was further deconvoluted into $BC_{ff}$ and $BC_{wb}$, presented similar seasonal and diurnal variability relatively to the NMHCs, providing the opportunity to investigate their possible emission sources. Thus, the morning maximum, which follows the $BC_{ff}$ tendency, was attributed to

traffic, while the second one during night which maximized on December and coincides with those of $BC_{wb}$ and $BC_{ff}$ was attributed mainly to heating by both fossil fuel and wood burning.

For the better understanding of the impact of sources on the NMHCs levels, the studied period was further separated into smog (SP) and no-smog (nSP) periods, based on the absence of rainfall and low wind speed. October and December were

chosen for further comparison due to different temperature conditions and possible sources taking into account the already proved increased winter-time heating demand (Athanasopoulou et al., 2017). The comparison of the morning maximum of NMHCs profile during SP days with those obtained at a street canyon of Athens (Patission) further confirms the role of traffic in the observed morning NMHCs peak. The October and December SP NMHCs' night profiles depicted differences attributed mainly to heating. However, NMHCs slopes versus $BC_{wb}$ are almost similar when compared to those versus $BC_{ff}$ (slight difference for ethylene), indicating that NMHCs are probably equally produced by wood and oil fossil fuel burning. An extended dataset of NMHCs and other organic tracers (future long-term measurements) is needed to apportion different sources types on seasonal basis and quantify their impact on the NMHCs levels.

## 5 Data availability

All the data presented in this paper are available upon request. For further information, please contact Dr. Eleni Liakakou (liakakou@noa.gr).

## Acknowledgements

Support from CEA, CNRS and the Charmex program are acknowledged. We thank F. Dulac and E. Hamonou for the successful management of the Charmex program, D. Baisnée and T. Leonardis for technical support with the GCs, the editor and the two anonymous reviewers for their comments which greatly improved the submitted version. The authors also acknowledge the Hellenic Ministry of the Environment & Energy, Dept. of Air Quality for the access at the Patission Station and their database for the purposes of the short intensive campaign; and the Attiki Odos,for the access to the tunnel.

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

**Table 1. Comparison of NMHCs mean levels between this study and already published works in Athens, Greek and other Mediterranean or European sites. Information about the analyzing or sampling techniques and data resolution are included when available. The number of measurements[a] for each compound determined on the current samples is included below the table.**

| Studies | Rappenglück et al., 1998 | | Rappenglück et al. 1999 | Moschonas and Glavas, 1996 | Kaltsonoudis et al. 2016 | | Baudic et al., 2016 | Salameh et al., 2015 | Durana et al., 2006 | Current work | | | |
|---|---|---|---|---|---|---|---|---|---|---|---|---|---|
| Analysis details | GC – FID Every 20min | | GC – FID Every 20min | GC – MS 60 min (morning sampling, 12 canisters) | PTR-MS Every 10s/24h | | GC – FID | GC - FID | GC - FID | GC – FID Every 30min | | | |
| | 20 August – 20 September 1994, Athens, Greece | | 30 May – 16 June 1996, Athens, Greece | June 1993, May and July 1994, Athens, Greece | 3- 26 July 2012 (Demokritos) & 9 January – 6 February 2013 (Thissio) | | 16 October – 22 November 2010 Paris, France | 28 January – 12 February 2012 Beirut, Lebanon | April-October 1998-2001 February-July 2004 Bilbao Spain[b] | 16 October 2015 - 15 February 2016, Athens, Greece | | | |
| NMHCs | | | | | | | | | | Thissio (Urban background) | | | |
| | Patision (Urban) | Demokrirtos (Suburban) | Tatoi (Suburban) | Ancient Agora (urban) | Demokrirtos (Suburban) | Thissio (Urban background) | Les Halles station (Urban background) | Saint Joseph University (Suburban) | Bilbao (Urban center) | Mean | Median | Min | Max |
| | ppbv | | ppbv | ppbv | ppb | | ppb | ppb | ppbv | ppb | | | |
| Ethane | | | | | | | 3.8 | 2.8 | 2.5 - 3.5 | 4.5 | 3.1 | 0.6 | 25.9 |
| Ethylene | | | | | | | 1.3 | 2.1 | 2 - 2.3 | 4.1 | 2.2 | 0.3 | 22.9 |
| Propane | | | | 1.2 | | | 1.6 | 3.0 | 1.7 - 2.5 | 3.1 | 1.8 | 0.2 | 17.8 |
| Propene | | | | 3.9 | | | 0.4 | 0.6 | 0.7-0.9 | 1.5 | 0.6 | 0.02 | 15.7 |
| i-Butane | | | | 1.1 | | | 0.9 | 1.9 | 0.7-2 | 2.3 | 1.1 | 0.1 | 14.9 |
| n-Butane | 12.4 (with 1-butene) | 1.6 | 0.19 (with 1-butene) | 2.1 | | | 1.5 | 3.6 | 1.8 - 2.6 | 2.6 | 1.3 | 0.1 | 15.2 |
| Acetylene | | | | | | | 0.5 | 2.2 | 1.5 - 2.7 | 4.2 | 2.4 | 0.1 | 28.5 |
| i-Pentane | 26.3 | 3.2 | 0.93 | 11.7 | | | 0.7 | 2.4 | 1 - 1.7 | 4.7 | 2.6 | 0.2 | 23.8 |
| n-Pentane | 14.2 (with 2-methyl-1-butene) | 1.7 | 0.27 (with 2-methyl-1-butene) | 4.2 | | | 0.3 | 0.5 | 0.4 - 0.7 | 1.1 | 0.6 | 0.1 | 9.3 |
| Isoprene | | | 3.18(with trans-2-pentene & cis-2-pentene) | | 0.7 | 1.1 | 0.1 | 0.1 | | 0.2 | 0.1 | 0.01 | 1.4 |
| Benzene | 11.7 | 2.5 | 2.12 | 5.0 | 0.2 | 1.0 | 0.4 | 0.5 | 0.5 - 1 | 0.8 | 0.5 | 0.02 | 5.3 |
| Toluene | 21.2 | 6.7 | 1.15 | 14.3 | 0.8 | 2.3 | 0.8 | 2.2 | 2 - 2.6 | 2.2[d] | 1.0[d] | 0.1[d] | 13.7[d] |
| Ethylbenzene | 4.0 | 1.3 | 0.20 | 2.7 | | | | 0.3 | 0.6 – 0.8 | 0.4[d] | 0.2[d] | 0.03[d] | 2.7[d] |
| m-/p- Xylenes | 11.3[c] | 3.2[c] | 0.63[c] | 12.1 | | | | 0.4 | 2.0 – 2.4 | 1.2[d] | 0.5[d] | 0.03[d] | 8.3[d] |
| o - Xylene | 5.5 | 1.5 | 0.3 | 3.7 | | | | 0.3 | 0.4 – 0.5 | 0.4[d] | 0.2[d] | 0.03[d] | 3.1[d] |

a ethane N= 2848, ethylene N=2859, propane N=2861, propene N=2842, i-Butane N=2876, n-butane N=2879, acetylene N=2565, i-pentane N=2874, n-pentane N=2859, isoprene N=264, benzene N=2683, toluene N=637.

b Range estimated from Figure 1, included in Durana et al., 2006.

c Sum of the reported mean value for m – Xylene and p – Xylene.

d Only from 21 January 2016 to 15 February 2016

**Table 2: Enhancement ratios of NMHC to NOx (ppb/ppb) and to CO (ppb/ppb), calculated from the present data-set for Thissio Station and the Street canyon measurements (Patission station) for the time-window of the traffic rush hours. The enhancement ratios presented in the 3rd and 6th column are reported in Kourtidis et al., (1999) and they were calculated for the same station in the street canyon.**

| Ratios of NMHCs to: | NOx (ppb/ppb) | | | CO (ppb/ppb) | | |
|---|---|---|---|---|---|---|
| | Thissio station (urban background) | Patission station (traffic) | Patission station (traffic), **1994**[*] | Thissio station (urban background) | Patission station (traffic) | Patission station (traffic), **1994** |
| | 21 January - 15 February 2016 | 23 - 24 February 2017 | 20 August - 20 September 1994 | 21 January - 15 February 2016 | 23 - 24 February 2017 | 20 August - 20 September 1994 |
| i - Pentane | 0.0639 | 0.0490 | 0.2468 | 0.0072 | 0.0058 | 0.0098 |
| Benzene | 0.0095 | 0.0083 | 0.1042 | 0.0012 | 0.0009 | 0.00414 |
| Toluene | 0.0417 | 0.0320 | 0.1799 | 0.0056 | 0.0034 | 0.00715 |
| Ethylbenzene | 0.0073 | 0.0053 | 0.0338 | | | |
| o - Xylene | 0.0082 | 0.0059 | 0.0471 | | | |

5   [*]The NMHC-to-NOx enhancement ratios of Kourtidis et al. given in w/w (weight/weight) were converted in ppb/ppb by dividing them with the ratio of the molecular weight of the NMHC to the molecular weight of NOx (equal to 31.6 according to Kourtidis et al., 1999)

Table 3. Correlation coefficients ($R^2$) of NMHCs and major gaseous pollutants for the total period of measurements (all significant at $p < 0.01$).

| | Ethane | Ethylene | Propane | Propene | i-Butane | n-Butane | Acetylene | i-Pentane | n-Pentane | Benzene | BC | $BC_{wb}$ | $BC_{ff}$ | CO |
|---|---|---|---|---|---|---|---|---|---|---|---|---|---|---|
| Ethane | | | | | | | | | | | | | | |
| Ethylene | 0.94 | | | | | | | | | | | | | |
| Propane | 0.92 | 0.94 | | | | | | | | | | | | |
| Propene | 0.94 | 0.97 | 0.96 | | | | | | | | | | | |
| i-Butane | 0.82 | 0.90 | 0.95 | 0.92 | | | | | | | | | | |
| n-Butane | 0.84 | 0.91 | 0.97 | 0.92 | 0.99 | | | | | | | | | |
| Acetylene | 0.89 | 0.91 | 0.90 | 0.91 | 0.88 | 0.88 | | | | | | | | |
| i-Pentane | 0.73 | 0.85 | 0.88 | 0.85 | 0.96 | 0.95 | 0.81 | | | | | | | |
| n-Pentane | 0.74 | 0.85 | 0.90 | 0.88 | 0.97 | 0.96 | 0.84 | 0.96 | | | | | | |
| Benzene | 0.87 | 0.95 | 0.93 | 0.96 | 0.91 | 0.92 | 0.89 | 0.87 | 0.89 | | | | | |
| BC | 0.93 | 0.95 | 0.92 | 0.96 | 0.88 | 0.89 | 0.90 | 0.84 | 0.85 | 0.93 | | | | |
| $BC_{wb}$ | 0.91 | 0.87 | 0.81 | 0.89 | 0.70 | 0.72 | 0.77 | 0.65 | 0.64 | 0.83 | 0.91 | | | |
| $BC_{ff}$ | 0.84 | 0.90 | 0.89 | 0.90 | 0.91 | 0.91 | 0.89 | 0.89 | 0.90 | 0.89 | 0.95 | 0.75 | | |
| CO | 0.91 | 0.95 | 0.94 | 0.96 | 0.92 | 0.93 | 0.92 | 0.87 | 0.89 | 0.95 | 0.97 | 0.87 | 0.93 | |
| NO | 0.86 | 0.90 | 0.90 | 0.90 | 0.90 | 0.91 | 0.89 | 0.90 | 0.88 | 0.89 | 0.91 | 0.76 | 0.92 | 0.94 |

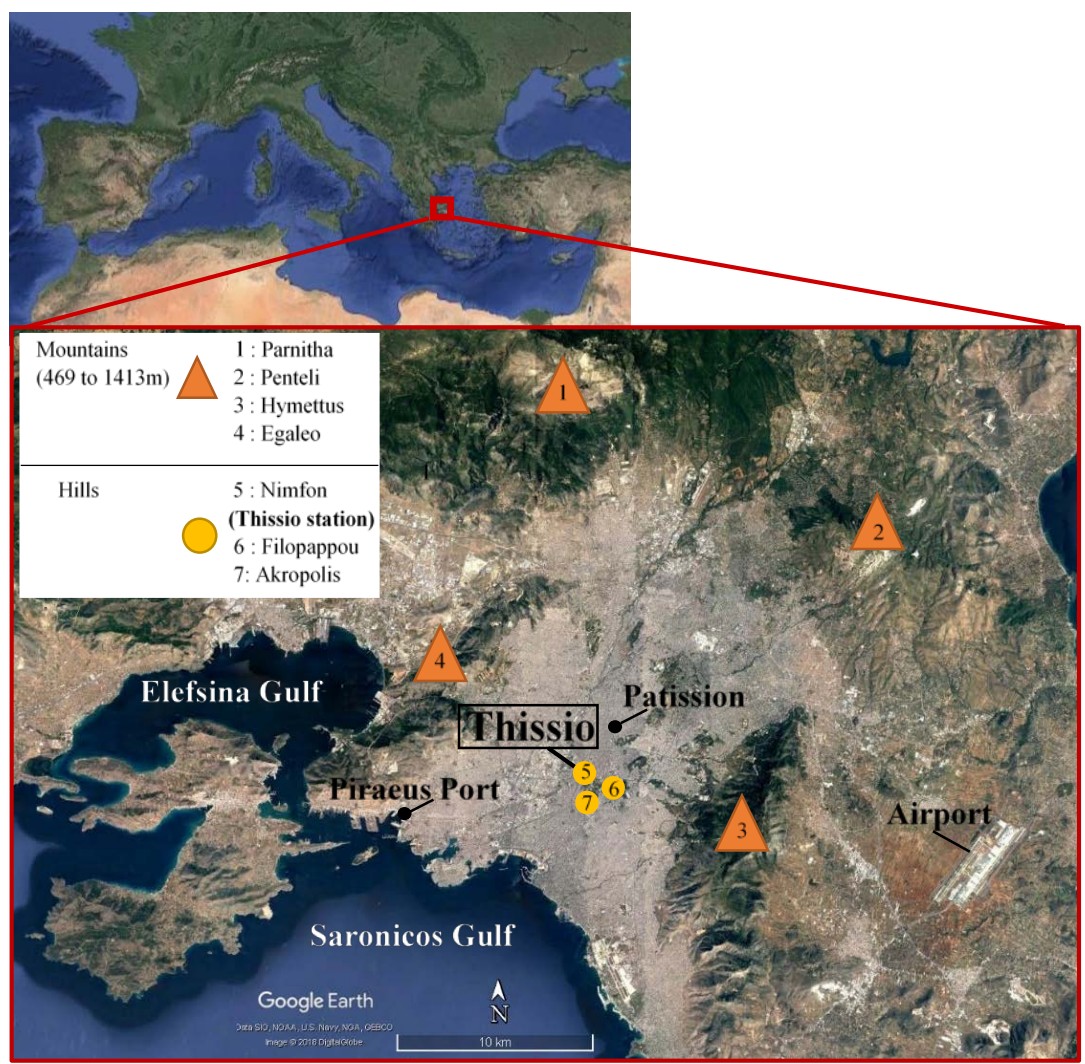

**Figure 1. Map of the Greater Athens Area. The four mountains define the borders of the area.**

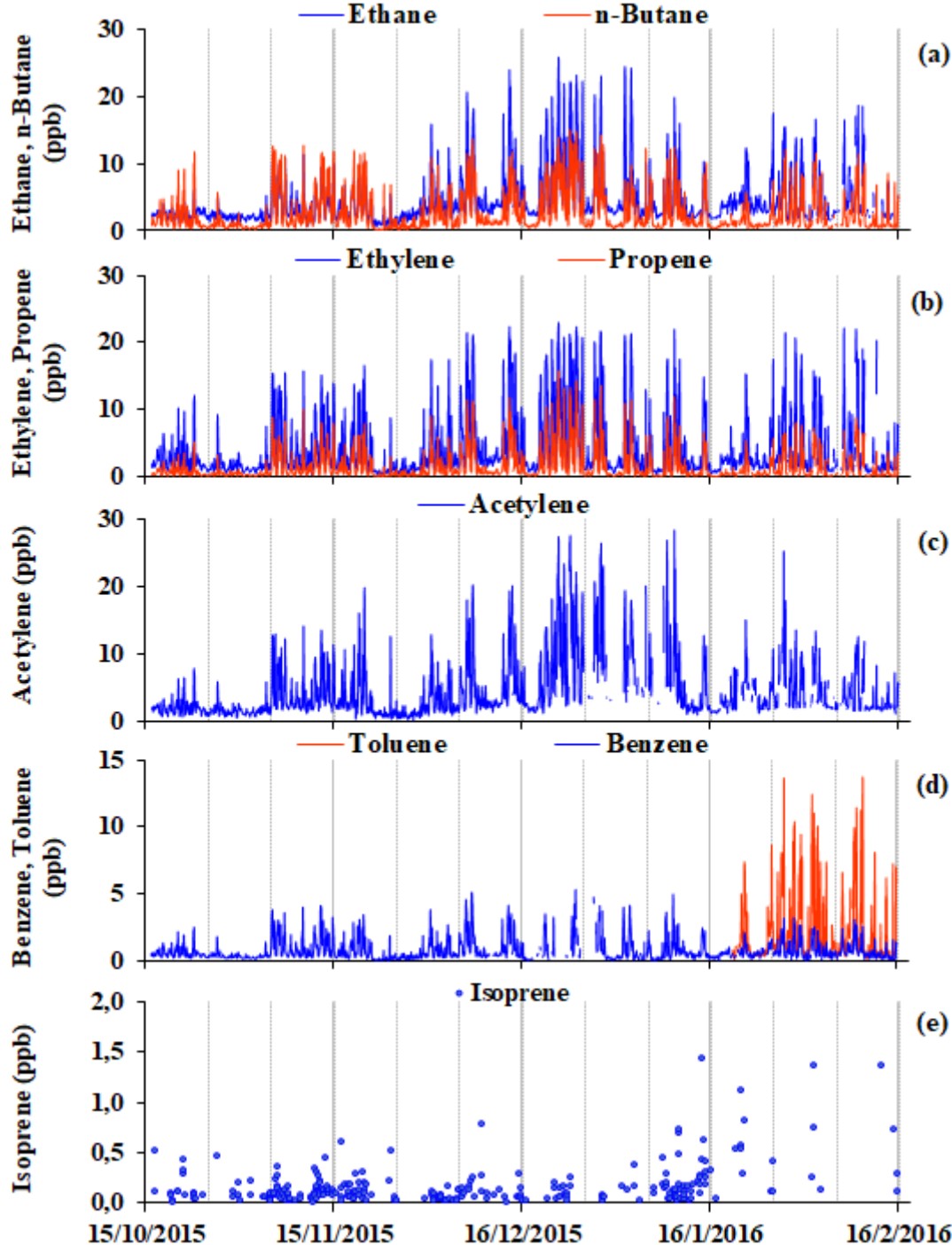

**Figure 2. Temporal variability of (a) ethane and n-butane, (b) ethylene and propene, (c) acetylene, (d) benzene and toluene and (e) isoprene, based on hourly averaged levels for the period 16 October 2015 - 15 February 2016, at NOA's urban background site in Thissio, downtown Athens.**

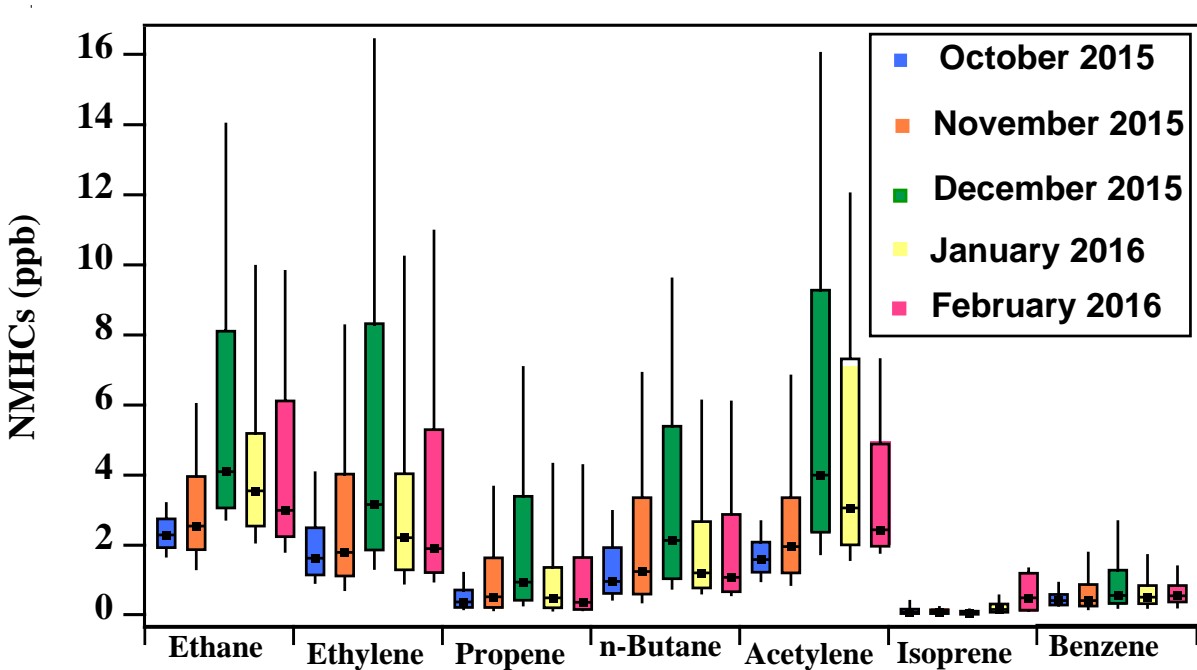

**Figure 3. Monthly box plots for ethane, ethylene, propene, n-butane, acetylene, isoprene and benzene. The black dot represents the median value and the box shows the interquartile range. The bottom and the top of the box depict the 1st and 3rd quartiles (i. e. Q1 and Q3). The whiskers correspond to the 1st and the 9th deciles (i. e. D1 and D9).**

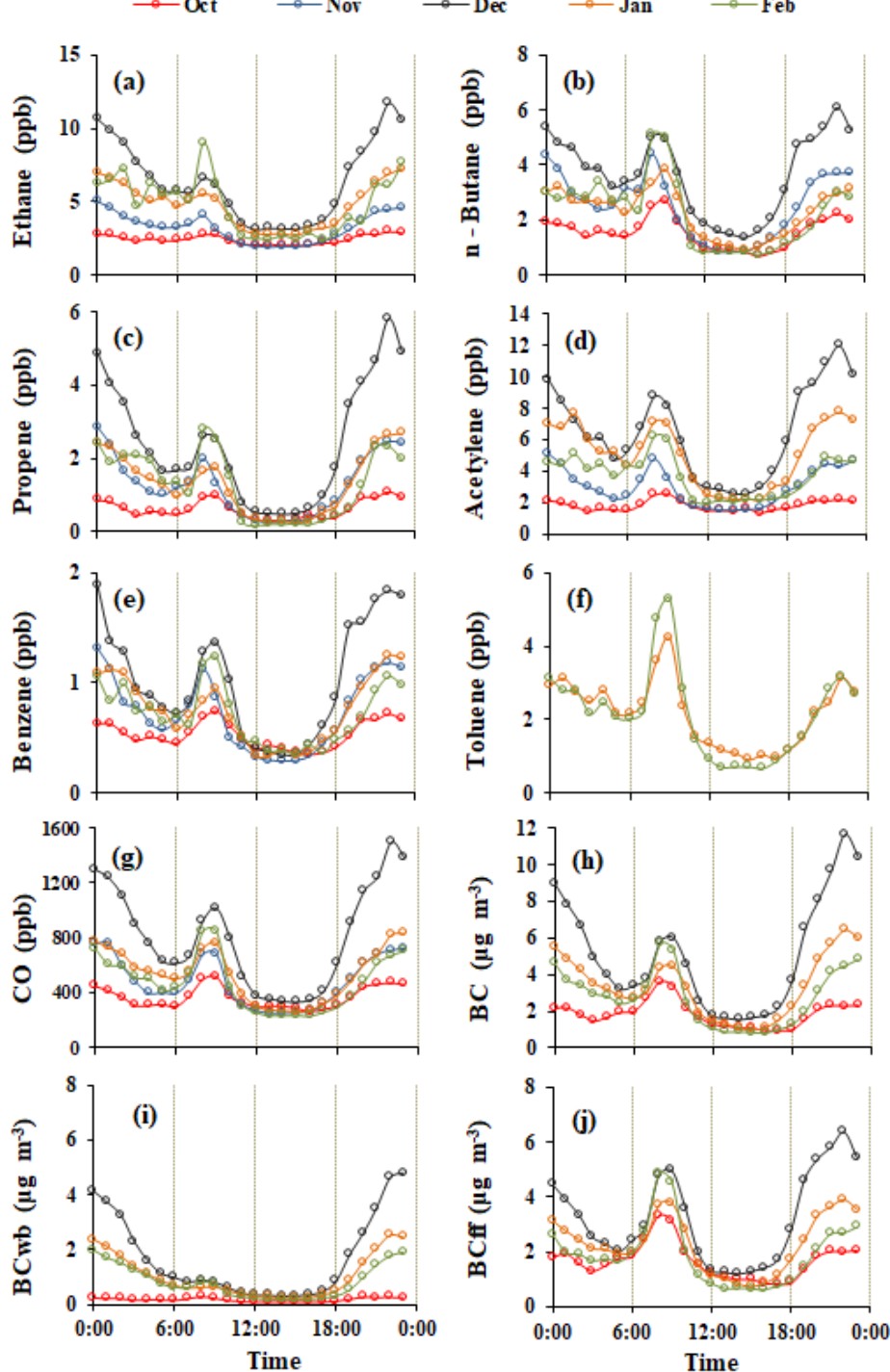

**Figure 4. Monthly diurnal variability of (a) ethane, (b) n-butane, (c) propene, (d) acetylene, (e) benzene, (f) toluene, g) CO, h) BC, i) BC<sub>wb</sub> and j) BC<sub>ff</sub> based on hourly averaged values.**

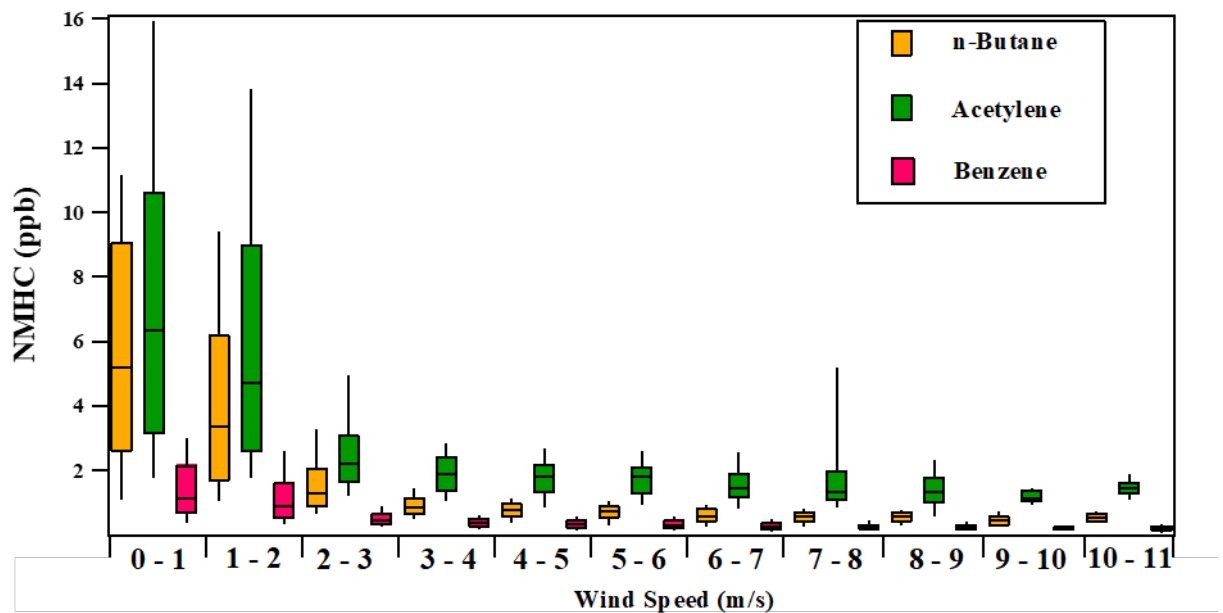

**Figure 5. Boxplots for (a) n-butane, (b) acetylene and (c) benzene relatively to wind speed for the period 16 October 2015 - 15 February 2016. The black line represents the median value and the box shows the interquartile range. The bottom and the top of the box depict the 1st and 3rd quartiles (i. e. Q1 and Q3). The whiskers correspond to the 1st and the 9th deciles (i. e. D1 and D9). The range of each wind speed bin is depicted on x-axis.**

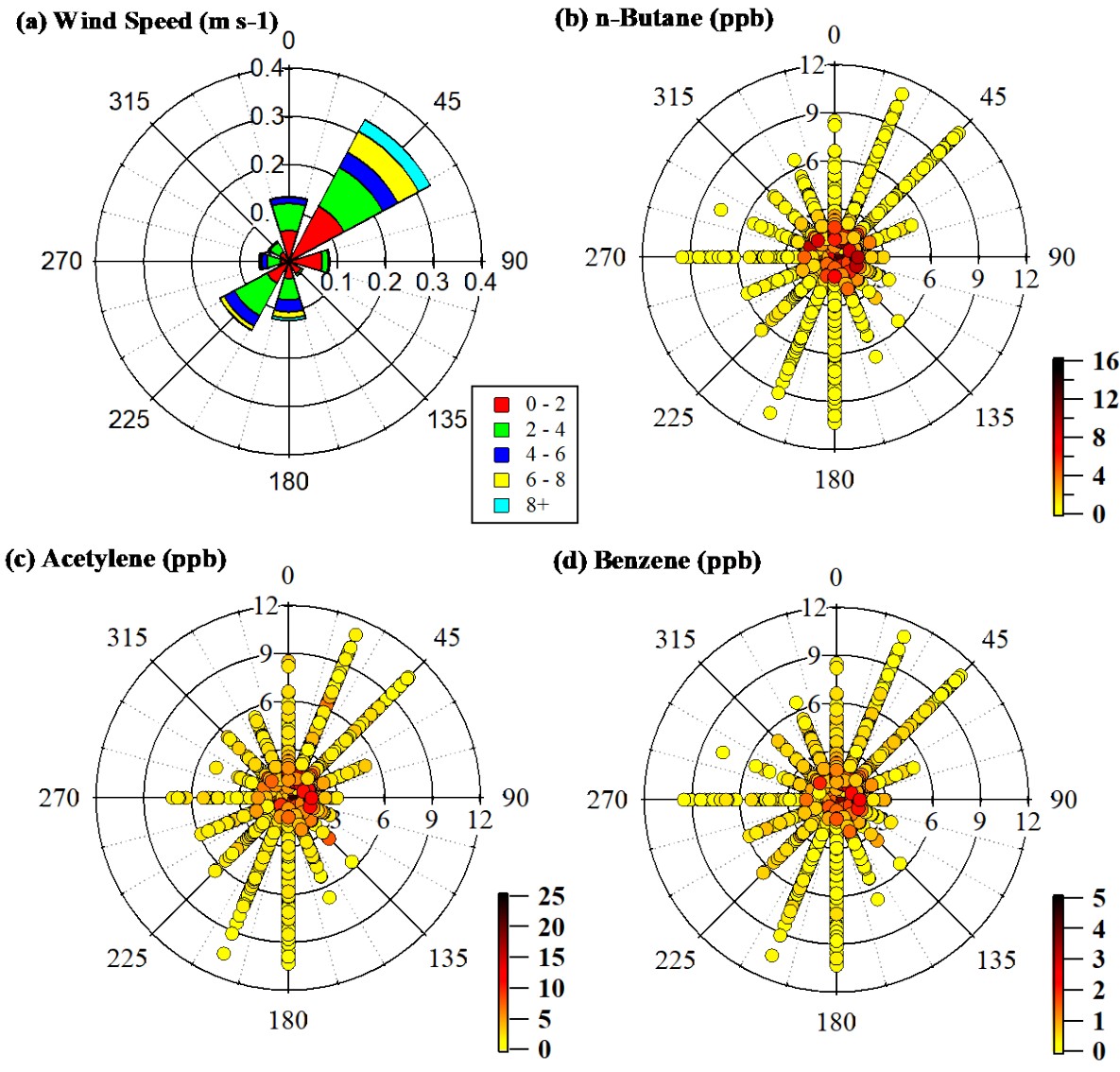

**Figure 6.** Wind rose (a) and concentration roses of (b) n-butane, (c) acetylene, and (d) benzene for the period 16 October 2015 to 15 February 2016.

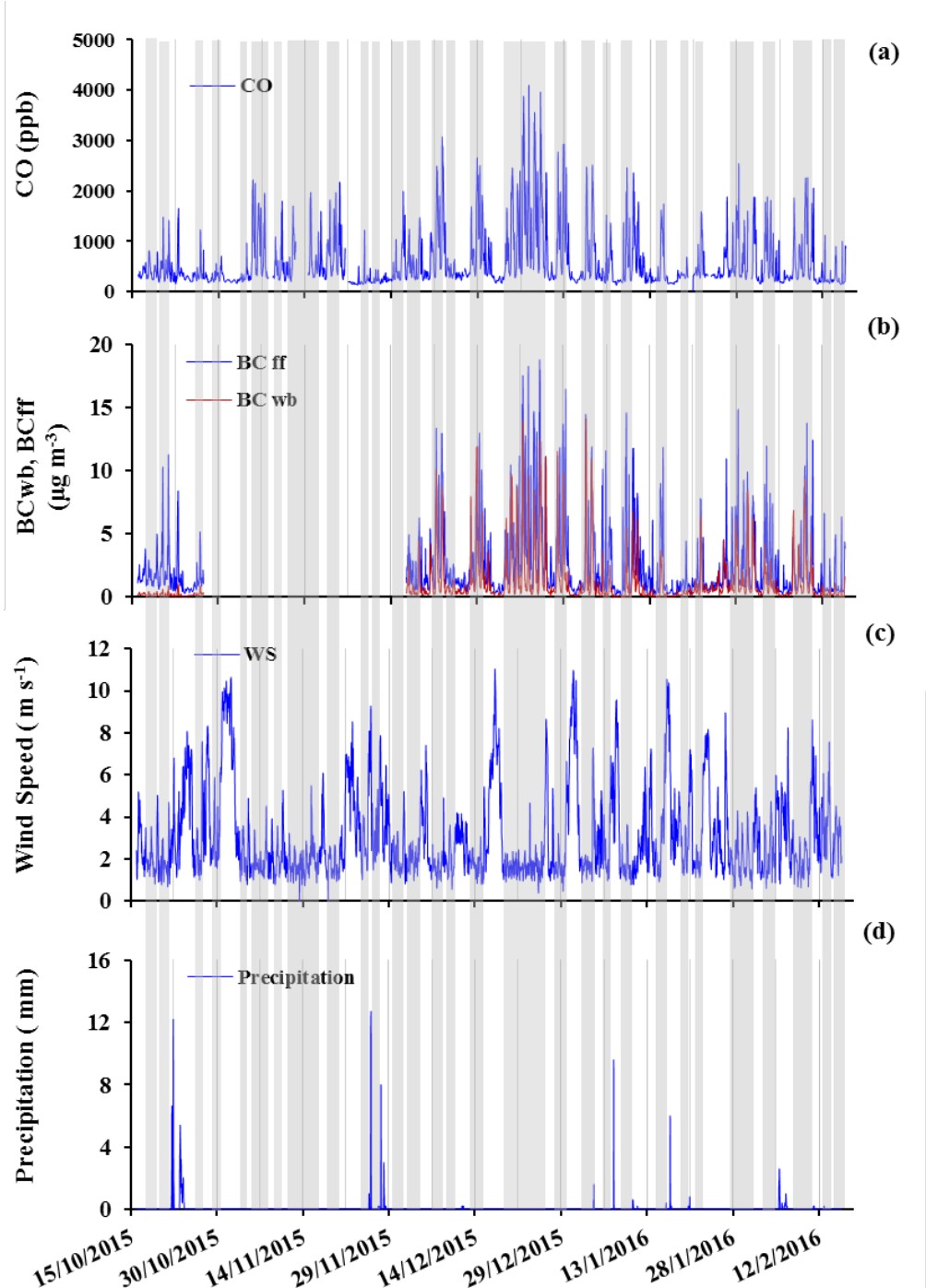

**Figure 7. Temporal variability of (a) CO, (b) BC$_{wb}$ and BC$_{ff}$ fractions, (c) wind speed and (d) precipitation for the experimental period. Grey frames correspond to smog periods (SP), while the remaining part to non-smog periods (nSP).**

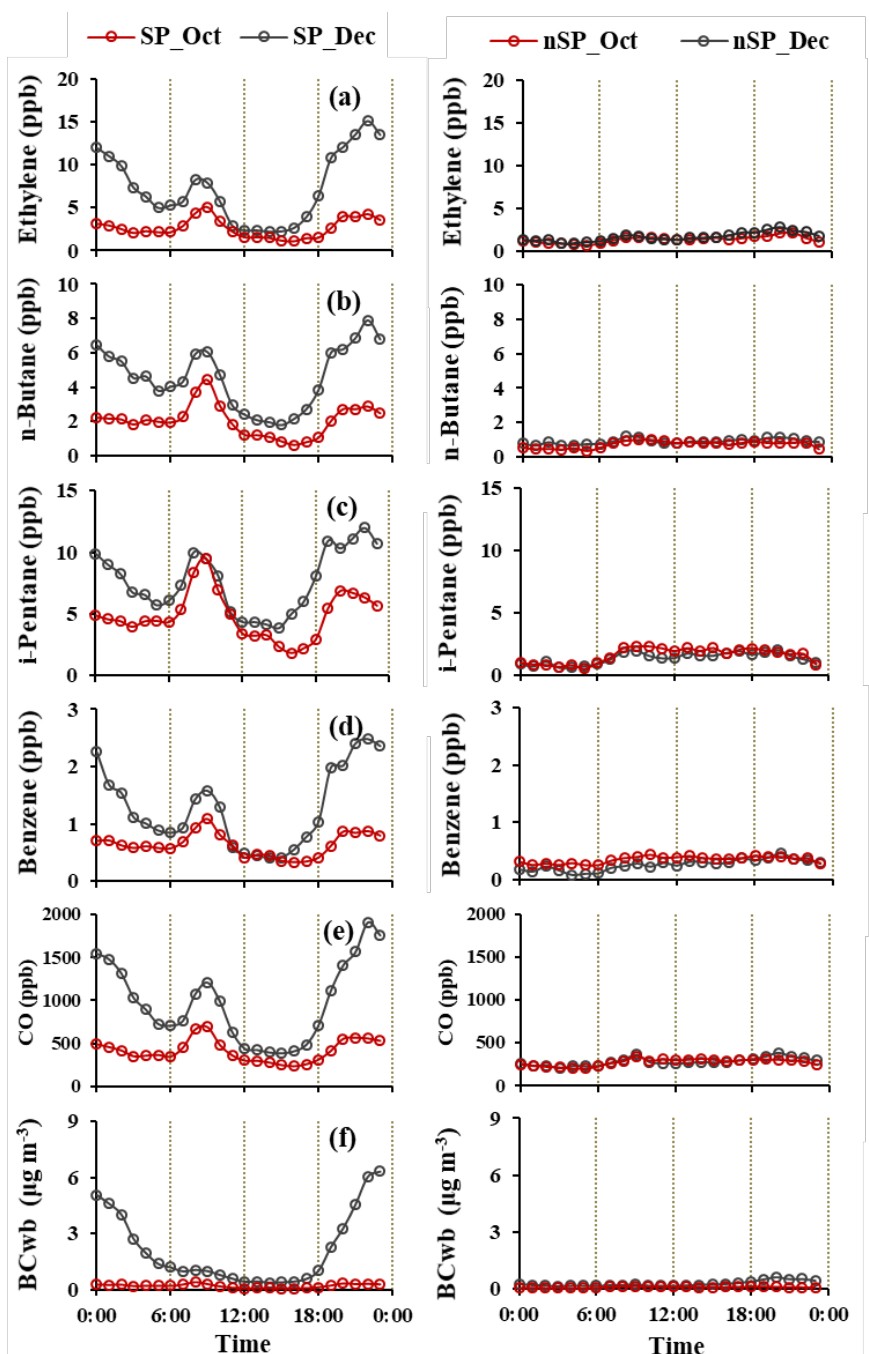

**Figure 8. Diurnal patterns of (a) ethylene, (b) n-butane, (c) i-pentane, (d) benzene, (e) CO, (f) BC_wb during the SP (left column) and the nSP (right column) periods identified during October 2015 (red) and December 2015 (black) respectively. Note: SP periods are defined by wind-speed lower than 3 m s$^{-1}$ and absence of rainfall, while nSP periods are defined by winds-speeds higher than 3 m s$^{-1}$ .**

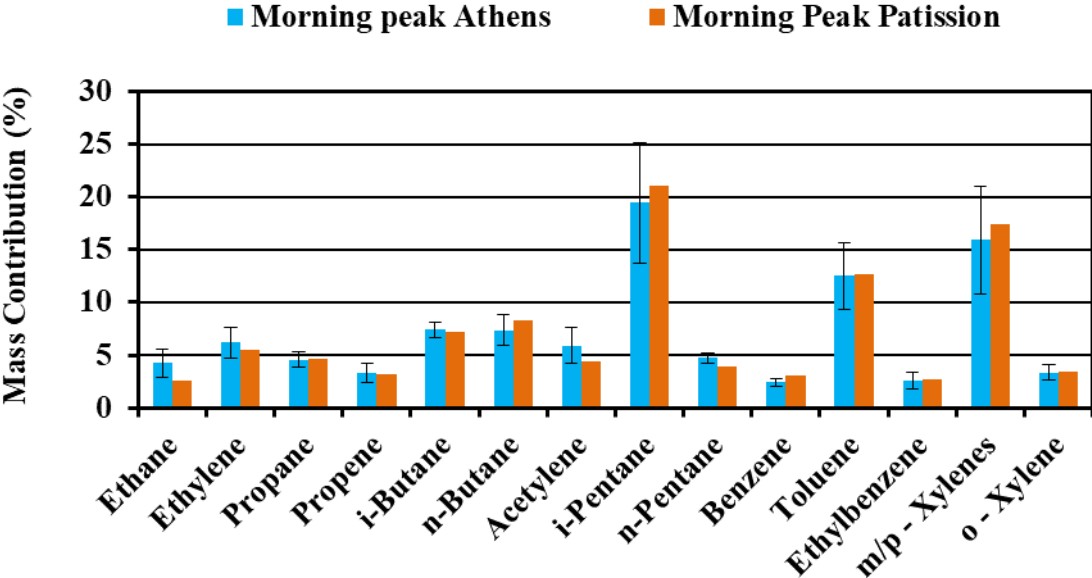

**Figure 9**. %Mass contribution of the measured NMHCs during the morning peak (07:00 – 10:00LT), median values in Thissio and mean values in Patission Monitoring Station.

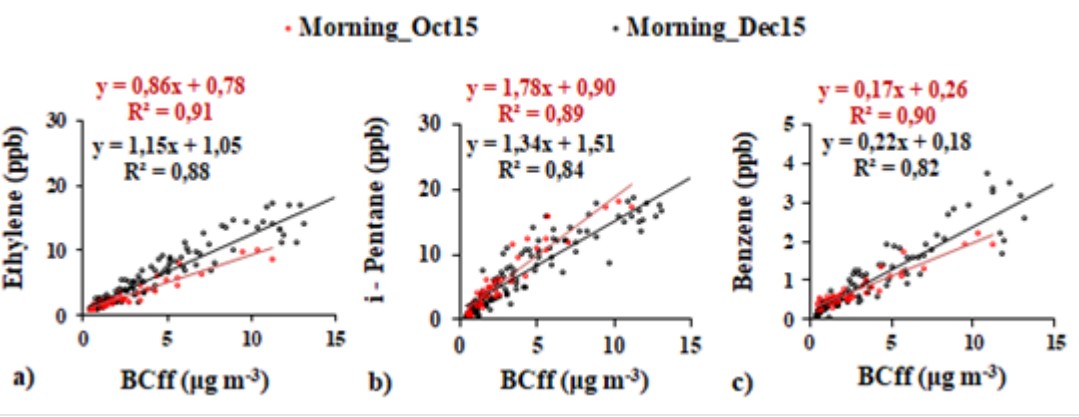

**Figure 10**. Regressions between ethylene, i-pentane, and benzene versus BC$_{ff}$ (a-c) for the morning period (07:00 – 10:00LT) in October and December 2015.

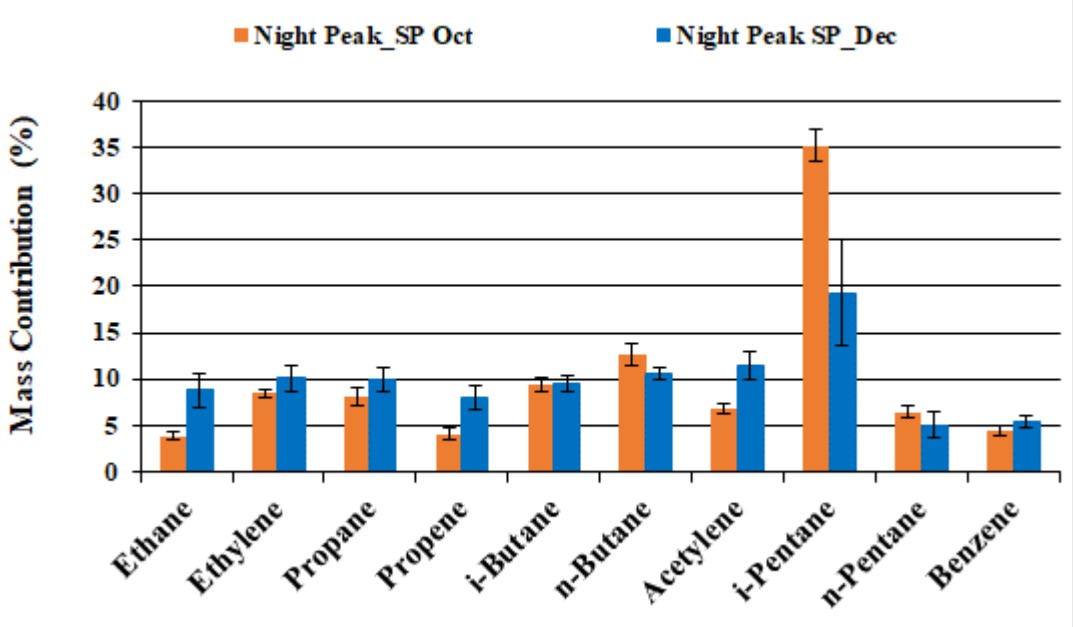

**Figure 11. % Mass contribution of the measured NMHCs during the night-time enhancement period (18:00 – 05:00LT) for the SP of October (orange) and the SP of December (black color).**

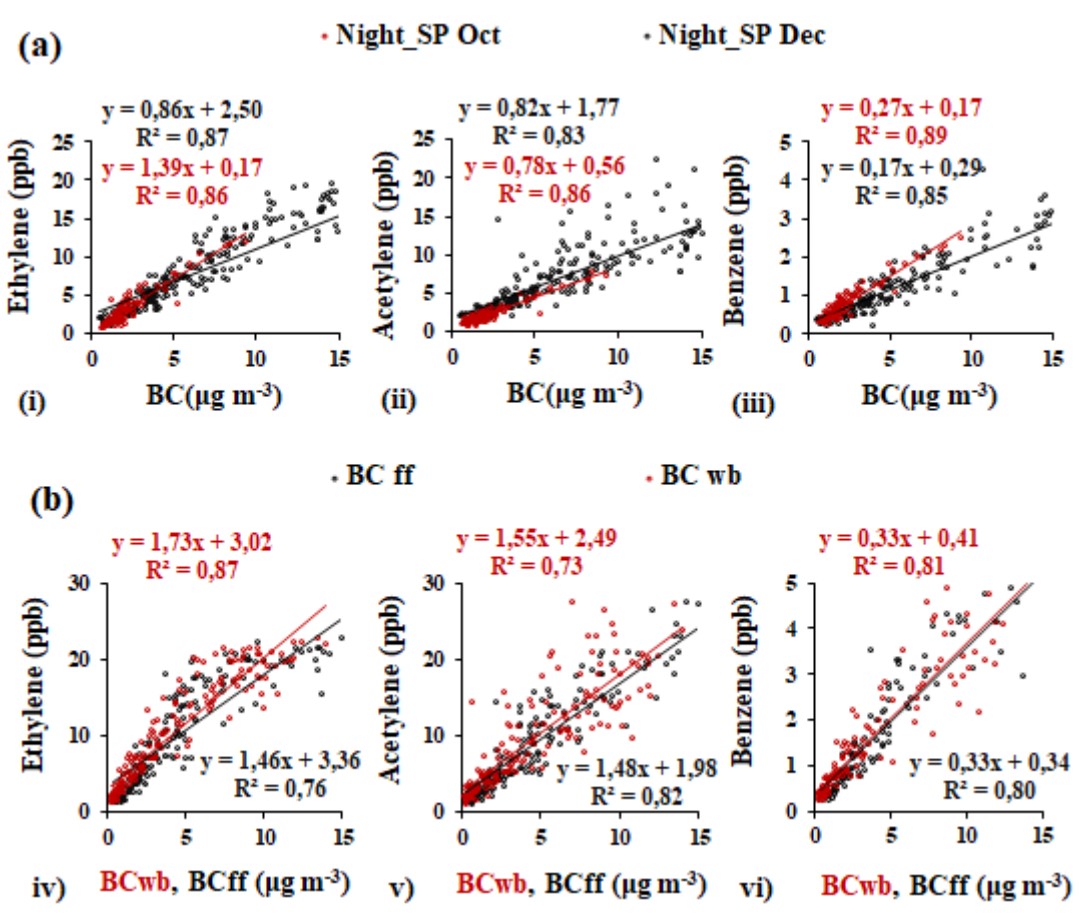

**Figure 12**. **Regressions between ethylene, acetylene and benzene (a) against BC (i-iii) for the night period (18:00 – 05:00LT) of SP October and December 2015 and (b) against BC$_{wb}$ (red) and BC$_{ff}$ (black) for the night period (22:00 – 04:00LT) of SP December 2015.**

