# Peer review of "Non Methane Hydrocarbons variability in Athens during wintertime: The role of traffic and heating"

_Atmospheric Chemistry and Physics, 2017_

## Referee Comment (RC1) · Anonymous Referee #1 · 21 Dec 2017

General comments:

The article presents a very nice data set of a range of VOCs from the Central Athens area during a five month measurement period. The data will be of use to the wider community for general interest as well as for comparison with datasets from other Cities. There article features some simple plots presenting the data in an informative way and the authors draw some conclusions about sources of VOCs and reasoning behind the seasonal changes observed. I do feel, however, that there is insufficient evidence presented here to support the authors' claim that the boundary layer height change between October and December is not the main cause of the observed increase in

[Figure]

VOC mixing ratios. I would suggest the authors either re-phrase the sections relating to this (to include the possibility of meteorology playing a major role) or provide more evidence in support of this. Despite this I would suggest that there is sufficient material here and am in favour of publication of this article.

Technical comments/corrections: (Note the numbers listed here indicate page number followed by line number (eg. 2, 11 indicates page 2, line 11).

Throughout the article: "C2-C6"

Should be written as C2 – C6

2, 11: "Athens, the capital of Greece and an important megacity,…"

Is Athens a megacity? Generally a megacity is considered as one with a population of more than 10 million. Further justification is required or this should be removed.

2, 27: Comment "The above demonstrate the increasing need for intensive measurement of NMHCs in Athens, to better understand their sources, temporal characteristics and role on smog formation, in the new conditions established during the economic crisis years, with competing traffic and wood burning." Perhaps more fundamentally there is need to observe the current atmospheric composition to allow for the impact of future changes (fuel composition changes or other control strategies) to be assessed. There is a need to establish a "current baseline" for Athens.

3, 24: "The trap was then heated rapidly to 22 °C…"

Is this a typo? Compounds won't desorb well (if at all) at these temperatures. Should this read 220 °C?

3, 28: "The overall estimated uncertainty of the measurement is 15%."

A more detailed discussion of the measurement uncertainty is required to understand how this value is derived. Which parameters are included within the uncertainty? Does the uncertainty vary with compound type? Is it dependent upon mixing ratio or constant across the measurement range? These are all important details which should be included here.

4, 28: "...with the exception of isoprene (approximately 10%)."

Why was the data coverage for isoprene worse than others? Were there interferences? This is important to inform other potential users of this equipment for the measurement of VOCs and also needed to confirm that the data quality of the other VOC measurements wasn't impacted by these issues.

5, 1: "... the significant night time levels (above 300 ppt in some cases)..."

Can this data be trusted given the aforementioned problems with the isoprene data coverage? Further discussion may be needed (unless this is covered in the explanations of the isoprene data coverage above).

5, 5: "... while lower values were below 5 ppb for the whole period."

I'm unsure what this statement means, clarification is needed. Does this relate to ethane and ethylene or the other VOCs measured?

5, 8: "The average concentration of benzene during the studied period was 0.7 ppb (still not a full year), which is considerably below the EU average annual limit of 5 $\mu$g m-3 or 1.5 ppb (Directive 2008/50/EC of the European Parliament)."

The data presented here doesn't include summer time values where we'd expect lower mixing ratios. If these were included then presumably the value would fall well below the threshold. This should be included here, it is of interest in- and of- itself, but also leads the reader to question whether the current directives are suitable and adequate?

5, 12: "The comparison with those already published for the GAA, indicates an apparent decrease by a factor of 2 to 6 for the majority of the species lying above C4 (taking as reference the case of Ancient agora urban area in the close vicinity of the Thissio Station), always bearing in mind differences in sampling period (summer versus winter), location, sampling method and analytical techniques."

Is it possible to estimate of the actual decrease? This would be of interest here, despite the various caveats that must be included.

5, 20: "Furthermore, our findings for benzene and toluene, were significantly lower than the 12 hour day-time average levels reported for a Cairo rural background area, as reported by Khoder et al., 2007 (mean levels of 5.8 and 7.5 ppb respectively)."

I don't see the significance of this statement? It needs expanding to make its relevance clear.

5, 23: "... pattern for all NMHC concentrations was their gradually increase from October..."

Typo gradually to gradual

5, 29: "... according to Kokkalis (personal communication) the winter-time decrease of PBL is in the range of 20%,..."

This is an important statement in the context of this article and needs more detailed supporting evidence. Later figures and text attempt to reaffirm this statement to support the implication that source-changes define the changes in VOC composition. In its current format I don't see a convincing argument to support this. Either the authors need to include substantial supporting evidence for this or the text should be altered to include the possibility that the variations in VOC mixing ratios could be due to changes in Meteorology.

6, 7: "Although the amplitude of both peaks is almost similar (with the exception of December), the duration of the night peak is at least a factor of 2 larger, indicating the predominant role of heating in air quality during wintertime."

This would also be conducive with boundary layer dynamics dictating the night time profile.

6, 25: "...the most frequent, resulting to moderate levels of NMHCs."

Typo "to" to "in"

7, 2: "...2 to 3 times higher levels of NMHCs were observed on December..."

Typo "on" to "in"

7, 7: "When NMHCs are examined against temperature (not shown here), a clear tendency is not evident, although the highest levels occur at lower temperatures."

A plot of this would be useful to be included here.

7, 21: "More precise picture..."

Typo "More" to "A more"

8, 15: "The most striking difference is related to the night and early morning peak, while during mid-day the difference is Minimum."

This requires more detail. While the concentration rise is smaller at mid-day, the relative rise looks to be more or less the same for the early morning and mid-day periods (approximately double). Including the percentage increase would clarify this.

8, 16: "...while during mid-day the difference is minimum...."

Typo "minimum" to "minimal"

8, 18: "For the nSP cases (Fig. 7 and S7) the concentrations of all compounds were very low (lower than the minimum of the SP periods) and almost equal, with the exception of ethane and acetylene that demonstrated higher concentrations in December by a factor of two (Fig. S7a,e)."

I can't make sense of this sentence, it needs re-writing/re-phrasing for clarity

8, 32: "It is interesting to note that the profiles, especially those derived from the morning peak, nicely fit with that reported for traffic by Baudic et al. (2016) in Paris (when

only the common NMHCs measured in this work have been used)."

I don't see a "nice fit" they look different in magnitude and relative composition. Please provide more detail of what the authors are implying by this.

9, 2: "... butanes are however be noted in the morning peak..."

Typo "be" needs to be removed

9, 4: "Toluene, an important contributor to the traffic profile (Fig. 8), was measured only for one month during winter."

Does this affect the plot? Is that why the toluene is lower than at other sites? This sentence needs expanding upon to clarify its effect, if any, upon the figure.

9, 12: "...(the contribution of the later was more evident during winter)."

Typo "latter" not "later"

Fig 7 caption: I suggest including the definition of SP and nSP within the figure caption

―――――――――――――

---

## Referee Comment (RC2) · Anonymous Referee #2 · 4 Jan 2018

The authors present an analysis of a continuous time series for selected C2-C7 NMHCs for a site in Athens, Greece, during wintertime 2015/16 and investigate this data with respect to the contribution of traffic and heating to the observed ambient NMHC levels. They also include speciation of BC data, which is either related to wood burning (BCwb) or fossil fuel combustion (BCff). They also include NO and CO data. While I see the value of this larger online NMHC data set, I am disappointed to see how this data has been presented and analyzed. Unfortunately, there are three major points, which - in my opinion - do not warrant its publication in its current stage in ACP without critical major revisions:

1) On the background of existing literature I am not sure what the real novelty of this paper in terms of methods and results is. While I agree that C2 and C3 measurements have not yet been done before in Athens, it seems the inclusion of these does not yield more findings than already reported by Kaltsonoudis et al (2016). On the other hand, the Panopoulou et al paper makes same sketchy description of meteorological impacts on NMHCs, but lacks some elaborate analysis similar to those presented in Rappengluck et al (1998) for transport effects and also Kourtidis et al (1999) for temperature effects for Athens. As the authors make an important point on page 2, L27-29, that there have been new conditions during the economic crisis years (i.e. competing traffic vs wood burning emissions) it would be actually meaningful to perform a comparative analysis between the data sets reported 20 years ago and the ones reported by Panopoulou et al. It seems both studies include continuous NMHC measurements and PMF source apportionment analysis would be feasible and would provide interesting insights.

2) It seems that for many statements made in this paper, statements can be made in contrast to the statements made by the authors.

3) Many statements and presentations were made in a negligent way.

Some more detailed review below:

Abstract:

Page 1 L18: "..among the highest in literature for the Mediterranean area..." It should be clarified if this refers to the same season or not. It is known that many primarily emitted pollutants reach higher values in wintertime.

Page 1 L20-21: What do the authors exactly mean by local meteorology, as this term is quite unusual? Its connotation would mean that it is not representative for a larger fetch. I disagree that "local" meteorology would control the variability of NMHC levels alone. What about the temporal variability of NMHC emissions?

Page 1 L24-25: Why would the fact that NMHCs nicely follow CO hint towards additional sources e.g. heating? This can also be true in the case of traffic emissions. Also, heating is a very general term. For instance electric heating would not emit CO and NMHCs locally (just to name one kind of heating)?

Page 1 L27-29: Why does the present data not allow for the quantification of the relative contribution of fossil fuel and wood burning for heating purposes?

Introduction:

Page 2 L3-4: This is only true for urban areas!

Page 2 L6: replace "Baudic et al., 2016b" by "Baudic et al., 2016".

Page 2 L7: This EEA report is not properly cited in the list of references.

Page 2 L14: The "Cvitas et al paper" should be properly cited as its primary appearance is in a journal.

Page 2 L14: As far as I understand it, the Kourtidis et al paper analysed online NMHC data contrary to what the authors state.

Page 2 L20-21: The term "limited" is not properly chosen, as it seems the range of the NMHC data and the number of NMHCs reported by Panopoulou et al. is also limited. It would be fair to mention the entire C range of those earlier measurements which Panopoulou et al cite in order to properly put them into perspective. Also, it looks to me that those measurements contained significantly more speciated NMHCs, and according to table 1 these earlier NMHC measurements include NMHCs with four C atoms contrary to what the authors state here ("...containing more than four atoms of carbon...").

Page 2, L24: The Saffari et al paper is on the Thessaloniki case, not on the Athens case.

Page 2, L26: Again, as mentioned above the term "limited" is not properly chosen.

The Kaltsonoudis et al paper actually reports VOCs, which are not measured by Panopoulou et al. Also, while it is true that the Athens winter campaign reported by Kaltsonoudis et al was shorter than the one reported by Panopoulou et al, it seems that Kaltsonoudis et al also report a summer campaign for Athens, which the Kaltsonoudis et al paper does not.

Page 2 L30: It would be fair to mention how many NMHCs were actually measured as it seems that the paper does not report some important NMHCs such as 1,3-butadiene and others, for instance.

Page 2 L34 - page 3 L1: The authors should mention why the analysis is restricted towards traffic and heating impact on NMHC levels.

Page 3 L3: Mention those selected tracers explicitly and mention what kind of sources those tracers are tracing.

Sampling site

In general: I suggest to include a map here. Not everyone is familiar with Athens

Page 3, L7: Actually, more important than the altitude above sea level would be the altitude of the hill site above the surrounding area.

Page 3 L9: How far away from the site are the Filopappou and Acropolis hills and how high are those? Again, a map would be helpful. It is not sufficient to refer to other publications here, as it seems the site the authors are talking about is a very specific one.

On line NMHC measurements

Page 3, L7-8: What do the authors mean by " ...was set to sample on a 10 min basis..."? Do they refer to the sampling time or sampling frequency? Is the GC continuously flushed with ambient air?

Page 3, L20: The sampling line has a pretty small diameter, which is very unusual. The

authors should clarify why they chose that small diameter. Also, why did the authors use a stainless-steel sampling line and not a glass line, which will have the least interaction with the sample, in particular when considering this very long line (6 m)? Did the authors check the sampling line for any potential losses of NMHCs, e.g. through looking into any deviations of the C-response? Did the authors make calibrations directly to the GC or through the sampling line? Did the authors use any filter at the sampling inlet?

Page 3, L28-29: Is this uncertainty true for all VOCs? Usually, it would be class specific. What are the detection limits?

Page 3, L30: Same as mentioned above with regard to the sampling line.

Page 4, L5: Why is only toluene used? Why not at least ethylbenzene and the xylenes in addition? Would the exclusion of these NMHCs not introduce a bias into the data analysis, as important tracers for solvent emissions are excluded? What are the uncertainties and the detection limits for this GC?

Auxiliary measurements

In general: The authors also report NO measurements at some point in the paper, but neglect to mention the instrumental description here.

Tunnel measurements

Page 4 L16: The authors should mention the length of the tunnel, whether lanes were for both directions (there could also be dedicated tunnels for one direction only), if there was any artificial ventilation and if there might have been any limitations on traffic through this tunnel (in some cases heavy duty traffic is not allowed). In any case an estimate of the traffic fleet composition (e.g. heavy duty vs light duty vehicles) would be helpful. All these factors have an impact on the NMHCs levels. At what location of the tunnel did the authors make the measurements exactly? I see the measurements were taken on 12 May 2016, which is different from wintertime. Wouldn't the temperature

be different from wintertime and wouldn't this have an enhanced impact on NMHC emissions through evaporation, for instance?

Page 4 L19: What type of canisters were used?

Page 4 L20-21: Why was it needed to dilute the sample by a factor of two? Why was a Teflon transfer line used? What kind of Teflon in particular? Did the authors check any NMHC artifacts in the canisters and the sampling path (i.e. Teflon line)?

Temporal variability of NMHCs

Page 4 L28: I do not understand the concept of data coverage here, as it is not explained. It could refer to the percentage of data above the detection limit vs maximum available data, but this does not make complete sense, as I doubt there were any data of ethane below the detection limit, for instance. However, it cannot be true either that it refers to the data availability vs maximum potential data availability during the time period reflecting instrumental potential instrumental malfunctions and/or failure. This should be clarified. The only thing I understand is that the there has been some interruption of NMHC data contrary to what the authors claim in the abstract of the paper.

Page 5, L12: Remove the term "worldwide" as Table 1 shows data from the Mediterranean/European area at the most.

Page 5, L15: The authors should clarify why the reader should bear in mind differences in sampling methods and analytical techniques. Are some of the sampling methods and/or analytical techniques and associated results listed suspicious and cannot be compared to each other?

Page 5, L15-18: While I agree that the authors choose Bilbao and Beirut since long-term NMHC measurements were reported for both sites, the authors neglect to describe similarities and differences among those sites in terms of urban size, morphology and climatological conditions.

Page 5, L18-20: The same comment as above applies here. As long as there is no

more elaborated comparison, the presentation of the data remains generic.

Page 5, L21: How much does "significantly" really mean here?

Page 5, L21-22: I am confused about the term "...Cairo rural background area...". It looks like a contradiction to me.

Page 5, L22: It is not clear what the two values of 5.8 and 7.5 ppb refer to. Do they refer to benzene and toluene, or do they refer to benzene (or toluene?) from Athens and Cairo?

Page 5, L25-27: The authors neglect to mention the annual variability of other NMHC sources, e.g. evaporation losses.

Page 5, L27: The authors neglect to mention that "atmospheric dynamics" would not only include PBL variations, but also synoptic meteorology. In many cases this would imply enhanced ventilation during wintertime (e.g. through frontal passages).

Page 5, L29: The authors mention that "...the winter-time decrease of the PBL is in the range of 20%....". Does this value refer to the maximum daily PBL height, an average daily PBL height or to the daytime or nighttime PBL height? This may all make a difference.

Page 5, L30: I think the authors only refer to vertical dilution only, here.

Page 5, L30-31: Those are very generic statements here, as it is well-known that dynamic meteorology plays a major role in the distribution and dilution of atmospheric trace gases and it would be rather surprising that "only one factor" would be important.

Page 6, L3-4: I reiterate my comment made above: Why would the fact that NMHCs nicely follow CO hint towards additional sources e.g. heating? This can also be true in the case of traffic emissions. Also, heating is a very general term. For instance electric heating would not emit CO and NMHCs locally (just to name one kind of heating)?

Page 6, L4-7: From Fig 3 I see that Bff increases similarily to Bwb at night. Why can

the authors make the statement that traffic would not be as important as heating?

Page 6 L9-10: I disagree. Usually, PBL heights are at a minimum during morning hours before sunrise, unless the authors can show other evidences for their statement.

The role of meteorology on NMHC levels

Page 6 L12: Some NMHCs also react with Cl. The latter potentially important for coastal areas. Also, in principle all reactions occur throughout the entire day. It just depends on the availability of reacting compounds.

Page 6 L15-20: This is a pretty generic description. It is well-known that the concentration of primarily emitted gaseous pollutants will decrease due to dilution regardless of their chemical class. However, windspeeds < 3 m/s alone would not indicate the presence of local sources. This would only be true for calm winds. From the plots it seems like these are skewed distributions with maximum concentration values around 2 m/s or so. This would rather indicate some regional flow impacts, which the authors neglected to consider. It seems a more elaborate analysis of windspeeds and their effect on NMHC levels in the Athens area has already been presented in Rappengluck et al (1998). With regard to potential long-range transport it is actually interesting to see that there is some acetylene data still around 5 ppb or so at windspeeds around 9 m/s and higher. In fact, those are very high acetylene values despite strong dilution. What is the reason for this?

Page 6 L21-22: In Fig 5a it is quite surprising to see that the minimum occurrence (north; 1% occurence) is just side-by-side with the maximum occurrence (northeast; I would guess 27% occurence). What is the reason for this quite unusual wind direction occurrence distribution?

Page 6 L24-29: Large part of the discussion here contradicts the authors' statement on the dependence of NMHC on windspeed made earlier. For instance, it looks to me that strongest wind speeds (e.g. NE) would not necessarily be associated with

lowest concentrations, while lowest wind speeds (e.g. SE) would not necessarily be associated with highest concentrations.

Page 6, L29: Those sources cannot be defined as local sources any more, as they are not located in the immediate vicinity of the Thissio site.

Page 6, L30-32: I do not understand this sentence. On the one hand the authors mention increased number of fireplaces, on the other hand the authors mention central heating systems (wouldn't central heating systems decrease the number of individual fireplaces?). Also, the authors state that the higher NMHC values to the N sector is due to the age of the buildings. Isn't another (potentially more) important factor that a higher fraction of Athens' population may be located north of Thissio than towards other directions? Also, wouldn't southerly winds bring in cleaner marine air, making the S-N difference in the NMHC concentration not even more drastic? If the authors want to point out heating sources, wouldn't it make sense to distinguish between day- and nighttime?

Page 7, L4: What biogenic compounds are not triggered by temperature?

Identification of NMHC emission sources

Page 7, L28: Remove "locally", as dispersion acts on all airborne compounds and is not confined to locally emitted pollutants.

Page 8, L8: Mention the NMHC tracers for wood burning exactly.

Page 8, L28-30: Did the authors also apply the baseline subtraction for the tunnel measurements?

Page 8, L30-31: How can the authors justify that their tunnel measurements are not influenced by outside air masses?

Page 8, L31-32: I completely disagree on the authors' statement. The authors neglect to mention what they consider "dominant species", however just looking into NMHCs

such as acetylene, benzene, and toluene, the two profiles "Morning Peak Athens" and "Highway Tunnel - Athens" are completely different: while acetylene for the "Morning Peak Athens" is about 6-7 times higher than for the case "Highway Tunnel - Athens", benzene and toluene values are about 2-3 times lower at the same time.

Page 8, L32 - Page 9, L1: I disagree here again! I do not see that profiles fit nicely. Instead, there are a lot of significant differences. Also, what do the authors consider "common NMHCs"?

Page 8, L1-3: Why should there be higher traffic related butane fraction due to evaporation in ambient air than in the tunnel? Even more surprising, as the tunnel measurements were taken in May, which presumably has warmer temperatures than wintertime. Also, when butanes should be related to evaporation why does propane, another prominent tracer for evaporation, show pretty similar values in the tunnel measurements compared to the "Morning Peak Athens" data?

Page 8, L7: "...during both months...". I disagree with this statement, as only results from two selected days are shown.

Page 8, L8-10: If this justification is true for ethylene, why would it be different for i-pentane? Still, photochemical decay should also be more active for i-pentane in October than in December. However, i-pentane shows a lower slope in December than in October contrary to ethylene. Also, could a difference in solar radiation energy in October vs December in the 7:00-10:00 LT time frame explain an increase in the slope of ethylene by about 60%?

Page 8, L12-14: The definition of the background concentration appears odd. How can the minimum value between 12:00-17:00LT be representative for the nighttime period 18:00-05:00LT? Both are pretty long periods (5 and 11 hours, respectively). From Figs. 4 and 5 we learnt that the NMHC concentration critically depends on wind speed and wind direction. How can the authors make sure that such changes in wind speed and/or wind direction would neither occur during the daytime reference period nor during the

nighttime period?

Page 8, L16-19: Are these differences statistically significant?

Page 10, L9-10: Not sure, how the authors know that BCff at night is due to fossil fuel heating only, and not also impacted by traffic.

Page 10, L13: What were those "different meteorological profiles"?

Page 10, L14-16: This is not supported by the data presented in the paper!

Page 10, L19-20: There are already 4 months of continuous NHMC measurement available. Why is there a longer data set needed to distinguish different source types?

Table 1: Remove the term "worldwide" in the table caption, as Table 1 shows a few selected data from the Mediterranean/European area at the most. What does the second sentence of the table caption refer to? What quantities are compared in this table: means or medians or ....? What do the authors mean by "sampling" frequency: sampling duration or measurement cycle? There is no information given for "sampling frequency" for Baudic et al., Salameh et al., and Durana et al.. Why are the results for the summer 2012 and winter 2013 Athens campaigns reported by Kaltsonoudis et al (2016) not listed in this table? At least, results for isoprene, benzene, and toluene would be comparable.

Figure 1: There is a quite unusual long-term baseline increase of acetylene starting about 1 ppb at the end of November until early January, when it reaches a bit more than 5 ppb (which is pretty high for an urban background site!). Then it abruptly decreases. This feature is not seen in other NMHCs shown in this plot. What is driving (a) this continuous increase and (b) its abrupt decrease?

Figure 2: Why is the mean and not the median shown, as Fig 1 clearly shows that NMHC data is not normally distributed?

Figure 3: Is the data shown based on mean or median hourly averaged values? It

[Figure]

should be median values.

Figure 4: The figure should be better shown with Box-Whisker plots for designated windspeed classes, as the data cloud in Figure 4 could be misleading, as the number of overlapping data points might differ significantly.

Figure 5: What quantity for the NMHC data is shown: mean or median data? Standard deviation bars should be included.

Figure 8: Error bars should be included.

Figure 10, figure caption: I disagree that such a long time period (18:00-05:00 LT; 11 hours!) can be considered a nighttime "peak". Error bars should be included. I do not see that the values shown in the figure add up to 100%.
* * *

---

## Author Comment (AC1) · 14 Mar 2018

Reference article: acp-2017-936

Title: Non Methane Hydrocarbons variability in Athens during winter-time: The role of traffic and heating

**Erratum**

Due to the continuous evaluation of the present data-set after the submission of this manuscript, an error in the calculation of the concentrations of NMHCs was found, resulting in an overestimation of ethane and ethylene (especially after December 2015). Propene and acetylene were also affected but not significantly ($< 3\%$), while the remaining NMHCs were not affected at all. In particular, the correction decreased the levels of ethane and ethylene by 20 to 50% (from November 2015 to February 2016) and 17 to 38% (same period) respectively.

Although the levels of ethane and ethylene changed, the main conclusions of this manuscript were not affected. As a consequence, the figures were corrected and our answers to the reviewers are based on the corrected data. The co-authors would like to apologize for this inconvenience and they express their gratitude for the understanding.

In a summary, the following changes were done in the manuscript from this change in NMHCs levels:

- All the tables figures changed based on the new concentrations of the 4 compounds (ethane, ethylene, propene, acetylene). The figures can be found at the end of the replies section.

- P5, L5: "The highest values have been observed for ethane and ethylene, ranged mostly between 26 and 23 ppb and were encountered in wintertime. For these compounds the lower values were above 0.3 ppb for the whole period."

- P7, L14: The phrase "Ethane, ethylene and acetylene were moderately ($R^2$ of 0.5–0.7) correlated with C4 or C5 compounds…." was changed with the "All NMHCs were well correlated ($R^2 > 0.81$) …".

- P7, L18: In the phrase "The strong correlation ($R^2 > 0.84$) of the hydrocarbons, except ethane, with $BC_{ff}$ could imply stronger emission of NMHCs from fossil fuel combustion processes relatively to wood burning", the term "except ethane" was removed.

- P8, L18: The phrase "For the nSP cases (Fig. 7 and S7) the concentrations of all compounds were very low (lower than the minimum of the SP periods) and almost equal, with the exception of ethane and acetylene that demonstrated higher concentrations in December by a factor of two (Fig. S7a,e)." was changed to the "NMHCs levels during the nSP periods in October and December were equal (Fig. 8 and S8). Furthermore, the concentrations of all compounds during nSP were very low; even lower than the minimum values observed during mid-day during SP periods of the same months.".

**Answers to reviewer**

We would like to thank the reviewer for his/her comments which help us to improve the submitted version. Below is a point by point reply to the comments (the comments are in Italics).

**Reviewer #1:**

*1: "I do feel, however, that there is insufficient evidence presented here to support the authors' claim that the boundary layer height change between October and December is not the main cause of the observed increase in VOC mixing ratios. I would suggest the authors either re-phrase the sections relating to this (to include the possibility of meteorology playing a major role) or provide more evidence in support of this."*

**Reply:** Unfortunately, we don't dispose detailed data of MBL variability not only for our site but for the whole GAA. We have however indications that support our statement on a minor role of MBL in explaining the significant increase of the NMHCs in winter relative to October (reference period).

1: If we examine the nighttime ratio (median value) of winter versus October for all NMHCs a great variability can be seen among the various compounds ranging from 1 or even lower to 2. For instance, the pentanes, compounds characteristic of traffic have a winter/October ratio during nighttime of 1, while other NMHCs emitted from wood burning in addition to traffic (e.g. benzene) have a ratio of 1.4. Same stands also for ethane, ethylene or acetylene compound also emitted by wood burning and depict a winter versus October ratio of 1.7, 1.5

and 2.7 respectively. The above information indicates that source impact is higher compared to MBL.

2: An additional indication corroborating with the above conclusion comes from $SO_4$ diurnal variability measured at the same site and period using an ACSM, thus with similar time resolution with the NMHCs. Sulfate is a regional pollutant thus MBL changes are expected to dictate its diurnal profile rather than local sources. No significant diurnal variability can be seen for sulfate during the winter months (December 2015, January and February 2016), indicating also higher source impact compared to MBL. For information, the $NO_3$, another regional pollutant but with significant local influence, maximizes during night compared to day indicative of impact from local sources especially wood burning (Theodosi C., personal communication).

*Throughout the article: "C2-C6"*
*Should be written as C2 – C6*
**Reply:** We follow his/her suggestion and it has been changed along the article with C6 – C12.

*P 2, L 11: "Athens, the capital of Greece and an important megacity…."*
*Is Athens a megacity? Generally a megacity is considered as one with a population of more than 10 million. Further justification is required or this should be removed.*
**Reply:** Indeed, Athens is not a megacity and the word "megacity" was removed.

*P 2, L 27: Comment "The above demonstrate the increasing need for intensive measurement of NMHCs in Athens, to better understand their sources, temporal characteristics and role on smog formation, in the new conditions established during the economic crisis years, with competing traffic and wood burning." Perhaps more fundamentally there is need to observe the current atmospheric composition to allow for the impact of future changes (fuel composition changes or other control strategies) to be assessed. There is a need to establish a "current baseline" for Athens.*
**Reply:** We thank the reviewer for his/her comment and the proposition was added to the manuscript.

"Consequently, the above demonstrate the increasing need for intensive measurement of NMHCs in Athens, to observe the current atmospheric composition to allow for the impact of

future changes (fuel composition changes or other control strategies) to be assessed. There is a need to establish a "current baseline" for Athens.".

*p. 3, l. 24: "The trap was then heated rapidly to 22 $^o$C…"*
*Is this a typo? Compounds won't desorb well (if at all) at these temperatures. Should this read 220$^o$C?*
**Reply:** We agree with the remark; indeed, it is a typo. The text was corrected accordingly in the same page and line.

*P 3, L 28: "The overall estimated uncertainty of the measurement is 15%."*
*A more detailed discussion of the measurement uncertainty is required to understand how this value is derived. Which parameters are included within the uncertainty? Does the uncertainty vary with compound type? Is it dependent upon mixing ratio or constant across the measurement range? These are all important details which should be included here.*
**Reply:** We thank the reviewer for his/her remark. As all this info is already reported in the literature and the following sentence is included in order to avoid repetitions. "Details about the equipment technique and performances, as well as the estimation of the uncertainty, are provided by Gros et al. (2011)".

*P 4, L 28: "…with the exception of isoprene (approximately 10%)."*
*Why was the data coverage for isoprene worse than others? Were there interferences? This is important to inform other potential users of this equipment for the measurement of VOCs and also needed to confirm that the data quality of the other VOC measurements wasn't impacted by these issues.*
**Reply:** The low coverage of isoprene is mainly due to the very low activity of its normal sources (biogenic) for the studied period and not to instrument malfunction. To avoid misunderstanding the sentence was rephrased as follows: "The latter can be attributed to the low activity of its principal source, that is emissions from vegetation (Fuentes et al., 2000; Guenther et al., 1995). Moreover, the significant night time levels (above 300 ppt in some cases) could be indicative of non-vegetation sources, like traffic or domestic wood burning (Borbon et al., 2001, 2003; Gaeggeler et al., 2008; Kaltsonoudis et al., 2016). Consequently, it is not possible to determine an accurate diurnal variability for this compound."

*P 5, L 1: "... the significant night time levels (above 300 ppt in some cases)."*

*Can this data be trusted given the aforementioned problems with the isoprene data coverage?*
*Further discussion may be needed (unless this is covered in the explanations of the isoprene data coverage above).*

**Reply:** The answer is included in the previous comment.

*P 5, L 5: "... while lower values were below 5 ppb for the whole period."*

*I'm unsure what this statement means, clarification is needed. Does this relate to ethane and ethylene or the other VOCs measured?*

**Reply:** It refers to ethane and ethylene and thus the sentence was corrected accordingly to the text at p.5, l. 5: "The highest values have been observed for ethane and ethylene ranged mostly between 26 and 23 ppb, and were encountered in wintertime. For these compounds the lower values were above 0.3 ppb for the whole period."

*P 5, L 8: "The average concentration of benzene during the studied period was 0.7 ppb (still not a full year), which is considerably below the EU average annual limit of 5 µg m$^{-3}$ or 1.5 ppb (Directive 2008/50/EC of the European Parliament)."*

*The data presented here doesn't include summer time values where we'd expect lower mixing ratios. If these were included then presumably the value would fall well below the threshold. This should be included here, it is of interest in- and of- itself, but also leads the reader to question whether the current directives are suitable and adequate?*

**Reply:** We agree with the reviewer and the sentence was removed.

*P 5, L 12: "The comparison with those already published for the GAA, indicates an apparent decrease by a factor of 2 to 6 for the majority of the species lying above C4 (taking as reference the case of Ancient agora urban area in the close vicinity of the Thissio Station), always bearing in mind differences in sampling period (summer versus winter), location, sampling method and analytical techniques."*

*Is it possible to estimate of the actual decrease? This would be of interest here, despite the various caveats that must be included.*

**Reply:** The following sentence was added. "The comparison with those already published for the GAA, indicates an apparent decrease by a factor of 2 to 6 for the majority of the species lying above $C_4$ (taking as reference the case of Ancient agora urban area in the close vicinity

of the Thissio Station). This decreasing trend is in agreement with a decrease in primary pollutants CO, SO$_2$ already reported by Kalabokas et al. (1999) and Gratsea et al. ( 2017), due to the air quality measures taken by the Greek government. However, this decrease has to be seen with cautious considering differences in sampling period (summer versus winter), location, sampling method and analytical techniques."

*P 5, L 20: "Furthermore, our findings for benzene and toluene, were significantly lower than the 12 hour day-time average levels reported for a Cairo rural background area, as reported by Khoder et al., 2007 (mean levels of 5.8 and 7.5 ppb respectively)."I don't see the significance of this statement? It needs expanding to make its relevance clear.*

**Reply:** The aim of this part is to compare our measurements with those reported in the literature for other Eastern Mediterranean locations. The text was rearranged in order to state the relevance of the comparison as follows: 'Furthermore, our measured benzene and toluene levels (Table 1), were significantly 7 and 3 times lower than the 12-hour day-time average levels reported for a Cairo rural area by Khoder et al. (2007), and equal to 5.8 and 7.5 ppb for benzene and toluene respectively."

*P 5, L 23: "... pattern for all NMHC concentrations was their gradually increase from October:.."*

*Typo gradually to gradual*

**Reply:** Based on the reviewer's comment the text was corrected accordingly.

*P 5, L 29:"... according to Kokkalis (personal communication) the winter-time decrease of PBL is in the range of 20%, ..."*

*This is an important statement in the context of this article and needs more detailed supporting evidence. Later figures and text attempt to reaffirm this statement to support the implication that source-changes define the changes in VOC composition. In its current format I don't see a convincing argument to support this. Either the authors need to include substantial supporting evidence for this or the text should be altered to include the possibility that the variations in VOC mixing ratios could be due to changes in Meteorology.*

**Reply:** The remark was answered in another comment in the beginning of the answers part.

*P 6, L 7: "Although the amplitude of both peaks is almost similar (with the exception of December), the duration of the night peak is at least a factor of 2 larger, indicating the predominant role of heating in air quality during wintertime."*

*This would also be conducive with boundary layer dynamics dictating the night time profile.*

**Reply:** Although from the indications presented above, BL seems not to be the main factor, the sentence was corrected to tone down the impact of heating "……..the duration of the night peak is at least a factor of 2 larger, which could imply the impact of heating in air quality during wintertime."

*P 6, L 25: "… the most frequent, resulting to moderate levels of NMHCs."*

*Typo "to" to "in"*

**Reply:** Based on the reviewer's comment the text was corrected accordingly.

*P 7, L 2: "… 2 to 3 times higher levels of NMHCs were observed on December…"*

*Typo "on" to "in"*

**Reply:** Based on the reviewer's comment the text was corrected accordingly.

*P 7, L 7: "When NMHCs are examined against temperature (not shown here), a clear tendency is not evident, although the highest levels occur at lower temperatures."*

*A plot of this would be useful to be included here.*

**Reply:** Following the reviewer's proposition the graph was added in the supplement.

*P 7, L 21: "More precise picture…"*

*Typo "More" to "A more"*

**Reply:** Based on the reviewer's comment the text was corrected accordingly.

*P 8, L 15: "The most striking difference is related to the night and early morning peak, while during mid-day the difference is Minimum."*

*This requires more detail. While the concentration rise is smaller at mid-day, the relative rise looks to be more or less the same for the early morning and mid-day periods (approximately double). Including the percentage increase would clarify this.*

**Reply:** Based on the reviewer's comment, the text was corrected as follows: "The most striking difference is related to the extensive night peak, while during mid-day the difference

is minimal. The night peak of the compounds in December (SP period) is 2 to 6 times higher than October's (SP period) with the highest values corresponding to ethane, ethylene, propene and acetylene. On the other hand the December to October ratio during mid-day is ranged between 2.6 (for propene and acetylene) to 0.9 (for benzene). "

*P 8, L 16: "… while during mid-day the difference is minimum…"*
*Typo "minimum" to "minimal"*
**Reply.** Based on the reviewer's comment the text was corrected accordingly.

*P 8, L 18: "For the nSP cases (Fig. 7 and S7) the concentrations of all compounds were very low (lower than the minimum of the SP periods) and almost equal, with the exception of ethane and acetylene that demonstrated higher concentrations in December by a factor of two (Fig. S7a, e)."*
*I can't make sense of this sentence, it needs re-writing/re-phrasing for clarity*
**Reply:** Following the reviewer's comment, the text was clarified as follows: **"**NMHCs levels during the nSP periods in October and December were equal (Fig. 8 and S8). Furthermore, the concentrations of all compounds during nSP were very low; even lower than the minimum values observed during mid-day during SP periods of the same months.**"**

*P 8, L 32: "It is interesting to note that the profiles, especially those derived from the morning peak, nicely fit with that reported for traffic by Baudic et al. (2016) in Paris (when only the common NMHCs measured in this work have been used)."*
*I don't see a "nice fit" they look different in magnitude and relative composition. Please provide more detail of what the authors are implying by this.*
**Reply:** The entire "Morning Peak" paragraph of the Sect. 3.4.3 was re-written and figure 8 will be replaced by the following one:

[Figure]

**Figure 8**. **% Mass contribution of the measured NMHCs during the morning peak (07:00 – 10:00LT, median values in Thissio, in Patission Monitoring Station, in a highway tunnel in GAA and a highway tunnel close to Paris.**

The previous profiles had common dominant species (i-pentane and toluene) but indeed there were other discrepancies. For that reason we included measurements of an intensive campaign that was performed during the Athens campaign, at Patission Monitoring station of the Hellenic Ministry of Environment and Energy, a site located in a street canyon and significantly impacted by traffic conditions, thus better representing the traffic profile of the GAA. The two profiles of Patission and Thissio are fitting nicely, but differences are apparent if they are compared to tunnel profiles. The conclusion is that the observed discrepancies are attributed to the type of fuel of the vehicles.

In the revised manuscript, the corresponding paragraph will be re-written as follows:

"As discussed in Sect. 3.2, the morning peak (07:00 – 10:00 LT) of NMHCs could be attributed mainly to traffic. Figure 8 presents the profile of this peak (% mass contribution of the measured NMHCs), during January and February SP days when toluene data were available. Additionally, in the same figure the morning profile obtained from a 2 – days campaign conducted in Patission Monitoring Station (a street canyon located at the center of Athens) and the profiles of two tunnel measurements in G.A.A and Paris are reported. Details on the calculations for the morning profile for the two sites are provided in Sect. S2. Patission profile reflects all types of traffic-related emissions due to the combination of the high

number of vehicles and buses that cross this street, frequent traffic jam conditions, the variety of types of fuels, vehicles age and their maintenance etc.

The two morning profiles, although performed at sites with different impact of traffic, agrees quite well ($R^2 > 0.97$). Iso - pentane and toluene are the two main compounds contributing to the morning profiles accounting by about 44% of the total measured NMHCs at both locations, followed by n- and i-butane and ethylene accounting for almost 30%. Differences among the two morning profiles between these 5 main species are minimum (less than a factor of 1.5). Note also that the morning profile at Thissio is the mean of a whole month period compared to a campaign of two days in Patission which could explain the differences between the two profiles. The profiles obtained at the two tunnels although differ in terms of tunnel length, city, and period have a lot of common features. Again i-pentane and toluene are the two main compounds of the profile accounting by about 56% of the total measured NMHCs at both sites, followed by n -butane, ethylene and benzene accounting for almost 20% in total again at both sites. The most striking difference between the two sites concerns n-pentane (almost a factor of two higher in Paris compared to Athens). Despite the differences between the two tunnel studies the similarity is almost 80% ($R^2 > 0.91$). The biggest difference between the two Athens morning peaks and tunnels concerns acetylene (factor of 4), benzene and toluene (factor of 2). The similarity of Thissio and Patission morning profiles and their difference from the Athens and Paris tunnel profiles, indicate the importance of the type of fuel used. The latter is also concluded in recent works (Ait-Helal et al., 2015; Q. Zhang et al., 2018; Y. Zhang et al., 2018), where important differences are reported between tunnel measurements worldwide, and attributed to the variance of the car-fleet (type of vehicle and fuel). In our case there is a possibility that the car-fleet in the tunnel is not representative for the GAA, since the existing tolls reduce the use of the tunnel due to financial issues. Also, measurements are performed during noon when the traffic density is quite low. In any case, the prevalence of i-pentane and toluene in all profiles, indicate the continuing dominance of gasoline powered cars. Moreover, higher values of ethane, propane and butanes that are depicted in the morning peaks of the urban sites relatively to the tunnel measurements, reflect the increased number of LPG powered vehicles in Athens and natural gas-powered buses (Fameli and Assimakopoulos, 2016). In fact, the connection of high levels of $C_2$-$C_4$ alkanes and the number of LPG-powered cars is highlighted in other tunnel works as well (Ait-Helal et al., 2015; Q. Zhang et al., 2018)."

*P 9, L 2: "… butanes are however be noted in the morning peak…"*

*Typo "be" needs to be removed*

**Reply:** The corresponding paragraph was modified (answer of the previous comment).

*P 9, L 4: "Toluene, an important contributor to the traffic profile (Fig. 8), was measured only for one month during winter."*

*Does this affect the plot? Is that why the toluene is lower than at other sites? This sentence needs expanding upon to clarify its effect, if any, upon the figure.*

**Reply:** The short-term measurement of toluene has no effect on fig. 8, because the morning profile derived from days that toluene data were available, so all the compounds have the same number of data.

*P 9, L 12: "…(the contribution of the later was more evident during winter)."*

*Typo "latter" not "later"*

**Reply:** The corresponding section was changed (previous comment).

*Fig 7 caption: I suggest including the definition of SP and nSP within the figure caption*

**Reply:** We agree with the reviewer and definition of SP and nSP was added in the figure caption.

**References**

[revised manuscript text omitted]

---

## Author Comment (AC2) · 14 Mar 2018

Reference article: acp-2017-936

Title: Non Methane Hydrocarbons variability in Athens during winter-time: The role of traffic and heating

**Erratum**

Due to the continuous evaluation of the present data-set after the submission of this manuscript, an error in the calculation of the concentrations of NMHCs was found, resulting in an overestimation of ethane and ethylene (especially after December 2015). Propene and acetylene were also affected but not significantly (< 3%), while the remaining NMHCs were not affected at all. In particular, the correction decreased the levels of ethane and ethylene by 20 to 50% (from November 2015 to February 2016) and 17 to 38% (same period) respectively.

Although the levels of ethane and ethylene changed, the main conclusions of this manuscript were not affected. As a consequence, the figures were corrected and our answers to the reviewers are based on the corrected data. The co-authors would like to apologize for this inconvenience and they express their gratitude for the understanding.

In a summary, the following changes were done in the manuscript from this change in NMHCs levels:

- All the tables figures changed based on the new concentrations of the 4 compounds (ethane, ethylene, propene, acetylene). The figures can be found at the end of the replies section.

- P5, L5: "The highest values have been observed for ethane and ethylene, ranged mostly between 26 and 23 ppb, and were encountered in wintertime. For these compounds the lower values were above 0.3 ppb for the whole period."

- P7, L14: The phrase "Ethane, ethylene and acetylene were moderately ($R^2$ of 0.5–0.7) correlated with C4 or C5 compounds…." was changed with the "All NMHCs were well correlated ($R² > 0.81$) …".

- P7, L18: In the phrase "The strong correlation ($R² > 0.84$) of the hydrocarbons, except ethane, with $BC_{ff}$ could imply stronger emission of NMHCs from fossil fuel combustion processes relatively to wood burning", the term "except ethane" was removed.

- P8, L18: The phrase "For the nSP cases (Fig. 7 and S7) the concentrations of all compounds were very low (lower than the minimum of the SP periods) and almost

equal, with the exception of ethane and acetylene that demonstrated higher concentrations in December by a factor of two (Fig. S7a,e)." was changed to the "NMHCs levels during the nSP periods in October and December were equal (Fig. 8 and S8). Furthermore, the concentrations of all compounds during nSP were very low; even lower than the minimum values observed during mid-day during SP periods of the same months.".

**Answers to reviewer**

We would like to thank the reviewer for his/her comments which help us to improve the submitted version. Below is a point by point reply to the comments (the comments are in Italics).

**Reviewer #2:**

*1: "On the background of existing literature I am not sure what the real novelty of this paper in terms of methods and results is. While I agree that C2 and C3 measurements have not yet been done before in Athens, it seems the inclusion of these does not yield more findings than already reported by Kaltsonoudis et al (2016). On the other hand, the Panopoulou et al paper makes same sketchy description of meteorological impacts on NMHCs, but lacks some elaborate analysis similar to those presented in Rappengluck et al (1998) for transport effects and also Kourtidis et al (1999) for temperature effects for Athens. As the authors make an important point on page 2, L27-29, that there have been new conditions during the economic crisis years (i.e. competing traffic vs wood burning emissions) it would be actually meaningful to perform a comparative analysis between the data sets reported 20 years ago and the ones reported by Panopoulou et al. It seems both studies include continuous NMHC measurements and PMF source apportionment analysis would be feasible and would provide interesting insights."*

**Reply:** We agree with the reviewer that this work reports for the first time C2 and C3 data in Athens. Note that Kaltsonoudis et al., paper although report measurements during wintertime in GAA did not report any NMHC data below C5 and the only common compounds with this work is isoprene, benzene and toluene. **Consequently, this work reports for the first time to our knowledge C2-C5 NMHCs measurements in Athens during winter time and for the first time ever C2-C3 data in GAA.**

We agree also that since the last pioneering work by Rappenglück et al., performed in Athens 20 years ago a lot happened in Athens in terms of source evolution (see for instance Kalabokas et al., 1999; Gratsea et al., 2017). The above demonstrate, as reviewer 1 highlighted, the increasing need for intensive measurement of NMHCs in Athens, to observe the current atmospheric composition to allow for the impact of future changes (fuel composition changes or other control strategies) to be assessed. However direct comparison with the work performed 20 years ago is difficult considering differences in sampling period (summer versus winter and thus different photochemistry), location, sampling method and analytical techniques. For that reason, the introduction was changed to include all the above remarks and points raised by the reviewers:

"Non-methane hydrocarbons (NMHCs) are key atmospheric constituents for atmospheric chemistry. In the presence of $NO_x$, their oxidation leads to formation of tropospheric ozone and other species, such as peroxy radicals ($RO_2$) and peroxy acetyl nitrate (PAN), thus affecting the oxidative capacity of the atmosphere (Atkinson, 2000 and references therein). NMHCs' oxidation contributes to the formation of secondary organic aerosols (SOA), which in turn affects light scattering, visibility and CCN formation (Tsigaridis and Kanakidou, 2003; Seinfeld and Pandis, 2016 and references therein). In urban areas they mainly originate from anthropogenic sources such as traffic, solvents' use, residential heating, natural gas use, industrial activity, but also emit from natural sources such as vegetation (Guenther et al., 1995; Barletta et al., 2005; Kansal, 2009; Sauvage et al., 2009; Salameh et al., 2015; Baudic et al., 2016; Jaimes-Palomera et al., 2016). Besides their key role as secondary pollutants precursors, NMHCs are of interest regarding their association with health issues (EEA report, No 28/2016, 2016). In particular and since 2013, atmospheric substances have been classified by the International Agency for Research on Cancer (WHO-IARC, 2013) in four major groups regarding their carcinogenicity to humans, with benzene and 1,3-butadiene among those NMHCs classified as potential carcinogens (IARC, 2012).

Athens, the capital of Greece, pollution-wise, with almost five million of inhabitants, is frequently subjected to intense air-pollution episodes, leading to exceedance of the EU air quality limits. The driving processes and atmospheric dynamics of these episodes have been scrutinized during the last decades (Cvitas et al., 1985; Lalas et al., 1982, 1983, 1987; Mantis et al., 1992; Nester, 1995; Melas et al., 1998; Ziomas et al., 1998; Kanakidou et al., 2011). However, the measurements of pollution precursors are mostly about ozone and nitrogen oxides. The few existing and non-continuous NMHC measurements in Athens by means of

canisters or sorbent tubes, performed for short period (days) during summertime or autumn (Moschonas and Glavas, 1996; Klemm et al., 1998; Moschonas et al., 2001; Giakoumi et al., 2009). Continuous measurements of NMHCs in Athens for a period of one month have been conducted 20 years ago at three locations, two suburban and one urban, containing almost 50 $C_4 - C_{12}$ compounds (Rappenglück et al., 1998, 1999), and recently by Kaltsonoudis et al. (2016), for 1 month in winter 2013 at an urban location (Thissio) and one in summer 2012 at a suburban one (A. Paraskevi), containing 11 aromatic and oxygenated organic gaseous compounds (5 NMHCs). Meanwhile, significant changes in pollutant sources occurred in Athens the last 20 years, which inflicted important decreases in the annual concentrations of major pollutants such as CO, $SO_2$, $NO_x$ (combustion marker) (Gratsea et al., 2017; Kalabokas et al., 1999). Because the latter trend is attributed to the car fleet renewal, fuel improvement, metro line extension and industrial emission controls, a decrease to the NMHCs levels originating from traffic and industrial emissions is also expected. However, after 2012, a new winter-time source of pollution emerged in Greece, due to uncontrolled wood burning for domestic heating (Saffari et al., 2013; Paraskevopoulou et al., 2015; Kaltsonoudis et al., 2016; Fourtziou et al., 2017; Gratsea et al., 2017). This is an important source of various pollutants such as particulate matter (PM), polycyclic aromatic hydrocarbons (PAHs), black carbon (BC) and CO (Gratsea et al., 2017; Hellén et al., 2008; Paraskevopoulou et al., 2015; Schauer et al., 2001 et references therein), while it can represent up to 50% of the mass of Volatile Organic Compounds (VOCs) during winter (case of Paris; Baudic et al., 2016). In general, there are some studies in a global scale for the characterization of the VOC emissions from domestic wood burning, however, differences are observed to the emission rates or the emission profiles of the VOCs, that are attributed to type of wood, stove, lightening material and the variety of emissions from the burning stages (Barrefors and Petersson, 1995; Baudic et al., 2016; Evtyugina et al., 2014; Gaeggeler et al., 2008; Gustafson et al., 2007; Hellén et al., 2008; Liu et al., 2008; Schauer et al., 2001 and references therein). In contrast, the studies including light NMHCs are very few (Barrefors and Petersson, 1995; Baudic et al., 2016; Liu et al., 2008; Schauer et al., 2001) and present discrepancies. For example, the higher contribution of benzene relatively to acetylene to the residential wood burning profile reported by Baudic et al. (2016) that is not depicted in the profile of Liu et al. (2008).

Nevertheless, the latest work on VOCs in Athens of Kaltsonoudis et al. (2016) gave a first insight about the positive effects of the Greek air pollution on aromatics and oxygenated VOC levels, pointing out the important contribution of wood burning to the wintertime night

concentrations. Consequently, the above demonstrate the increasing need for intensive measurement of NMHCs in Athens, to observe the current atmospheric composition to allow for the impact of future changes (fuel composition changes or other control strategies) to be assessed. There is a need to establish a "current baseline" for Athens. In addition, it would be interesting to investigate the contribution of traffic and wood burning to the light NMHCs, which are two competitive sources with similar NMHC tracers that could lead to an overestimation of the first due to the contribution of the second (Schauer et al., 2001).

The current study presents, time-resolved, uninterrupted data of 11 NMHCs with two to six carbon atoms, during a time span of several months (October 2015 to mid-February 2016) in the Great Athens Area (GAA). The emphasis of this work is on: (1) the determination of the ambient levels of C2-C6 NMHCs during autumn and winter, twenty years after their first summer-time measurements (especially for C2-C3, these are the first ever continuous measurements of NMHCs in Athens); (2) the study of their temporal characteristics and the determination of the factors controlling their variability, and (3) the investigation of traffic and residential heating impact on NMHCs levels."

*P 1, L18:"...among the highest in literature for the Mediterranean area..." It should be clarified if this refers to the same season or not. It is known that many primarily emitted pollutants reach higher values in wintertime.*

**Reply:** Based on the reviewer comments, the text was clarified as follows: "The measured NMHC levels are among the highest reported in literature for the Mediterranean area during winter months and the majority of the compounds demonstrate a remarkable day to day variability."

*P 1, L20-21: What do the authors exactly mean by local meteorology, as this term is quite unusual? Its connotation would mean that it is not representative for a larger fetch. I disagree that "local" meteorology would control the variability of NMHC levels alone. What about the temporal variability of NMHC emissions?*

**Reply:** By local meteorology the authors mean the microscale meteorology. For that reason, the term "local meteorology" is replaced by "microscale meteorological conditions" in the manuscript. It is worthwhile noting that previous work performed in Athens on aerosols showed that during summer the majority of pollutants originate from regional sources outside Greece (more than 60%), whereas during winter local sources prevails (more than 80%) (

Theodosi et al., 2011; Paraskevopoulou et al., 2015). Due to the financial crisis, industrial activities have been considerably decreased in Athens with most industries either closing or moved out Greece (Vrekoussis et al., 2013). Thus, traffic and heating can be considered as the main sources of pollutants in GAA during winter. Temporal variability of these sources is indeed an important factor controlling NMHCs levels and our analytical resolution of 1h captures quite well this temporal variability.

*P 1 L24-25: Why would the fact that NMHCs nicely follow CO hint towards additional sources e.g. heating? This can also be true in the case of traffic emissions. Also, heating is a very general term. For instance, electric heating would not emit CO and NMHCs locally (just to name one kind of heating)?*

**Reply:** Based on the suggestion of the reviewer, the text was clarified as follows: "The amplitude (intensity) of both peaks is gradually increasing towards winter, respectively to autumn, by a factor of 3 to 6 and nicely follow that of carbon monoxide (CO), indicating contribution from sources other than traffic, related to traffic related to combustion e.g. domestic heating (fuel or wood burning).".

*P 1 L27-29: Why does the present data not allow for the quantification of the relative contribution of fossil fuel and wood burning for heating purposes?*

**Reply:** This sentence at the abstract is based on the results presented in detail in the manuscript, where the reader can find all the necessary details. In any case latter the sentence in the abstract was rephrased as follows: "Following the same comparison for the night peak, the tracers and source profiles clearly indicate the presence of traffic and domestic combustion of fossil fuel and wood burning for heating purposes. However, the present data-set does not allow for quantification of each source due to the similarity of emissions, thus measurements of more specific compounds are needed for the better understanding of the contribution of these three nocturnal VOC sources."

*Page 2 L3-4: This is only true for urban areas!*

**Reply:** Since the manuscript refers to urban measurements, the text is corrected as follows: "In urban areas they mainly originate from anthropogenic sources such as traffic, solvents' use, residential heating, natural gas use, industrial activity, but also emit from natural sources

such as vegetation (Guenther et al., 1995; Barletta et al., 2005; Kansal, 2009; Sauvage et al., 2009; Salameh et al., 2015; Baudic et al., 2016; Jaimes-Palomera et al., 2016)."

*P 2, L6: replace "Baudic et al., 2016b" by "Baudic et al., 2016".*
**Reply:** The typo was corrected accordingly.

*P 2, L7: This EEA report is not properly cited in the list of references.*
**Reply:** The citation was corrected accordingly.

*P 2, L14: The "Cvitas et al paper" should be properly cited as its primary appearance is in a journal.*
**Reply:** The citation was corrected accordingly.

*P 2, L14: As far as I understand it, the Kourtidis et al paper analyzed online NMHC data contrary to what the authors state.*
**Reply:** The citation did not correspond to the statement of the phrase and it was removed.

*P 2, L20-21: The term "limited" is not properly chosen, as it seems the range of the NMHC data and the number of NMHCs reported by Panopoulou et al. is also limited. It would be fair to mention the entire C range of those earlier measurements which Panopoulou et al cite in order to properly put them into perspective. Also, it looks to me that those measurements contained significantly more speciated NMHCs, and according to table 1 these earlier NMHC measurements include NMHCs with four C atoms contrary to what the authors state here ("...containing more than four atoms of carbon...").*
**Reply:** We agree with the not properly use of the term "limited" and for that reason it was changed throughout the manuscript and also to the corresponding line: "Continuous measurements of NMHCs in Athens for a period of one month have been conducted 20 years ago at three locations, two suburban and one urban, containing almost 50 $C_4 - C_{12}$ compounds (Rappenglück et al., 1998, 1999), and recently by Kaltsonoudis et al. (2016), for 1 month in winter 2013 at an urban background location (Thissio) and one in summer 2012 at a suburban one (A. Paraskevi), containing 11 aromatic and oxygenated organic gaseous compounds in total (5 NMHCs).".

*P 2, L24: The Saffari et al paper is on the Thessaloniki case, not on the Athens case.*

**Reply:** We clarified the phrase as follows: "However, after 2012, a new winter-time source of pollution emerged in Greece, due to uncontrolled wood burning for domestic heating (Saffari et al., 2013; Paraskevopoulou et al., 2015; Kaltsonoudis et al., 2016; Fourtziou et al., 2017; Gratsea et al., 2017)".

*P 2, L26: Again, as mentioned above the term "limited" is not properly chosen. The Kaltsonoudis et al paper actually reports VOCs, which are not measured by Panopoulou et al. Also, while it is true that the Athens winter campaign reported by Kaltsonoudis et al was shorter than the one reported by Panopoulou et al, it seems that Kaltsonoudis et al also report a summer campaign for Athens, which the Kaltsonoudis et al paper does not.*

**Reply:** This remark was already addressed before (answer of the comment for P2 L20-21).

*P 2, L30: It would be fair to mention how many NMHCs were actually measured as it seems that the paper does not report some important NMHCs such as 1,3-butadiene and others, for instance.*

**Reply:** In the NMHCs campaign from 16 October 2015 to 16 February 2016 15 NMHCs were measured, however we chose for this paper the 11 that are the most representative of the sources that the manuscript is dealing with. In the intensive campaign of January and February 2016, 15 additional NMHCs were measured, that will be evaluated for their publication in the future.

*P 2, L34 - P 3, L1: The authors should mention why the analysis is restricted towards traffic and heating impact on NMHC levels.*

**Reply:** Based on the comment the text was corrected accordingly "In addition, it is interesting to investigate the contribution of traffic and wood burning to the light NMHCs, which are two competitive sources with similar NMHC tracers that could lead to an overestimation of the first due to the contribution of the second (Schauer et al., 2001)."

As mentioned before traffic and heating are the two most important local sources of pollution in GAA during wintertime. There is no main industrial zone around Athens anymore.

*P 3, L3: Mention those selected tracers explicitly and mention what kind of sources those tracers are tracing.*

**Reply:** The corresponding phrase was removed.

*In general: I suggest to include a map here. Not everyone is familiar with Athens.*

**Reply:** The following map will be included in the revised manuscript:

Figure XX: Map of the Greater Athens Area. The four mountains define the borders of the area.

[Figure]

*P 3, L7: Actually, more important than the altitude above sea level would be the altitude of the hill site above the surrounding area.*

**Reply:** Based on the comment, the text was corrected accordingly "Measurements were conducted from 16 October 2015 to 15 February 2016, at the urban background station of the National Observatory of Athens (NOA, 37.97° N, 23.72° E, 105 m a.s.l and about 50m above

city level) at Thissio, considered as receptor of pollution plumes of different origins (Paraskevopoulou et al, 2015)." Note that GAA is not exactly a flat area thus for the information requested by the reviewer is not easy to give an accurate answer as a clear reference site is missing.

*P 3 L9: How far away from the site are the Filopappou and Acropolis hills and how high are those? Again, a map would be helpful. It is not sufficient to refer to other publications here, as it seems the site the authors are talking about is a very specific one.*

**Reply:** Based on the comment, the text was corrected accordingly "The station is located in the historical center of Athens, on top of a hill (Lofos Nimfon), surrounded by a pedestrian zone, a residential area and by the Filopappou (108m a.s.l) and Acropolis Hills (150m a.s.l), which are located 500m and 800m away respectively". The site is indeed very specific as it holds the oldest meteorological station of Greece which provides continuous data since 1858.

*P 3, L7-8: What do the authors mean by " ...was set to sample on a 10 min basis..."? Do they refer to the sampling time or sampling frequency? Is the GC continuously flushed with ambient air?*

**Reply:** We refer to sampling time. The GC is continuously flushed with ambient air. Based on the comment, the text was corrected accordingly "The C2-C6 NMHC analyzer was set to sample ambient air on a 10 min basis followed by an analysis time of 20 min, while for the C6-C12 the respective timing was 20 min and 10 min.

*P 3, L20: The sampling line has a pretty small diameter, which is very unusual. The authors should clarify why they chose that small diameter. Also, why did the authors use a stainless steel sampling line and not a glass line, which will have the least interaction with the sample, in particular when considering this very long line (6 m)? Did the authors check the sampling line for any potential losses of NMHCs, e.g. through looking into any deviations of the C-response? Did the authors make calibrations directly to the GC or through the sampling line? Did the authors use any filter at the sampling inlet?*

**Reply:** The diameter of the sampling line is 0.315cm and not mm as was written. The typo was corrected accordingly.

We used a stainless-steel line for sampling following the recommendation by ACTRIS guidelines of 2014 for trace gases networking: Volatile organic compounds and nitrogen oxides (ACTRIS, Deliverable WP4 / D4.9 (42), 2014/09/30).

In general, the calibrations were made directly on the sampling port of the GC and the sampling line was checked for losses, by performing a number of calibrations the same day through the sampling line and also directly on the GC-FID, resulting in similar results.

There was a stainless-steel filter (screens) at the sampling inlet with a pore size of 4μm.

Part of these information will be included in the revised manuscript as follows: "For the airmoVOC C2 – C6 analyzer, 189 mL of air was drawn through a 0.315 cm diameter, 6 m-long stainless-steel line with a filter of 4μm pore size at the sampling inlet, and a flow rate of 18.9 mL min$^{-1}$.

*P 3, L28-29: Is this uncertainty true for all VOCs? Usually, it would be class specific. What are the detection limits?*

**Reply:** Based on the reviewer's #1 comment, we have clarified the sentence as follows: "Details about the equipment technique and performances, as well as the estimation of the uncertainty, are provided by Gros et al. (2011). The detection limit is in the range of 0.02ppb (propene, n-pentane) to 0.05ppb (propane), while for ethane and ethylene is 0.1ppb."

*P 3, L30: Same as mentioned above with regard to the sampling line.*

**Reply:** The remark was already addressed before for the comment of P3 L20 and the text will be corrected accordingly.

*P 4, L5: Why is only toluene used? Why not at least ethylbenzene and the xylenes in addition? Would the exclusion of these NMHCs not introduce a bias into the data analysis, as important tracers for solvent emissions are excluded? What are the uncertainties and the detection limits for this GC?*

**Reply:** The main part of the paper is focused on C2 – C6 NMHCs, from which we chose only the compounds dedicated to the scope of this paper (the role of traffic and heating on the winter levels of NMHCs, the two most important sources of pollution during wintertime in the area). Toluene was available only for one month at the end of the campaign, thus it was selected as an additional tool for our analysis. Moreover, it was only used for the interpretation of the morning profile of Thissio, for which we have subtracted a background

concentration (described in the section S2 of the supplement) in order to compare with the traffic profiles available in the literature.

The uncertainty of the instrument is less than 20% and the detection limit of toluene is 0.026 ppb. These elements were introduced also in the manuscript.

*Auxiliary measurements*

*In general: The authors also report NO measurements at some point in the paper, but neglect to mention the instrumental description here.*

**Reply:** Based on the comment, the Sect. 2.3 was modified as follows: "Real time monitoring of carbon monoxide (CO), black carbon (BC) and nitrogen oxides ($NO_x = NO$ and $NO_2$) was also conducted during the reported period. For CO and $NO_x$, Horiba 360 Series Gas Analyzers of one-minute resolution were used and calibrated with certified standards. A seven wavelength Magee Scientific AE33 aethalometer (five minutes resolution) was operated for the measurement of BC; and its fractions associated with fossil fuel and wood burning ($BC_{ff}$ and $BC_{wb}$, respectively) were calculated based on the biomass burning contribution derived automatically by the instrument software. Meteorological data were provided by NOA's meteorological station at Thissio premises."

*Tunnel measurements*

*P 4, L16: The authors should mention the length of the tunnel, whether lanes were for both directions (there could also be dedicated tunnels for one direction only), if there was any artificial ventilation and if there might have been any limitations on traffic through this tunnel (in some cases heavy duty traffic is not allowed). In any case an estimate of the traffic fleet composition (e.g. heavy duty vs light duty vehicles) would be helpful. All these factors have an impact on the NMHCs levels. At what location of the tunnel did the authors make the measurements exactly? I see the measurements were taken on 12 May 2016, which is different from wintertime. Wouldn't the temperature be different from wintertime and wouldn't this have an enhanced impact on NMHC emissions through evaporation, for instance?*

**Reply:** The tunnel where the measurements are performed is at the periphery of GAA. Its length is 200m and measurements are performed at the middle of the tunnel. The tunnel is at a

highway with toll and mainly used by private cars to avoid Athens center and consequently traffic jams. It has 3 lanes at each direction with no specific restrictions for heavy duty vehicles. Measurements at the tunnel were indeed performed a different period with the measurements and the idea was to obtain a NMHCs profile characteristic for traffic. Given the different period of the measurements and the fact that the toll could prevent buses and trucks to use it frequently, a campaign performed during wintertime at a street canyon in the center of Athens with heavy traffic was preferred to obtain a traffic profile (see new figure 8). Part of this information will be added at the revised manuscript (also in the answer for the comment of P4 L20 – 21).

*P 4, L19: What type of canisters were used?*

**Reply:** The canisters were Entech's Silonite (Silonite™ VS Summa). They have an internal Silonite™ coating that provides a high-quality, long-term sample storage solution. These canisters are certified to meet the technical specifications required for EPA methods TO-14a and TO-15. Part of these information will be added at the revised manuscript (also in the answer for the comment of P4 L20 – 21).

*P 4, L20-21: Why was it needed to dilute the sample by a factor of two? Why was a Teflon transfer line used? What kind of Teflon in particular? Did the authors check any NMHC artifacts in the canisters and the sampling path (i.e. Teflon line)?*

**Reply:** The canisters were evacuated prior sampling and the sampling procedure was held up to atmospheric pressure. To push the sample from the canister to the GC line, pressure higher than the atmospheric is needed and this is achieved by over-pressurizing the canister by an addition of a known volume of zero air into the container.
The Teflon line (PTFE) was used for the connection of the canister with the GC. The artifacts in the canisters and the sampling line were checked and no artifacts were found.
In order to better explain the sampling and analysis methods with canisters, the Sect. 2.4 was updated as follows:

"2.4 Street canyon and tunnel measurements

In order to identify the NMHCs fingerprint of traffic emissions, NMHCs measurements were conducted at a monitoring station belonging to the Hellenic Ministry of Environment and Energy, located at a street canyon downtown Athens with increased traffic and frequent traffic jams (Patission street) on 22 to 24 February 2017 (37.99°N, 23.73°E) as well as in a tunnel at the peripheral highway of Athens, (Attiki Odos), on 12 May 2016 from 12:00 LT to 12:45 LT (LT = UTC+2). The tunnel's length is 200m and it has 3 lanes at each direction with no specific restrictions for heavy duty vehicles. The measurements are performed at the middle of tunnel to avoid as possible the influence of ambient air from outside. Concerning the Patission campaign, samples were collected during the morning rush hour, every 1-hour from 06:55 LT to 10:15 LT (LT = UTC+2).

In both cases for the sampling were used 6L stainless steel – silonite canisters - and the sampling time ranged between 2 and 10 minutes. The analysis method is described elsewhere (Sauvage et al., 2009). In summary, before the analysis, the samples were diluted by a factor of two with zero-air, afterwards each canister was connected to the GC-FID system using a Teflon (PTFE) sampling line and finally analyzed by the same GC-FID method as described in Sect. 2.2. Before the sampling, the canisters were cleaned by filling them up with zero air and re-evacuated, at least three times. The content of the cylinders was then analyzed by the GC-FID system to verify the efficiency of the cleaning procedure. The canisters were evacuated a few days prior to the analysis and they were analyzed maximum 1 day after the sampling."

*Temporal variability of NMHCs*

*P 4, L28: I do not understand the concept of data coverage here, as it is not explained. It could refer to the percentage of data above the detection limit vs maximum available data, but this does not make complete sense, as I doubt there were any data of ethane below the detection limit, for instance. However, it cannot be true either that it refers to the data availability vs maximum potential data availability during the time period reflecting instrumental potential instrumental malfunctions and/or failure. This should be clarified. The only thing I understand is that there has been some interruption of NMHC data contrary to what the authors claim in the abstract of the paper.*

**Reply:** Based on the comment, the phrase was corrected as follows: "During the reported period, the data availability (in comparison with the maximum potential data availability) for

all C2-C6 NMHCs was higher than 87% with the exception of isoprene (approximately 10%)".

*P 5, L12: Remove the term "worldwide" as Table 1 shows data from the Mediterranean/European area at the most.*
**Reply:** Based on the comment, the term worldwide was removed.

*P 5, L15: The authors should clarify why the reader should bear in mind differences in sampling methods and analytical techniques. Are some of the sampling methods and/or analytical techniques and associated results listed suspicious and cannot be compared to each other?*
**Reply:** The sentence was re-phrased as follows: "The comparison with those already published for the GAA, indicates an apparent decrease by a factor of 2 to 6 for the majority of the species lying above $C_4$ (taking as reference the case of Ancient agora urban area in the close vicinity of the Thissio Station). This decreasing trend is in agreement with a decrease in primary pollutants CO, $SO_2$ already reported by Kalabokas et al. (1999) and Gratsea et al. (2017), due to the air quality measures taken by the Greek government. However, this decrease has to be seen with cautious considering differences in sampling period (summer versus winter), location, sampling method and analytical techniques."

*P 5, L15-18: While I agree that the authors choose Bilbao and Beirut since long-term NMHC measurements were reported for both sites, the authors neglect to describe similarities and differences among those sites in terms of urban size, morphology and climatological conditions.*
**Reply:** Based on the comment, the text was rephrased, and a new sentence was added: "Comparison with other Mediterranean or south European locations with long-term winter observations is possible only with Beirut (Salameh et al., 2015) and Bilbao (Durana et al., 2006). Beirut, located in the Eastern Mediterranean basin (approximately 200Km SE of Greece, 230m above sea level), has a population of 2000000 citizens and a typical Mediterranean climate with mild winter and hot summer (Salameh et al., 2015). On the contrary Bilbao is an urban and industrial city of 400000 citizens in north Spain, located along a river delta in SE–NW direction, where two mountain ranges run parallel to the river (Ibarra-

Berastegi et al., 2008). Due to their location, both cities experience intense sea breeze cycles. The levels of NMHCs observed in Athens are higher, almost by a factor of two, with the exception of propane, butanes and toluene for Beirut and n-butane, benzene and toluene for Bilbao, which are quite similar to Athens. 'Furthermore, our measured benzene and toluene levels (Table 1), were significantly 7 and 3 times lower than the 12-hour day-time average levels reported for a Cairo rural area by Khoder et al. (2007), and equal to 5.8 and 7.5 ppb for benzene and toluene respectively."

*P 5, L18-20: The same comment as above applies here. As long as there is more elaborated comparison, the presentation of the data remains generic.*

**Reply:** Based on the comment, the sentence was clarified: "NMHC levels are also compared with those from Paris, the latter as representative of a mid-latitude, northern hemisphere (urban) location. It is one of the European megacities, with more than 10 million population. The climate is both oceanic and continental, with cold winters (temperatures can be below 0°C) and mild but wet summers."

*P 5, L21: How much does "significantly" really mean here?*

**Reply:** The remark was already addressed before for the comment of P5 L15-18.

*P 5, L21-22: I am confused about the term "...Cairo rural background area...". It looks like a contradiction to me.*

**Reply:** We agree with the reviewer and the term background was removed.

*P 5, L22: It is not clear what the two values of 5.8 and 7.5 ppb refer to. Do they refer to benzene and toluene, or do they refer to benzene (or toluene?) from Athens and Cairo?*

**Reply:** The remark was addressed previously for the comment of P 5, L15-18.

*P 5, L25-27: The authors neglect to mention the annual variability of other NMHC sources, e.g. evaporation losses.*

**Reply:** As the present manuscript includes only the October 2015 to February 2016 data-set of NMHCs in order to provide information on winter-time sources, annual variability of the other NMHC sources could not be addressed. However, due to the variability of temperatures that have been encountered during October and November, we investigated the relationship between NMHCs levels and temperature. The corresponding section will be placed in the supplement (Sect. S3) as follows:

**S.3 Investigation of evaporation losses (Sect. 3.4.3, Fig. 8).**

In Sect. 3.4.3, the increased mass contribution of butanes and propane to the morning profiles of Thissio and Patission was attributed to LPG fuels, thus to fuel evaporation. To better investigate this possibility, the monthly variation of i-butane relatively to n-butane (Fig. S10) has been examined. The two compounds have linear relationship with no significant temporal differences on the slopes (only October and December equations are presented). In addition, the regression is similar to the one derived from the Patission measurements (depicted on the box on Fig. S10), thus enhancing our assumption that the observations of these compounds are traffic related.

[Figure]

**Figure S10**. **Scatter plots between i-butane relatively to n-butane for October 2015, November 2015, December 2015, January 2016 and February 2016 for the Thissio site. The black line corresponds to the Patission data regression.**

Furthermore, a similar approach as Na and Kim (2001) proposed for Seoul (South Korea) has been applied, in order to investigate the relationship of the ratio Butanes-to-Alkanes C2 – C5

(%) and temperature for every month (Fig. S11). More specifically, the ratio of the sum of i-butane and n-butane versus the sum of ethane, propane, i-butane, n-butane, i-pentane and n-pentane for every sample was calculated. Ethylene, propene and acetylene are excluded from this ratio due to their reactivity. The mean and standard deviation values of the ratio were derived for the temperatures between 1$^{o}$C to 25$^{o}$C (minimum and maximum of the period respectively) and were plotted against the temperature for each month. The highest values of the ratio are observed for high temperatures and the lowest for low ambient temperature, although the standard deviation is considerable. It is interesting to note that the same pattern is observed when the ratio Pentanes-to-AlkanesC2 – C5 (%) versus the temperature is examined (Fig. S12). Taking into account the positive dependence of the two ratios to temperature, we can assume that fuel evaporation losses from cars are also an important source of NMHCs. In addition, the above results could indicate why the Athens tunnel results performed in May differ from Patission and Thissio winter morning profiles."

[Figure]

**Figure S11**. **Scatter plots of the ratio Butanes-to-Alkanes C2C5 (%) to temperature for October 2015, November 2015, December 2015, January 2016 and February 2016.**

[Figure]

**Figure S12**. **Scatter plots of the ratio Pentanes-to-Alkanes C2C5 (%) to temperature for October 2015, November 2015, December 2015, January 2016 and February 2016.**

*P 5, L27: The authors neglect to mention that "atmospheric dynamics" would not only include PBL variations, but also synoptic meteorology. In many cases this would imply enhanced ventilation during wintertime (e.g. through frontal passages).*

**Reply:** We agree with the reviewer but his remark is valid only during nSP, as during the SP events are associated with low winds, i.e stagnant conditions, no frontal passages occur. Note also that in our manuscript the role of meteorology was highlighted (P.6, L.14).

*P 5, L29: The authors mention that "...the winter-time decrease of the PBL is in the range of 20%....". Does this value refer to the maximum daily PBL height, an average daily PBL height or to the daytime or nighttime PBL height? This may all make a difference.*

**Reply:** This decrease by 20% refers to both day and night time and when the wintertime data is compared to the October (reference period). The sentence was corrected as follows: "Nevertheless, according to Kokkalis et al. (personal communication) the mean winter-time

decrease of the PBL is in the range of 20% for both day and night period, thus the limited dilution couldn't be the only factor determining the enhancement of NMHCs level.'

*P 5, L30: I think the authors only refer to vertical dilution only, here.*
**Reply:** Yes, the text will be changed by including the vertical dilution term.

*P 5, L30-31: Those are very generic statements here, as it is well-known that dynamic meteorology plays a major role in the distribution and dilution of atmospheric trace gases and it would be rather surprising that "only one factor" would be important.*
**Reply:** See answer above for the role of dynamic meteorology during SP and nSP events (comment for P5 L27).

*P 6, L3-4: I reiterate my comment made above: Why would the fact that NMHCs nicely follow CO hint towards additional sources e.g. heating? This can also be true in the case of traffic emissions. Also, heating is a very general term. For instance, electric heating would not emit CO and NMHCs locally (just to name one kind of heating)?*
**Reply:** As we have seen before the impact of traffic remains almost unchanged from October to wintertime. Given also the fact that BDL change is within 20% from October to January then the increase in NMHCs could be related to an additional source during wintertime which is heating. Although the word heating is generic the contribution of electric heating is limited in Greece and the majority of the households used fuel for heating before the crisis. During the crisis and due to the increase in fuel and consequently in electricity most of the people used wood for burning. The increased role of wood for burning was clearly seen at the tracers used in this work ($BC_{wb}$). In addition, PMF analysis on aerosol data obtained before and after the crisis clearly showed the absence of wood burning as a source of pollution the years before 2012. Based on the comment, the sentence was clarified as follows: "The amplitude (intensity) of both peaks is gradually increasing from October to winter time by a factor of 3 to 6 and nicely follows that of carbon monoxide (CO), as well as BC and its fractions associated with wood burning ($BC_{wb}$) and fossil fuel combustion ($BC_{ff}$) (Fig. 3). As it was noted in Gratsea et al. (2017), the morning maximum of CO is attributed to morning traffic, while the winter night-time increase to additional sources except traffic, e.g. domestic heating (petroleum oil or wood burning stoves). Although the amplitude of both CO peaks (morning

and night) is almost similar (with the exception of December), the duration of the night peak is at least by a factor of 2 larger, which could imply the predominant role of heating in air quality during wintertime. Consequently, these observations are indicative of the contribution of traffic and heating to the NMHCs levels".

*P 6, L4-7: From Fig 3 I see that Bff increases similarily to Bwb at night. Why the authors make the statement that traffic would not be as important as heating?*

**Reply:** The remark was also addressed before. Traffic is also a night-time emission source of NMHCs, however the increase in $BC_{ff}$ during night is due both to traffic and fossil fuel used for heating. A comparison between the morning $BC_{ff}$ peak which is exclusively due to traffic to that observed during night clearly shows that although amplitude could be similar the duration of the night peak is much longer (by almost a factor of 2). In addition, as it follows that of $BC_{wb}$ emitted by wood burning for heating, indicates that heating rather than traffic is the main contributor of the $BC_{ff}$ during night. However to avoid misinterpretation the phrase was changed into: "By comparing the NMHC diurnal variability with that of BC, as well as its fractions associated with wood burning ($BC_{wb}$) and fossil fuel combustion ($BC_{ff}$), it is deduced that the morning peak could be mainly attributed to traffic and the late evening to both traffic and heating, the later from the combined use of heavy oil and wood burning.'

*P 6 L9-10: I disagree. Usually, PBL heights are at a minimum during morning hours before sunrise, unless the authors can show other evidences for their statement.*

**Reply:** The statement for a higher morning PBL relatively to the night-time concerns the period after the sunrise, i.e around mid-day and morning was replaced by "mid-day".

*The role of meteorology on NMHC levels*

*P 6 L12: Some NMHCs also react with Cl. The latter potentially important for coastal areas. Also, in principle all reactions occur throughout the entire day. It just depends on the availability of reacting compounds.*

**Reply:** On the specific section only the major NMHCs reactions were considered, by taking into account that the Thissio site is located in an urban location. Nevertheless, the Cl role could be mentioned, since marine originated air masses could affect the area. Thus the text is changed into: "Once emitted in the atmosphere, NMHCs react mainly with OH and $NO_3$ radicals during day and night-time, respectively, and with ozone throughout the day (Crutzen

1995, Atkinson 2000), whereas the role of Cl could not omitted, especially for coastal areas such as GAA (Arsene et al., 2007)."

*P 6 L15-20: This is a pretty generic description. It is well-known that the concentration of primarily emitted gaseous pollutants will decrease due to dilution regardless of their chemical class. However, windspeeds < 3 m/s alone would not indicate the presence of local sources. This would only be true for calm winds. From the plots it seems like these are skewed distributions with maximum concentration values around 2 m/s or so. This would rather indicate some regional flow impacts, which the authors neglected to consider. It seems a more elaborate analysis of windspeeds and their effect on NMHC levels in the Athens area has already been presented in Rappengluck et al (1998). With regard to potential long-range transport it is actually interesting to see that there is some acetylene data still around 5 ppb or so at windspeeds around 9 m/s and higher. In fact, those are very high acetylene values despite strong dilution. What is the reason for this?*

**Reply:** We would like to thank the reviewer for this comment that helped us investigate this issue. For the better interpretation of the effect of wind speed and wind direction, the graphs of figure 5 were changed. According to the figure 5, high concentrations of NMHCs are mainly depended on wind speed, since they occur only for wind speeds ranging from 1 to 3 m s$^{-1}$ and independently on wind directions.

Furthermore, concerning the acetylene issue, we isolated the corresponding concentrations of the enhanced baseline levels based on the acetylene-to-CO ratio. After investigating the possible relationship of these levels with other compounds or meteorological factors, we found no specific relationship or trend for these values. For that reason, we examined the corresponding chromatograms and we found a slight moving of the peak of acetylene only, which is an indication of mis-identification of this compound. As it is not clear if the high values are due to an instrumental default or a co-elution with another compound, while there are indications that there is a link to the prolonged SP events that occurred in this period with the few nSP in-between, we decided to remove the values (293 values were removed out of 2862, i.e c.a 10% mainly from December and January).

*P 6 L21-22: In Fig 5a it is quite surprising to see that the minimum occurrence (north; 1% occurence) is just side-by-side with the maximum occurrence (northeast; I would guess 27% occurence). What is the reason for this quite unusual wind direction occurrence distribution?*

**Reply:** We thank the reviewer for his/her comment that helped us investigate this matter. Due to an error in the calculation of the mean wind direction values, we re-plot the data by using hourly values and the new figure reveal increased contributions under the N sector.

*P 6, L24-29: Large part of the discussion here contradicts the authors' statement on the dependence of NMHC on windspeed made earlier. For instance, it looks to me that strongest wind speeds (e.g. NE) would not necessarily be associated with lowest concentrations, while lowest wind speeds (e.g. SE) would not necessarily be associated with highest concentrations.*
**Reply:** The remark was addressed before for the comment of P6 L15-20 and P6 L21-22.

*P 6, L29: Those sources cannot be defined as local sources any more, as they are not located in the immediate vicinity of the Thissio site.*
**Reply:** With the word "local" we refer not to immediate vicinity of Thissio but to the GAA in general as our site is considered as representative of a wider area in the GAA.

*P 6, L30-32: I do not understand this sentence. On the one hand the authors mention increased number of fireplaces, on the other hand the authors mention central heating systems (wouldn't central heating systems decrease the number of individual fireplaces?). Also, the authors state that the higher NMHC values to the N sector is due to the age of the buildings. Isn't another (potentially more) important factor that a higher fraction of Athens' population may be located north of Thissio than towards other directions? Also, wouldn't southerly winds bring in cleaner marine air, making the S-N difference in the NMHC concentration not even more drastic? If the authors want to point out heating sources, wouldn't it make sense to distinguish between day- and nighttime?*
**Reply:** According to the demographic data of 2011, the population around the center of Athens is almost equal distributed (1 million citizens in the center and approximately 5 hundred thousand citizens in every sector around the center – North, West and South) which means that the population distribution couldn't be the only factor driving the emissions. Apart from the southern cleaner marine originated air masses but with population of low income, other factors could also affect the pollution sources of the city. The majority of the buildings in GAA have central heating systems regardless their age; while the residencies that have been constructed the last 20-30 years usually have both fire places and central heating,

reflecting the combined use of several types of fuels such as wood, fuel, or other mixtures for heating purposes. The northern part of Athens has been mostly constructed based on this mixed heating systems pattern and in combination with the financial welfare of its residents relatively to areas close to the historic center of the city, south and west sectors, enhanced use of heating systems and thus increased emissions could be expected from this direction.

Based on the comment, we focused our investigation on morning and nights of October and December 2015 and it is apparent the contribution of the N to ESE sector to the December night concentrations especially at night. Consequently, the text was corrected accordingly: "The impact of the N to ESE sector on NMHCs levels can be also seen when comparing the concentrations of the morning (07:00 – 09:00) and night (21:00 – 23:00) peak of October and December. The wind probability from N to ESE is similar for both months, however significantly higher concentrations are observed in the December night peak, that seem to affected by low wind speed ( <2 m s$^{-1}$) that mainly originates from the N to NE sector."

*P 7, L4: What biogenic compounds are not triggered by temperature?*

**Reply:** Based on the comment, the phrase was clarified as follows: "…..also trigger the production of biogenic compounds, whereas……"

*Identification of NMHC emission sources*

*P 7, L28: Remove "locally", as dispersion acts on all airborne compounds and is not confined to locally emitted pollutants.*

**Reply:** Following the suggestion of the reviewer the term "locally" was removed.

*P 8, L8: Mention the NMHC tracers for wood burning exactly.*

**Reply:** Based on the comment, the sentence was clarified as follows: "….as emission source of specific organic compounds, such as ethane, ethylene, acetylene, benzene, methanol, acetaldehyde and acetonitrile…."

*P 8, L28-30: Did the authors also apply the baseline subtraction for the tunnel measurements?*

**Reply:** Due to the small number of samples a baseline subtraction from the tunnel data was not possible.

*P 8, L30-31: How can the authors justify that their tunnel measurements are not influenced by outside air masses?*

**Reply:** To avoid any dilution from outside air masses the tunnel measurements have been conducted in the middle.

*P 8, L31-32: I completely disagree on the authors' statement. The authors neglect to mention what they consider "dominant species", however just looking into NMHCs such as acetylene, benzene, and toluene, the two profiles "Morning Peak Athens" and "Highway Tunnel - Athens" are completely different: while acetylene for the "Morning Peak Athens" is about 6-7 times higher than for the case "Highway Tunnel - Athens", benzene and toluene values are about 2-3 times lower at the same time..*

**Reply:** We would like to thank the reviewer for his detailed questions regarding our tunnel measurements. Indeed, our tunnel measurements were performed in May and temperature is different compared to winter-time. Note also that the tunnel experiments in Paris were performed in autumn that is a warmer period for Paris compared to winter.

To resolve the issue on the possible temperature influence, a campaign was conducted during winter time in Athens by collecting samples at Patission (a street canyon heavily influenced by traffic). Patission profile reflects all types of traffic-related emissions due to the combination of the high number of vehicles and buses that cross this street, the traffic jam, the variety of types of fuels, vehicles age and their maintenance etc. Samples were collected during the morning rush hours (from 6-10am) and reported together with the tunnel data at figure 8.

Several conclusions can be drawn from this figure. Firstly, the two morning profiles although performed at sites with different impact of traffic agrees quite well ($R^2 > 0.97$). Iso - pentane and toluene are the two main compounds contributing to the morning profiles accounting by about 44% of the total measured NMHCs at both locations, followed by n- and i-butane and ethylene accounting for almost 30%. Differences among the two morning profiles between these 5 main species are minimum (less than a factor of 1.5). Note also that the morning

profile at Thissio is the mean of a whole month period compared to a campaign of two days in Patission which can explain the differences between the two profiles.

The profiles obtained at the two tunnels although differ in terms of tunnel length, city, and period have a lot of common features. Again i-pentane and toluene are the two main compounds of the profile accounting by about 56% of the total measured NMHCs at both sites, followed by n -butane, ethylene and benzene accounting for almost 20% again at both sites. The most striking difference between the two sites concerns n-pentane (almost a factor of two higher in Paris compared to Athens). Despite the differences between the two tunnel studies the similarity is almost 80% ($R^2$ > 0.91). The biggest difference between the two Athens morning peaks and tunnels concerns acetylene (factor of 4), benzene and toluene (factor of 2). The similarity of Thissio and Patission morning profiles and their difference from the Athens tunnel profile, as well as the Paris tunnel profile, indicate the importance of the type of fuel used. The latter is also implied in recent works (Ait-Helal et al., 2015; Q. Zhang et al., 2018; Y. Zhang et al., 2018), where important differences are reported between tunnel measurements worldwide, attributed to the variance of the car-fleet (type of vehicle and fuel). In our case there is a possibility that the car-fleet in the tunnel is not representative for the GAA, since the existing tolls in short distance reduce the use of the tunnel due to financial issues. The latter scenario is further enhanced by the time of the measurements, since the traffic density is quite low at noon.  In any case, the prevalence of i-pentane and toluene in all profiles, indicate the continuing dominance of gasoline powered cars. Moreover, higher values of ethane, propane and butanes that are depicted in the morning peaks of the urban sites relatively to the tunnel measurements reflect the increased number of LPG powered vehicles in Athens and natural gas-powered buses (Fameli and Assimakopoulos, 2016). In fact, the connection of high levels of C2-C4 alkanes and the number of LPG-powered cars is highlighted in other tunnel works as well (Ait-Helal et al., 2015; Q. Zhang et al., 2018).

In the revised manuscript, the corresponding paragraph will be re-written as follows:

"As discussed in Sect. 3.2, the morning peak (07:00 – 10:00 LT) of NMHCs could be attributed mainly to traffic. Figure 8 presents the profile of this peak (% mass contribution of the measured NMHCs), during January and February SP days when toluene data were available. Additionally, in the same figure the morning profile obtained from a 2 – days campaign conducted in Patission Monitoring Station (a street canyon located at the center of Athens) and the profiles of two tunnel measurements in G.A.A and Paris are reported. Details

on the calculations for the morning profile for the two sites are provided in Sect. S2. Patission profile reflects all types of traffic-related emissions due to the combination of the high number of vehicles and buses that cross this street, frequent traffic jam conditions, the variety of types of fuels, vehicles age and their maintenance etc.

The two morning profiles, although performed at sites with different impact of traffic, agrees quite well ($R^2 > 0.97$). Iso - pentane and toluene are the two main compounds contributing to the morning profiles accounting by about 44% of the total measured NMHCs at both locations, followed by n- and i-butane and ethylene accounting for almost 30%. Differences among the two morning profiles between these 5 main species are minimum (less than a factor of 1.5). Note also that the morning profile at Thissio is the mean of a whole month period compared to a campaign of two days in Patission which could explain the differences between the two profiles. The profiles obtained at the two tunnels although differ in terms of tunnel length, city, and period have a lot of common features. Again i-pentane and toluene are the two main compounds of the profile accounting by about 56% of the total measured NMHCs at both sites, followed by n -butane, ethylene and benzene accounting for almost 20% in total again at both sites. The most striking difference between the two sites concerns n-pentane (almost a factor of two higher in Paris compared to Athens). Despite the differences between the two tunnel studies the similarity is almost 80% ($R^2 > 0.91$). The biggest difference between the two Athens morning peaks and tunnels concerns acetylene (factor of 4), benzene and toluene (factor of 2). The similarity of Thissio and Patission morning profiles and their difference from the Athens and Paris tunnel profiles, indicate the importance of the type of fuel used. The latter is also concluded in recent works (Ait-Helal et al., 2015; Q. Zhang et al., 2018; Y. Zhang et al., 2018), where important differences are reported between tunnel measurements worldwide, and attributed to the variance of the car-fleet (type of vehicle and fuel). In our case there is a possibility that the car-fleet in the tunnel is not representative for the GAA, since the existing tolls reduce the use of the tunnel due to financial issues. Also, measurements are performed during noon when the traffic density is quite low.  In any case, the prevalence of i-pentane and toluene in all profiles, indicate the continuing dominance of gasoline powered cars. Moreover, higher values of ethane, propane and butanes that are depicted in the morning peaks of the urban sites relatively to the tunnel measurements reflect the increased number of LPG powered vehicles in Athens and natural gas-powered buses (Fameli and Assimakopoulos, 2016). In fact, the connection of high levels

of C$_2$-C$_4$ alkanes and the number of LPG-powered cars is highlighted in other tunnel works as well (Ait-Helal et al., 2015; Q. Zhang et al., 2018)."

*P 8, L32 - P 9, L1: I disagree here again! I do not see that profiles fit nicely. Instead, there are a lot of significant differences. Also, what do the authors consider "common NMHCs"?*
**Reply:** The remark was addressed in the previous comment for P 8 L31-32.

*P 8, L1-3: Why should there be higher traffic related butane fraction due to evaporation in ambient air than in the tunnel? Even more surprising, as the tunnel measurements were taken in May, which presumably has warmer temperatures than wintertime. Also, when butanes should be related to evaporation why does propane, another prominent tracer for evaporation, show pretty similar values in the tunnel measurements compared to the "Morning Peak Athens" data?*
**Reply:** The remark was addressed before in the comment for P 8, L31-32.

*P 8, L7: "...during both months...". I disagree with this statement, as only results from two selected days are shown.*
**Reply:** We don't agree. At page 8, line 17, the results from two selected months are shown, not days.

*P 8, L8-10: If this justification is true for ethylene, why would it be different for i-pentane? Still, photochemical decay should also be more active for i-pentane in October than in December. However, i-pentane shows a lower slope in December than in October contrary to ethylene. Also, could a difference in solar radiation energy in October vs December in the 7:00-10:00 LT time frame explain an increase in the slope of ethylene by about 60%?*
**Reply:** First of all, after the correction of the ethylene concentrations, the slope of the graph changed. Based on the new graph, the December increase to the slope is 34%. Thus, given that also in all graphs the difference cases in slopes is within 30% we can assume that slopes remain almost stable during both months and the comment on ethylene was removed.

*P 8, L12-14: The definition of the background concentration appears odd. How can the minimum value between 12:00 - 17:00LT be representative for the nighttime period 18:00-*

*05:00LT? Both are pretty long periods (5 and 11 hours, respectively). From Figs. 4 and 5 we learnt that the NMHC concentration critically depends on wind speed and wind direction. How can the authors make sure that such changes in wind speed and/or wind direction would neither occur during the daytime reference period nor during the nighttime period?*

**Reply:** As background value we use the minimum value observed during mid-day. For the morning and night profiles (Fig. 8 and 10) we used in the analysis only SP days, which means that wind speed is always less than 3m/s during all hours, and thus no major changes in wind speed, which is the most important parameter controlling NMHCs levels, are expected.

*P 8, L16-19: Are these differences statistically significant?*

**Reply:** Yes differences are statistically significant ($p<0.01$).

*P 10, L9-10: Not sure, how the authors know that BCff at night is due to fossil fuel heating only, and not also impacted by traffic.*

**Reply:** This comment was already addressed before for P.6 L4-7.

*P 10, L13: What were those "different meteorological profiles"?*

**Reply:** Based on the comment, the sentence was clarified as follows: "……due to different temperature profiles….."

*P 10, L14-16: This is not supported by the data presented in the paper!*

**Reply:** The discussion on the profile of the morning peak was changed as discussed in the earlier comments related to tunnel.

*P 10, L19-20: There are already 4 months of continuous NHMC measurement available. Why is there a longer data set needed to distinguish different source types?*

**Reply:** Based on the comment the sentence was clarified as follows: "An extended dataset of NMHCs and other organic tracers (future long-term measurements) is needed in order to distinguish the contribution of different sources types on seasonal basis and to further quantify their impact on the NMHCs levels".

*Table 1: Remove the term "worldwide" in the table caption, as Table 1 shows a few selected data from the Mediterranean/European area at the most. What does the second sentence of the table caption refer to? What quantities are compared in this table: means or medians or ....? What do the authors mean by "sampling" frequency: sampling duration or measurement cycle? There is no information given for "sampling frequency" for Baudic et al., Salameh et al., and Durana et al.. Why are the results for the summer 2012 and winter 2013 Athens campaigns reported by Kaltsonoudis et al (2016) not listed in this table? At least, results for isoprene, benzene, and toluene would be comparable.*

**Reply:** The caption of Table 1 has respectively been corrected. Mean levels are used in the table. The second sentence referring to information about sampling or analysis and sampling frequency was replaced by data resolution. The results of Kaltsonoudis et al. (2016) are not listed on the table since datasets with numerous common compounds have been selected for the comparison. Nevertheless, the authors include now the compounds in common according to the suggestion of the reviewer.

*Figure 1: There is a quite unusual long-term baseline increase of acetylene starting about 1 ppb at the end of November until early January, when it reaches a bit more than 5 ppb (which is pretty high for an urban background site!). Then it abruptly decreases. This feature is not seen in other NMHCs shown in this plot. What is driving (a) this continuous increase and (b) its abrupt decrease?*

**Reply:** The remark was addressed before for the comment of P6 L15 – 20.

*Figure 2: Why is the mean and not the median shown, as Fig 1 clearly shows that NMHC data is not normally distributed?*

**Reply:** A box-wisker plot was now used to better represent the NMHC data.

*Figure 3: Is the data shown based on mean or median hourly averaged values? It should be median values.*

**Reply:** The data shown on the graphs are based on mean values, since part of this paper is focused on the elevated concentrations of the SP period. For that reason, we believe that they are better represented by mean than median values.

*Figure 4: The figure should be better shown with Box-Whisker plots for designated windspeed classes, as the data cloud in Figure 4 could be misleading, as the number of overlapping data points might differ significantly.*

**Reply:** The graphs was re-made accordingly.

*Figure 5: What quantity for the NMHC data is shown: mean or median data? Standard deviation bars should be included.*

**Reply:** Figure 5 was replotted again with all data**.**

*Figure 8: Error bars should be included.*

**Reply:** Error bars can be added only for the Thissio data.

*Figure 10, figure caption: I disagree that such a long time period (18:00-05:00 LT; 11 hours!) can be considered a nighttime "peak". Error bars should be included. I do not see that the values shown in the figure add up to 100%.*

**Reply:** As there is no light during this period and this is the period that heating starts, for simplicity it is considered as nighttime. Also, as discussed before this is the best way to consider the evening/night maximum which is quite broad (more than 8h).

Error bars have been included. Finally, by adding-up the values the sum is almost 100%. Differences are too small.

**References**

[revised manuscript text omitted]

---

## Referee Report (RR1)

The authors made considerable efforts to modify the paper. However, unfortunately, there are still multiple issues which have not been addressed in a satisfactory way. Still, the current state of the paper and the replies made by the authors do not warrant the publication of the paper in its current stage in ACP without additional discussion.

Here I repeat those issues, which I believe need some additional clarification. First I list my initial review point, then I add my updated remark, denoted as "Reviewer Remark":

1) On the background of existing literature I am not sure what the real novelty of this paper in terms of methods and results is. While I agree that C2 and C3 measurements have not yet been done before in Athens, it seems the inclusion of these does not yield more findings than already reported by Kaltsonoudis et al (2016). On the other hand, the Panopoulou et al paper makes same sketchy description of meteorological impacts on NMHCs, but lacks some elaborate analysis similar to those presented in Rappengluck et al (1998) for transport effects and also Kourtidis et al (1999) for temperature effects for Athens. As the authors make an important point on page 2, L27-29, that there have been new conditions during the economic crisis years (i.e. competing traffic vs wood burning emissions) it would be actually meaningful to perform a comparative analysis between the data sets reported 20 years ago and the ones reported by Panopoulou et al. It seems both studies include continuous NMHC measurements and PMF source apportionment analysis would be feasible and would provide interesting insights.

**Reviewer Remark:**
In their reply the authors did not really demonstrate the novelty of their work apart from the fact their measurements include C2-C3 data. However, including these NMHCs did not yield new knowledge about NMHC sources in Athens beyond the paper by Kaltsonoudis et al (2016). Kaltsonoudis et al (2016) even included detailed source apportionment for traffic and biomass burning for the Athens case, which the authors did not apply and did not address. What do we learn from this paper which has not yet been described earlier. Even worse, now the authors have removed the reference to Kourtidis et al (1999) who described temperature effects on traffic NMHCs emissions for the same city.
The authors argue: "However direct comparison with the work performed 20 years ago is difficult considering differences in sampling period (summer versus winter and thus different photochemistry), location, sampling method and analytical techniques." I cannot completely follow this argument. Emission studies can be done for nighttime periods minimizing photochemical effects, all the locations are within the GAA and sampling methods and analytical methods are similar. Even, if the suite of NMHCs may not be exactly the same, the application of source apportionment methods such as PMF would just require a few

representative NMHC describing source profiles. There is not necessarily a "complete" suite of NMHCs necessary as it is also shown by the Kaltsonoudis et al (2016) paper, who applied a rather limited VOC dataset. Did the authors try at all to make comparisons? Again, in my opinion this would be feasible and would provide interesting insights.

**Abstract:**

Page 1 L20-21:
What do the authors exactly mean by local meteorology, as this term is quite unusual? Its connotation would mean that it is not representative for a larger fetch.
I disagree that "local" meteorology would control the variability of NMHC levels alone. What about the temporal variability of NMHC emissions?

**Reviewer Remark:**
In general, the term microscale meteorology is fine. However, still this would rather describe specific condition for mixing processes observed at a given site, which may not necessarily be similar for a different location within the urban area. For instance, the built environment would have a significant impact on these exchange processes.
In addition, what about PBL variations, which are not included in the term microscale meteorology? Also, still I believe, that the temporal variability of NMHC emissions is an important factor. Just one example: a typical morning rush hour peak is not entirely explained by low winds that are often observed during the same time frame. I would believe that low a PBL height and increased traffic emission would be an important factor. I would still doubt that "Microscale meteorological conditions and especially wind speed seems to **_control_** the variability of NMHC levels...".

Page 1 L27-29:
Why does the present data not allow for the quantification of the relative contribution of fossil fuel and wood burning for heating purposes?

**Reviewer Remark:**
Still, I have some trouble with the revised sentence. On one side the authors state that "For the night peak, the selected tracers and profiles **_clearly_** indicate contribution from both traffic and domestic heating..." and then continue that more specific tracers would be necessary for quantification of these sources. The latter statement contradicts the first statement: either the data set has tracers and profiles that clearly indicate these specific sources or not. Also, the authors emphasize the night peak. This implies that daytime emissions would be different. I would assume that domestic heating sources would become less, while traffic sources would

increase and it seems the authors can differentiate this different behaviour in their data-set. Why would, for instance, a source apportionment method based on these specific tracers and profiles not be able to estimate relative source contributions?

**Introduction:**

Page 2 L30: It would be fair to mention how many NMHCs were actually measured as it seems that the paper does not report some important NMHCs such as 1,3-butadiene and others, for instance.

**Reviewer Remark:**

The reply given by the authors is not convincing. Still, it would be fair to mention the number of NMHCs measured in this range and the number of NMHCs in this range used in this study in order to make a precise statement here and avoid any speculation.

Page 2 L34 - page 3 L1: The authors should mention why the analysis is restricted towards traffic and heating impact on NMHC levels.

**Reviewer Remark:**

There is still an issue: in order to perform an "..investigation of traffic and residential heating impact _**on**_ the NMHC levels..." the authors would still need to verify and quantify any other potential NMHC source, regardless of its magnitude. Even, if there are no industrial sources as the authors point out, I still would think there are solvent and evaporative sources. Also, what about the port of Athens?
Otherwise, I am not sure what still needs to be investigated exactly for this task (3).

**On line NMHC measurements**

Page 4, L5: Why is only toluene used? Why not at least ethylbenzene and the xylenes in addition? Would the exclusion of these NMHCs not introduce a bias into the data analysis, as important tracers for solvent emissions are excluded? What are the uncertainties and the detection limits for this GC?

**Reviewer Remark:**

This appears cherry-picking to me. The authors state that they focus on C2-C6 NMHCs, then include a C7 compound from a different instrument, but exclude other NMHCs from that different instrument. The authors did not address my question whether the exclusion of these other NMHCs would introduce a bias in the data analysis. As the authors have a complete suite of NMHC measurements, it would be straightforward to include all NMHCs, and make a

source apportionment analysis first, and not rely on a reference only (Vrekoussis et al., 2013), which did not include VOCs.

**Tunnel measurements**

Page 4 L16: The authors should mention the length of the tunnel, whether lanes were for both directions (there could also be dedicated tunnels for one direction only), if there was any artificial ventilation and if there might have been any limitations on traffic through this tunnel (in some cases heavy duty traffic is not allowed). In any case an estimate of the traffic fleet composition (e.g. heavy duty vs light duty vehicles) would be helpful. All these factors have an impact on the NMHCs levels. At what location of the tunnel did the authors make the measurements exactly? I see the measurements were taken on 12 May 2016, which is different from wintertime. Wouldn't the temperature be different from wintertime and wouldn't this have an enhanced impact on NMHC emissions through evaporation, for instance?

**Reviewer Remark:**

It seems the description of the tunnel is still missing in the manuscript. Also, the authors did not mention explicitly whether this tunnel contains all lanes for both driving directions or whether there were separated compartments for each driving direction. They did not mention either, whether there was any artificial ventilation or not.

**Temporal variability of NMHCs**

Page 4 L28: I do not understand the concept of data coverage here, as it is not explained. It could refer to the percentage of data above the detection limit vs maximum available data, but this does not make complete sense, as I doubt there were any data of ethane below the detection limit, for instance. However, it cannot be true either that it refers to the data availability vs maximum potential data availability during the time period reflecting instrumental potential instrumental malfunctions and/or failure. This should be clarified. The only thing I understand is that the there has been some interruption of NMHC data contrary to what the authors claim in the abstract of the paper.

**Reviewer Remark:**

The authors statement of 87% data availability would imply that the instrument did not work uninterrupted (again, I believe that such a species like ethane would always be above the detection limit, if the instrument is working properly). As a consequence I would recommend to replace the statement made in the abstract of the paper (page 1, L16) "..to our knowledge,

time resolved, uninterrupted data of NMHCs..." by "..to our knowledge, time resolved data of NMHCs...".

Page 5, L15: The authors should clarify why the reader should bear in mind differences in sampling methods and analytical techniques. Are some of the sampling methods and/or analytical techniques and associated results listed suspicious and cannot be compared to each other?

**Reviewer Remark:**
The authors did not change much. The authors still state that "....this decrease has to be seen with cautious considering differences in [...]sampling method and analytical techniques". Are some of the sampling methods and/or analytical techniques and associated results listed suspicious and cannot be compared to each other?

Page 5, L18-20: The same comment as above applies here. As long as there is no more elaborated comparison, the presentation of the data remains generic.

**Reviewer Remark:**
Why would it be important to note that Paris is located at the mid-latitude and in the northern hemisphere with regard to NMHC emissions? Would it be different for city at the same latitude, but in the southern hemisphere? For instance, would traffic emissions depend on the latitude?
There is not such a thing as oceanic continental climate; this does not exist! Apart from that the climate of Paris is definitely not continental! There are no cold winters in Paris! Also, in Paris, it can be pretty hot during summer time, which cannot be considered "mild". I would rather use that term for the winter period in Paris. The authors want to check an appropriate climate description for Paris.

Page 5, L25-27: The authors neglect to mention the annual variability of other NMHC sources, e.g. evaporation losses.

**Reviewer Remark:**
With regard to traffic related evaporative NMHC emissions the authors need to consider the work by Kourtidis et al. (1999) who specifically addressed the processes related to these emissions for the same city, Athens. The NMHC mixture might have changed, but not the underlying processes. Unfortunately, the authors removed this reference from the previous manuscript version.

Page 6, L4-7: From Fig 3 I see that $B_{ff}$ increases similarly to $B_{wb}$ at night. Why can the authors make the statement that traffic would not be as important as heating?

**Reviewer Remark:**

The authors do not consider the nighttime variation of the PBL which would also contribute to a sustained higher level of pollutants, regardless of their origin.

Page 6 L9-10: I disagree. Usually, PBL heights are at a minimum during morning hours before sunrise, unless the authors can show other evidences for their statement.

**Reviewer Remark:**

I did not see the change the authors may have done in the text (mentioning the line would be helpful).

**The role of meteorology on NMHC levels**

Page 6 L15-20:

This is a pretty generic description. It is well-known that the concentration of primarily emitted gaseous pollutants will decrease due to dilution regardless of their chemical class. However, windspeeds < 3 m/s alone would not indicate the presence of local sources. This would only be true for calm winds. From the plots it seems like these are skewed distributions with maximum concentration values around 2 m/s or so. This would rather indicate some regional flow impacts, which the authors neglected to consider. It seems a more elaborate analysis of windspeeds and their effect on NMHC levels in the Athens area has already been presented in Rappengluck et al (1998).

With regard to potential long-range transport it is actually interesting to see that there is some acetylene data still around 5 ppb or so at windspeeds around 9 m/s and higher. In fact, those are very high acetylene values despite strong dilution. What is the reason for this?

**Reviewer Remark:**

As the x-axis of figure 5 shows discrete values, it would be good to have a clear definition of these ranges. Also, are calm wind situations considered, i.e. below the threshold of the wind sensor (both, in figures 5 and 6)?

**Identification of NMHC emission sources**

Page 8, L28-30: Did the authors also apply the baseline subtraction for the tunnel measurements?

**Reviewer Remark:**

The authors should mention this limitation in the text.

Page 8, L30-31: How can the authors justify that their tunnel measurements are not influenced by outside air masses?

**Reviewer Remark:**

There must be some dilution. A tunnel is not a closed box and it seems the length of the tunnel is only 200 m.

Page 8, L31-32: I completely disagree on the authors' statement. The authors neglect to mention what they consider "dominant species", however just looking into NMHCs such as acetylene, benzene, and toluene, the two profiles "Morning Peak Athens" and "Highway Tunnel - Athens" are completely different: while acetylene for the "Morning Peak Athens" is about 6-7 times higher than for the case "Highway Tunnel - Athens", benzene and toluene values are about 2-3 times lower at the same time.

**Reviewer Remark:**

I agree that the "Morning Peak Thissio" and "Morning Peak Patission" profiles agree with each other within the uncertainties. Still their comparison with the tunnel measurements and both tunnel measurments among themselves are very different. The authors themselves make a reference to various papers which report important differences between tunnel measurements worldwide and also state that there is a possibility that the car-fleet in the tunnel is not representative for the GAA. Why then compare two consistent real-world street canyon measurements with tunnel measurements, whose data is questionable?

Page 8, L32 - Page 9, L1: I disagree here again! I do not see that profiles fit nicely. Instead, there are a lot of significant differences. Also, what do the authors consider "common NMHCs"?

**Reviewer Remark:**

See my Reviewer Remark above/

Page 8, L1-3: Why should there be higher traffic related butane fraction due to evaporation in ambient air than in the tunnel? Even more surprising, as the tunnel measurements were taken in May, which presumably has warmer temperatures than wintertime. Also, when butanes should be related to evaporation why does propane, another prominent tracer for evaporation, show pretty similar values in the tunnel measurements compared to the "Morning Peak Athens" data?

**Reviewer Remark:**

The remarks made by the authors in the comments for P8, L31-32 were: "Moreover higher values of ethane, propane and butanes that are depicted in the morning hours at the urban sites relatively to the tunnel measurements, reflect the increased numbers of LPG powered vehicles in Athens and natural gas-powered buses". I deduce again, that the tunnel measurements were not representative for the GAA. For me, it does not make much sense to include the tunnel data then.

Page 8, L12-14: The definition of the background concentration appears odd. How can the minimum value between 12:00-17:00LT be representative for the nighttime period 18:00-05:00LT? Both are pretty long periods (5 and 11 hours, respectively). From Figs. 4 and 5 we learnt that the NMHC concentration critically depends on wind speed and wind direction. How can the authors make sure that such changes in wind speed and/or wind direction would neither occur during the daytime reference period nor during the nighttime period?

**Reviewer Remark:**

The authors did not address the impact on wind direction. Background NMHC concentrations can critically change with wind direction, in particular at coastal locations. How can one minimum NMHC value observed during mid-day and potentially under one distinct wind direction be representative for all potential wind direction conditions at nighttime from 18:00-05:00 LT? Also, the authors neglect daytime photochemical degradation, albeit limited, but still present in those months.

Page 8, L16-19: Are these differences statistically significant?

**Reviewer Remark:**

The statement made by the authors should be added in the manuscript.

Page 10, L13: What were those "different meteorological profiles"?

**Reviewer Remark:**

Still, the term "profile" does not make sense, as long as it is not defined (for instance, there are vertical temperature profiles).

Page 10, L14-16: This is not supported by the data presented in the paper!

**Reviewer Remark:**

Still, the tunnel measurements would not support this as discussed earlier.

Table 1: Remove the term "worldwide" in the table caption, as Table 1 shows a few selected data from the Mediterranean/European area at the most. What does the second sentence of the table caption refer to? What quantities are compared in this table: means or medians or ....? What do the authors mean by "sampling" frequency: sampling duration or measurement cycle? There is no information given for "sampling frequency" for Baudic et al., Salameh et al., and Durana et al.. Why are the results for the summer 2012 and winter 2013 Athens campaigns reported by Kaltsonoudis et al (2016) not listed in this table? At least, results for isoprene, benzene, and toluene would be comparable.

**Reviewer Remark:**
The authors should state explicitly that mean values are shown in the table.

Figure 10, figure caption:
I disagree that such a long time period (18:00-05:00 LT; 11 hours!) can be considered a nighttime "peak". Error bars should be included. I do not see that the values shown in the figure add up to 100%.

**Reviewer Remark:**
I did not argue the term "nighttime", I argued the term "peak", which does not make much sense for a time period of 11 hours.

References:
Kourtidis, K. A., Ziomas, I. C., Rappenglueck, B., Proyou, A. and Balis, D.: Evaporative traffic hydrocarbon emissions, traffic CO and speciated HC traffic emissions from the city of Athens, Atmos. Environ., 33(23), 3831–3842, doi:10.1016/S1352-2310(98)00395-1, 1999

---

## Author Response (AR2)

Reference article: acp-2017-936

Title: Non Methane Hydrocarbons variability in Athens during winter-time: The role of traffic and heating

**The authors want to thank both the editor and the reviewer for their helpful comments. All these comments have**
5 **been considered in the revised version and below a point by point answer is presented. To facilitate their reading we**
**decided to first summarize the main and significant changes performed in the new manuscript, before presenting the**
**answers to the reviewer.**

**Changes in the new manuscript**

10 On the new version of the manuscript the following major changes were performed according to the remarks of the reviewer and the editor:

- Our dataset was compared to previously reported values for the area not only arithmetically (as in the submitted version), but also by means of the emission ratios calculated using the same procedure presented in the work of Kourtidis et al., (1999), performed almost 20 years ago at the same street canyon (Patission). The comparison
15 depicted evidence of changes on the emissions sources over time. To sum up, we use the same tools as the previous (and past) publications with VOC measurements in the same area of Athens.

- Furthermore, an extra comparatively study with the past findings is made in the new version of the Supplement. We recall that in order to address the concerns of the reviewer about evaporative losses, we examined the ratio of Butanes-to-(C2 – C5)Alkanes (%) and Pentanes-to-(C2 – C5)Alkanes (%) relatively to temperature at the
20 previously submitted version (as an answer in his/her 1st peer-review). Now, in the new version of the manuscript and the supplement, we complete the previous investigation by following the same methodology as in Kourtidis et al. (1999), thus we additionally examined the relationship of the normalized to OH reaction ratio of NMHC/benzene for the 15:00 and 07:00 to the boiling point of the selected NMHCs (as an answer to the updated suggestions of the reviewer to his/her 2nd peer-review). The latter is compared to the observations in Kourtidis et al. (1999).

25 - The tunnel part was transferred to the Supplement and corrections on Figure 9 (comparison of morning profiles) were accordingly made. In addition, following his/her suggestions we added in the profile (Figure 9) results on xylenes and ethylbenzene. The reasons that could possibly lead to the differences between the tunnel and street canyon observations were already explained at the previous version and still exist at the supplementary material justifying our findings and our choice to maintain the tunnel part on our study.

30 In addition, we would like to point out (as indicated from the title and from the introduction) that the objective of this work is not VOC source apportionment in Athens, but the investigation of the contribution of traffic and heating to the observed levels.

**Answers to reviewer**

We would like to thank the reviewer for his/her comments which help us to improve the submitted version. Below is a point by point reply to the comments. Changes on the manuscript (discussed on the authors reply) are indicated with italics. Furthermore, minor grammar and typo corrections were done on the manuscript on behalf of the authors.

The authors made considerable efforts to modify the paper. However, unfortunately, there are still multiple issues which have not been addressed in a satisfactory way. Still, the current state of the paper and the replies made by the authors do not warrant the publication of the paper in its current stage in ACP without additional discussion. Here I repeat those issues, which I believe need some additional clarification. First I list my initial review point, then I add my updated remark, denoted as "Reviewer Remark":

1) On the background of existing literature I am not sure what the real novelty of this paper in terms of methods and results is. While I agree that C2 and C3 measurements have not yet been done before in Athens, it seems the inclusion of these does not yield more findings than already reported by Kaltsonoudis et al (2016). On the other hand, the Panopoulou et al paper makes same sketchy description of meteorological impacts on NMHCs, but lacks some elaborate analysis similar to those presented in Rappengluck et al (1998) for transport effects and also Kourtidis et al (1999) for temperature effects for Athens. As the authors make an important point on page 2, L27-29, that there have been new conditions during the economic crisis years (i.e. competing traffic vs wood burning emissions) it would be actually meaningful to perform a comparative analysis between the data sets reported 20 years ago and the ones reported by Panopoulou et al. It seems both studies include continuous NMHC measurements and PMF source apportionment analysis would be feasible and would provide interesting insights.

**Reviewer Remark:**

In their reply the authors did not really demonstrate the novelty of their work apart from the fact their measurements include C2-C3 data. However, including these NMHCs did not yield new knowledge about NMHC sources in Athens beyond the paper by Kaltsonoudis et al (2016). Kaltsonoudis et al (2016) even included detailed source apportionment for traffic and biomass burning for the Athens case, which the authors did not apply and did not address. What do we learn from this paper which has not yet been described earlier. Even worse, now the authors have removed the reference to Kourtidis et al (1999) who described temperatureeffects on traffic NMHCs emissions for the same city.

The authors argue: "However direct comparison with the work performed 20 years ago is difficult considering differences in sampling period (summer versus winter and thus different photochemistry), location, sampling method and analytical techniques." I cannot completely follow this argument. Emission studies can be done for nighttime periods minimizing photochemical effects, all the locations are within the GAA and sampling methods and analytical methods are similar. Even, if the suite of NMHCs may not be exactly the same, the application of source apportionment methods such as PMF would just require a few representative NMHC describing source profiles. There is not necessarily a "complete" suite of NMHCs necessary as it is also shown by the Kaltsonoudis et al (2016) paper, who applied a rather limited VOC dataset. Did the authors try at all to make comparisons? Again, in my opinion this would be feasible and would provide interesting insights.

*Authors Reply*:

We would like to thank the reviewer for his/her comments and explanation. Here is a point-by-point answer to the previous statements:

a) The authors demonstrated clearly in the introduction the novelty of this work. This work report levels of C2 – C6 NMHC (mainly) in Athens for winter. These levels for winter (or cold) season were never reported again in the literature, with only two exceptions. Indeed, Kaltsonoudis et al., (2016) report winter levels of isoprene, benzene, toluene, xylenes and monoterpenes (C5 – C8 NMHC), however it is apparent that the only common species between that work and ours are isoprene and benzene (although there are references for toluene, ethylbenzene and o- / m- / p- xylenes in the new manuscript). Concerning previous ground-based measurements in Athens that were conducted more than 20 years ago, we have to take into account three facts: 1) they, indeed, report levels of C4 – C12 NMHC, which is a greater number of compounds compared to this work (however, it is already mentioned that there are no measurements of C2 and only one work for C3 NMHC), 2) the majority of these measurements were conducted in summer, in which the meteorology and dynamics are different from winter and 3) it was demonstrated in the introduction that between 20 years ago and today, changes in pollutant sources occurred in Athens, as a result of the environmental protection strategies that were employed for the emissions control. Consequently, there is missing information for C2 – C6 NMHC in Athens that reflects the current situation, while there is no information about these species for winter (with the exception of isoprene and benzene). In addition, in Athens a new source of pollutants arose, i.e. wood burning for residential heating in winter. From the literature we know that this source contributes more to light NMHC than heavier, so our work is trying to provide an insight of the contribution of this source to the observed levels.

b) The authors still argue that direct comparison with the works performed 20 years ago is difficult considering differences in sampling period (summer versus winter and thus different photochemistry), location, sampling method and analytical techniques (or analytical resolution to be more precise). Nevertheless, they do agree that a comparison of emission ratios between the past and today is possible. Consequently, in the new manuscript a paragraph with comparison of enhancement ratios for Thissio and Patission station is now included.

c) The authors have indicated already from the title that the objective of this work is not VOC source apportionment in Athens, but the investigation of the contribution of traffic and heating to the observed levels. Consequently, the statement of the reviewer that "…including these NMHCs did not yield new knowledge about NMHC sources in Athens beyond the paper by Kaltsonoudis et al (2016)" is contradicting because, as it was mention previously, this paper is not dedicated to source apportionment and in addition, Kaltsonoudis et al (2016) did not include 9 out of the 11 NMHC that are reported in the current work. Thus, our work is yielding new knowledge for the two main NMHC **current** sources in Athens (traffic and heating), that is our focus.

d) The authors believe that the current state of the analysis of the measurements and the results are serving the goal of the paper and no PMF is needed. Based on their experience on PMF analysis, the authors are convinced that such an approach requires more species and a longer period in order to get more variability and contrast between factors (sources). Concerning the winter-time period under investigation, a more complete dataset is available only for a month. A PMF analysis would result in more than 2 factors, which would be certainly difficult to interpret due to a lack of robustness. That's why our current strategy is to focus first on these two wintertime sources (traffic and domestic heating) using specific ratios and additional tracers (e.g. BC wood burning and fossil fuel fractions) and in a second step in the near future to perform such PMF analysis using a one-year dataset.

Concerning the statement for the PMF that "Even, if the suite of NMHCs may not be exactly the same, the application of source apportionment methods such as PMF would just require a few representative NMHC describing source profiles. There is not necessarily a "complete" suite of NMHCs necessary as it is also shown by the Kaltsonoudis et al (2016) paper, who applied a rather limited VOC dataset", we would like to address here the "PMF overview" of the paper in question, Kaltsonoudis et al (2016), paragraph 5.4, page 14839: "In the present study the PMF analysis did not result in such separations since the VOC portfolio did not include light alkanes, which are usually included in datasets obtained by gas chromatographic (GC) techniques. The vehicular exhaust and LPG sources widely reported in the literature include a large fraction of these species; thus, such a separation was not favored by the PMF analysis.". Although in the current work are reported light alkanes, as we demonstrated before, we don't believe that a PMF analysis is suitable for the objectives of this paper.

**Abstract:**

Page 1 L20-21:

What do the authors exactly mean by local meteorology, as this term is quite unusual? Its connotation would mean that it is not representative for a larger fetch.

I disagree that "local" meteorology would control the variability of NMHC levels alone. What about the temporal variability of NMHC emissions?

**Reviewer Remark:**

In general, the term microscale meteorology is fine. However, still this would rather describe specific condition for mixing processes observed at a given site, which may not necessarily be similar for a different location within the urban area. For instance, the built environment would have a significant impact on these exchange processes.

In addition, what about PBL variations, which are not included in the term microscale meteorology? Also, still I believe, that the temporal variability of NMHC emissions is an important factor. Just one example: a typical morning rush hour peak is not entirely explained by low winds that are often observed during the same time frame. I would believe that low a PBL height and increased traffic emission would be an important factor. I would still doubt

that "Microscale meteorological conditions and especially wind speed seems to **control** the variability of NMHC levels...".

*Authors Reply*:

In general, we agree that depending on the location of the sampling station, the built environment could play a significant role to the observations. However, in our case, the station is situated on a hill, away from buildings and dense built environment and since the remark refers to the abstract we avoid clarifying further the microscale conditions. Nevertheless, given the structure of the current manuscript, we decided that this information could be included in Sect. 3.1 page 6 line 12 as follows: "…..*Apart changes in emission sources and source strength during the last twenty years, differences in sampling period (summer versus winter), location and built environment (kerbside versus urban background), analytical resolution…*".

Furthermore, the role of PBL was not underestimated in the manuscript. In page 7 lines 14-18 there is a reference discussing the role of the diurnal variability of PBL that could trigger the observed winter-time enhancement of the NMHC levels. Furthermore, according to Kassomenos et al. (1995) the day-night difference on PBL is more pronounced during summer compared to winter, supported also by the recent work of Alexiou et al. (2018). Thus, the night-time accumulation of the pollutants during the winter months highlights essentially the impact of additional emission sources. This statement is added on the manuscript (page 7, lines 18 - 20) and the following additional comment about the PBL is also now included on the abstract (page 1, line 20 – 23) regarding its role synergistically to the wind speed impact on the winter-time enhancement of NMHCs levels: '*Microscale meteorological conditions and especially wind speed in combination with the PBL height, seem to contribute significantly on the variability of NMHC levels, with an increase up to a factor of 10 under low wind speed (<3 m s$^{-1}$), reflecting the impact of local sources rather than long range transport*'.

Page 1 L27-29:

Why does the present data not allow for the quantification of the relative contribution of fossil fuel and wood burning for heating purposes?

**Reviewer Remark:**

Still, I have some trouble with the revised sentence. On one side the authors state that "For the night peak, the selected tracers and profiles clearly indicate contribution from both traffic and domestic heating..." and then continue that more specific tracers would be necessary for quantification of these sources. The latter statement contradicts the first statement: either the data set has tracers and profiles that clearly indicate these specific sources or not. Also, the authors emphasize the night peak. This implies that daytime emissions would be different. I would assume that domestic heating sources would become less, while traffic sources would increase and it seems the authors can differentiate this different behaviour in their data-set. Why would, for instance, a source apportionment method based on these specific tracers and profiles not be able to estimate relative source contributions?

*Authors Reply*:

We disagree with the reviewer on his/her comment that the two above mentioned sentences contradict each other. We claim that the data set used and the selected tracers clearly indicate the presence of two different sources during night but we cannot accurately quantify the role of each. In other words, it is a question of quantification and not qualification. The diurnal variability of $BC_{ff}$ and $BC_{wb}$ clearly corroborate our hypothesis. However, to avoid any misunderstanding we removed the sentence indicated by the reviewer "However, the present data-set does not allow for quantification of each source due to the similarity of emissions, thus measurements of more specific tracers are needed for the better understanding of the contribution of these nocturnal VOC sources". We replace it by "*NMHCs slopes versus $BC_{wb}$ are almost similar when compared to those versus $BC_{ff}$ (slight difference for ethylene), indicating that NMHCs are probably equally produced by wood and oil fossil fuel burning*" (page 1, lines 30-31) which is in line with the similar levels of $BC_{ff}$ and $BC_{wb}$ observed during night. Nevertheless, the quantification of each emission source is not possible by means of the specific methodology.

**Introduction:**

Page 2 L30: It would be fair to mention how many NMHCs were actually measured as it seems that the paper does not report some important NMHCs such as 1,3-butadiene and others, for instance.

**Reviewer Remark:**

The reply given by the authors is not convincing. Still, it would be fair to mention the number of NMHCs measured in this range and the number of NMHCs in this range used in this study in order to make a precise statement here and avoid any speculation.

*Authors Reply*:

The text was corrected accordingly in page 3 lines 14-17: '*The current study presents, time-resolved data of 11 selected from 15 determined NMHCs with two to six carbon atoms, during a time span of several months (October 2015 to mid-February 2016) in the Great Athens Area (GAA). In addition, time-resolved data of toluene, ethylbenzene, m- /p- xylenes and o-xylene are used, which were monitored simultaneously from mid-January to mid-February 2016.*'.

Page 2 L34 - page 3 L1: The authors should mention why the analysis is restricted towards traffic and heating impact on NMHC levels.

**Reviewer Remark:**

10 There is still an issue: in order to perform an "..investigation of traffic and residential heating impact **on** the NMHC levels..." the authors would still need to verify and quantify any other potential NMHC source, regardless of its magnitude. Even, if there are no industrial sources as the authors point out, I still would think there are solvent and evaporative sources. Also, what about the port of Athens?

Otherwise, I am not sure what still needs to be investigated exactly for this task (3).

*Authors Reply*:

Great changes in Athens VOCs sources occurred the last 20 years such as significant decrease in industrial emissions around Athens (the 2 power plants were decommissioned), renewal of the cars fleet and since 2013 industrial activities were significantly reduced due to economic recession. Our suggestion for the existence of two main sources namely traffic and

20 heating is in line with Kaltsonoudis et al (2016) who reported that wood burning and traffic consist the main winter-time emission sources in Athens, whereas the industrial impact is low. Regarding the reviewer suggestion to tackle the evaporative sources note that the *Section S.3. Investigation of the evaporation losses* of the manuscript was already presenting an approach for the investigation of the evaporation losses based on the ratio Butanes-to-(C2 – C5)Alkanes (%) and Pentanes-to-(C2 – C5)Alkanes (%) versus the temperature. For both cases low evaporation occurs under low

25 temperatures, which is the winter case. The relationship of temperature and evaporation emission in Athens was previously investigated by Kourtidis et al (1999). Thus, in order to compare our findings with this past report, in the revised supplement (*Section S.3. Investigation of the evaporation losses*) a new graph based on the approach described in Kourtidis et al. (1999) is included. Finally, activities in the harbor are much linked to touristic activities and recreation which is significantly reduced in winter and the biggest part is related to traffic of heavy trucks for loading and uploading.

30 Nevertheless, since the remark concerns part of the introduction, there is no reason to justify in this sentence our choice to further investigate these two sources. Moreover, in the paragraph 3.4.3 of the Sect. 3 "*Results and Discussion*", we have demonstrated that we investigate specific SP samples (meaning certain hours) for the morning peak and the night-time enhancement period so (1) SP periods reflect the impact of local sources rather than long range transport and (2) based on the analysis and comparison with other pollutants, we can assume that we have minimal contribution of other sources than

traffic and heating. However, in order to better clarify our intentions, we decided to change the title of the paragraph 3.4 (page 9 line 1) from "*Identification of NMHC emission sources*" to "*Identification of NMHC emission sources with emphasis on traffic and heating*".

**On line NMHC measurements**

Page 4, L5: Why is only toluene used? Why not at least ethylbenzene and the xylenes in addition? Would the exclusion of these NMHCs not introduce a bias into the data analysis, as important tracers for solvent emissions are excluded? What are
10    the uncertainties and the detection limits for this GC?

**Reviewer Remark:**

This appears cherry-picking to me. The authors state that they focus on C2-C6 NMHCs, then include a C7 compound from a different instrument, but exclude other NMHCs from that different instrument. The authors did not address my question whether the exclusion of these other NMHCs would introduce a bias in the data analysis. As the authors have a complete
15    suite of NMHC measurements, it would be straight forward to include all NMHCs, and make a source apportionment analysis first, and not rely on a reference only (Vrekoussis et al., 2013), which did not include VOCs.

*Authors Reply*:

In order to better address the concerns of the reviewer, in the morning profile ethylbenzene and o-m-p xylenes are now
20    additionally included in order to eliminate the bias into the data analysis. In the new figure 9 (morning profile) we observe the important contribution of the m/p – xylenes, however the general image of the profile didn't change. Thus, Figure 9 is corrected as follows on the manuscript:

[Figure]

**Figure 9**. %Mass contribution of the measured NMHCs during the morning peak (07:00 – 10:00LT), median values in Thissio and mean values in Patission Monitoring Station.

5    The old figure 9 is transferred on the Supplement:

[Figure]

**Figure 9**. % Mass contribution of the measured NMHCs during the morning peak (07:00 – 10:00 LT), median values in Thissio, mean values in Patission Monitoring Station, in a highway tunnel in GAA and a highway tunnel close to Paris.

Moreover, as it was already mentioned in the previous comment (for Page 2 L34 - page 3 L1) (1) PMF source apportionment is not an objective of this paper and (2) the chosen samples of the analysis are unlikely to be influenced by other sources than traffic and heating.

5 Furthermore, the reference to Vrekoussis et al., (2013) concerns the reduction on the industrial activity and is not connected with emission sources since it is used in order to support the scope of the study.

**Tunnel measurements**

Page 4 L16: The authors should mention the length of the tunnel, whether lanes were for both directions (there could also be dedicated tunnels for one direction only), if there was any artificial ventilation and if there might have been any limitations on traffic through this tunnel (in some cases heavy duty traffic is not allowed). In any case an estimate of the traffic fleet composition (e.g. heavy duty vs light duty vehicles) would be helpful. All these factors have an impact on the NMHCs

15 levels. At what location of the tunnel did the authors make the measurements exactly? I see the measurements were taken on 12 May 2016, which is different from wintertime. Wouldn't the temperature be different from wintertime and wouldn't this have an enhanced impact on NMHC emissions through evaporation, for instance?

**Reviewer Remark:**

It seems the description of the tunnel is still missing in the manuscript. Also, the authors did not mention explicitly whether

20 this tunnel contains all lanes for both driving directions or whether there were separated compartments for each driving direction. They did not mention either, whether there was any artificial ventilation or not.

*Authors Reply*:

The authors decided to transfer the tunnel measurements (experimental and figures) to the supplement in Sect. S.2a.

25 Furthermore, in the description of the tunnel (now in the supplement) the additional information is added in page 12, lines 20 - 21 as follows: "*Each driving direction consist a separated compartment and a ventilation system was installed but not operated*".

30 **Temporal variability of NMHCs**

Page 4 L28: I do not understand the concept of data coverage here, as it is not explained. It could refer to the percentage of data above the detection limit vs maximum available data, but this does not make complete sense, as I doubt there were any data of ethane below the detection limit, for instance. However, it cannot be true either that it refers to the data availability vs

maximum potential data availability during the time period reflecting instrumental potential instrumental malfunctions and/or failure. This should be clarified. The only thing I understand is that the there has been some interruption of NMHC data contrary to what the authors claim in the abstract of the paper.

**Reviewer Remark:**

5   The authors statement of 87% data availability would imply that the instrument did not work uninterrupted (again, I believe that such a species like ethane would always be above the detection limit, if the instrument is working properly). As a consequence I would recommend to replace the statement made in the abstract of the paper (page 1, L16) "..to our knowledge, time resolved, uninterrupted data of NMHCs..." by "..to our knowledge, time resolved data of NMHCs...".

10   *Authors Reply*:

We followed his/her suggestion and now in page 1, line 16 reads as follows: "*The current work presents, for the first time to our knowledge, time resolved data of NMHCs, from two to six carbon atoms..*".

15   Page 5, L15: The authors should clarify why the reader should bear in mind differences in sampling methods and analytical techniques. Are some of the sampling methods and/or analytical techniques and associated results listed suspicious and cannot be compared to each other?

**Reviewer Remark:**

The authors did not change much. The authors still state that "....this decrease has to be seen with cautious considering
20   differences in [...] sampling method and analytical techniques". Are some of the sampling methods and/or analytical techniques and associated results listed suspicious and cannot be compared to each other?

*Authors Reply*:

The analytical techniques used in the past were reliable and all results are totally comparable in technical basis. Our
25   statement about the techniques was mainly related to the sampling frequency and it is corrected to avoid any misunderstanding. Nevertheless, there are differences between the past and the current conditions that should be stated in order to be consistent with parameters could drive the observed changes. Measurements during cold or warm period definitely determine the levels of NMHCs due to changes on emission sources strength, dilution processes or photochemical rate. The levels of traffic related compounds on samples collected close to major street canyon could also be differentiated
30   by those in further distance. In the work of Moschonas and Glavas, (1996) performed close to our sampling site, the samples (n = 12) were collected during morning in summer, thus the comparison with the current results considering periods including the total day fraction could result into discrepancies. One more important parameter that should be taken into consideration is the reduction of the industrial activity in Athens and the close located areas during the last decade; and the increase of wood burning after 2012 as a consequence of the financial recession. Both processes and other antipollution

practices applied the last years as well, are clearly expected to differentiate the emissions into the atmosphere and possibly change the pollution profile in GAA. However, as it is already mentioned before, the authors agree with the reviewer for a comparison of emission ratios between past measurements and current results.

To summarize all the above, the authors firstly change the statement in question as follows (page 6, lines 10 - 16): "*This decreasing trend is in agreement with a decrease in primary pollutants CO, SO$_2$ already reported by Kalabokas et al. (1999) and Gratsea et al. (2017), due to the air quality measures taken by the Greek government and economic recession (since 2012). Apart changes in emission sources and source strength during the last twenty years, differences in sampling period (summer versus winter), location and built environment (kerbside versus urban background), analytical resolution (morning collected samples compared to continuously averaged levels) should be considered rendering the direct comparison between the present and past measurements quite difficult in the overall evaluation of the NMHCs decrease'*.

[revised manuscript text omitted]

Page 5, L18-20: The same comment as above applies here. As long as there is no more elaborated comparison, the presentation of the data remains generic.

**Reviewer Remark:**

Why would it be important to note that Paris is located at the mid-latitude and in the northern hemisphere with regard to NMHC emissions? Would it be different for city at the same latitude, but in the southern hemisphere? For instance, would traffic emissions depend on the latitude?

There is not such a thing as oceanic continental climate; this does not exist! Apart from that the climate of Paris is definitely not continental! There are no cold winters in Paris! Also, in Paris, it can be pretty hot during summer time, which cannot be

considered "mild". I would rather use that term for the winter period in Paris. The authors want to check an appropriate climate description for Paris.

*Authors Reply*:

5   We thank the reviewer for the comment and we corrected the phrase as follows (page 7 line 7-9), since the emissions are not latitude depended: "*NMHC levels are also compared with those obtained in Paris, one of the European megacities, with more than 10 million inhabitants with relatively mild winters and warm summers. Again, the observed levels in Athens are significantly higher compared to those reported for Paris (Baudic et al., 2016).*

Page 5, L25-27: The authors neglect to mention the annual variability of other NMHC sources, e.g. evaporation losses.

**Reviewer Remark:**

With regard to traffic related evaporative NMHC emissions the authors need to consider the work by Kourtidis et al. (1999)

15   who specifically addressed the processes related to these emissions for the same city, Athens. The NMHC mixture might have changed, but not the underlying processes. Unfortunately, the authors removed this reference from the previous manuscript version.

*Authors Reply*:

20   The reference of Kourtidis et al. 1999 was erroneously removed from the revised version and now was included again. More specifically, to compare our observations for the relationship between temperature and VOC emissions from fuel evaporation with those of Kourtidis et al. (1999), we now included in the supplement a new graph presenting the ratio of NMHC/benzene for the 15:00 and 07:00 (normalized to the OH reaction) to the boiling point of the selected NMHC, following the approach described in Kourtidis et al (1999). Accordingly, in the manuscript page 11 line 13-17 was modified as follows: "*These*

25   *observations are in agreement with the general behavior of the temperature dependency reported in Kourtidis et al., (1999) (Fig. S13 and Sect. S3), that performed an investigation of the dependence of the fractionation of NMHCs in evaporative emissions from temperature in Athens. Although the examined periods differ in ambient temperature (winter is colder than autumn), the exponential curve fitting of both datasets was similar.*"

30   The new figure for the supplement:

[Figure]

**Figure S13.Ratios of the NMHC/benzene ratio for 15:00 (higher daily temperature) and 07:00 (lower daily temperature) to the boiling points of the selected NMHC, divided by the reaction rate constant of each species with OH. The plotted NMHC are propane, i-/n- butane,i-/n- pentane, toluene, ethylbenzene and o-xylene.The red cycles indicate compounds (not values) in common with the work of  Kourtidis et al., (1999).**

Page 6, L4-7: From Fig 3 I see that Bff increases similarily to Bwb at night. Why can the authors make the statement that traffic would not be as important as heating?

**Reviewer Remark:**

The authors do not consider the nighttime variation of the PBL which would also contribute to a sustained higher level of pollutants, regardless of their origin.

*Authors Reply*:

From figure 4 it is obvious that $BC_{ff}$ during the months with residential heating (November to February) is similar to $BC_{wb}$. Taking into account that $BC_{ff}$ originates from both heating using fuel and traffic and that during October traffic contributes almost 50% of the $BC_{ff}$ observed during the months with heating activities (November to February) one could deduce that heating from both fuel and wood burning is more important compared to the traffic. Regarding the reviewer comment on the role of PBL on NMHCs variability information is now added (page 7 line 30-32) '*Moreover, night-time emissions occur in a shallower boundary layer relatively to the mid-day, resulting into accumulation of pollutants (Alexiou et al., 2018).*'.

Page 6 L9-10: I disagree. Usually, PBL heights are at a minimum during morning hours before sunrise, unless the authors can show other evidences for their statement.

**Reviewer Remark:**

I did not see the change the authors may have done in the text (mentioning the line would be helpful).

*Authors Reply*:

The statement for a higher morning PBL relatively to the night-time concerns the period after the sunrise, i.e around mid-day and morning was replaced by mid-day'. During the preparation of the second version of the manuscript the paragraph *3.2 Diurnal variability of NMHCs* was rearranged and the PBL comment was removed. We add the comment again with the proper correction '*Moreover, night-time emissions occur in a shallower boundary layer relatively to the mid-day, resulting into accumulation of pollutants (Alexiou et al., 2018).*' in page 7 line 30-32.

**The role of meteorology on NMHC levels**

Page 6 L15-20:

This is a pretty generic description. It is well-known that the concentration of primarily emitted gaseous pollutants will decrease due to dilution regardless of their chemical class.

However, windspeeds < 3 m/s alone would not indicate the presence of local sources. This would only be true for calm winds. From the plots it seems like these are skewed distributions with maximum concentration values around 2 m/s or so. This would rather indicate some regional flow impacts, which the authors neglected to consider. It seems a more elaborate analysis of windspeeds and their effect on NMHC levels in the Athens area has already been presented in Rappengluck et al (1998).

With regard to potential long-range transport it is actually interesting to see that there is some acetylene data still around 5 ppb or so at windspeeds around 9 m/s and higher. In fact, those are very high acetylene values despite strong dilution. What is the reason for this?

**Reviewer Remark:**

As the x-axis of figure 5 shows discrete values, it would be good to have a clear definition of these ranges. Also, are calm wind situations considered, i.e. below the threshold of the wind sensor (both, in figures 5 and 6)?

*Authors Reply*: All wind speed values above the threshold of wind sensor are used on Figures 5 and 6. The lower monitored value is 0.4m/s. The x-axis is properly adjusted in order to depict the range of each bin as also noted in the figure caption.

[Figure]

**Figure 5. Boxplots for (a) n-butane, (b) acetylene and (c) benzene relatively to wind speed for the period 16 October 2015 - 15 February 2016. The black line represents the median value and the box shows the interquartile range. The bottom and the top of the box depict the 1$^{st}$ and 3$^{rd}$ quartiles (i. e. Q1 and Q3). The whiskers correspond to the 1$^{st}$ and the 9$^{th}$ deciles (i. e. D1 and D9). The range of each wind speed bin is depicted on x-axis.**

**Identification of NMHC emission sources**

Page 8, L28-30: Did the authors also apply the baseline subtraction for the tunnel measurements?

**Reviewer Remark:**

The authors should mention this limitation in the text.

*Authors Reply*:

Our reply is now mentioned in Sect. S.2a page 13 line 11-12 of the Supplement: '*Due to the small number of samples a baseline subtraction from the tunnel data was not possible*'.

Page 8, L30-31: How can the authors justify that their tunnel measurements are not influenced by outside air masses?

**Reviewer Remark:**

There must be some dilution. A tunnel is not a closed box and it seems the length of the tunnel is only 200 m.

*Authors Reply*:

We agree with the reviewer on the potential dilution, especially for such a short tunnel and for that reason, the sampling was conducted in the middle of the tunnel in order to eliminate any transfer of air from the ambient environment. We have already justified our choice in Sect. S.2a page 12, line 21-22.

Page 8, L31-32: I completely disagree on the authors' statement. The authors neglect to mention what they consider "dominant species", however just looking into NMHCs such as acetylene, benzene, and toluene, the two profiles "Morning Peak Athens" and "Highway Tunnel - Athens" are completely different: while acetylene for the "Morning Peak Athens" is about 6-7 times higher than for the case "Highway Tunnel - Athens", benzene and toluene values are about 2-3 times lower at the same time.

**Reviewer Remark:**

I agree that the "Morning Peak Thissio" and "Morning Peak Patission" profiles agree with each other within the uncertainties. Still their comparison with the tunnel measurements and both tunnel measurements among themselves are very different. The authors themselves make a reference to various papers which report important differences between tunnel measurements worldwide and also state that there is a possibility that the car-fleet in the tunnel is not representative for the GAA. Why then compare two consistent real-world street canyon measurements with tunnel measurements, whose data is questionable?

*Authors Reply*:

What we compare in our study is the profile at an urban background location (Thissio) with that of a street canyon (Patission) and a tunnel and not two real-world street canyon measurements with tunnel as the reviewer states. Also, the tunnel measurements were now moved to the supplement (as Sect. S.2a). The profile of the morning peak at Thissio and Patission are providing similarities, highlighting the common fuel type. The profile of the Athens tunnel is quite differentiated by the Paris one and the findings are in accordance with previous works elsewhere as stated in the manuscript. The investigation of the evaporative losses in the Supplement, indicates that the emission differences between the Athens tunnel and the other two locations could be attributed to the higher temperatures during the tunnel measurements in combination with the decreased car fleet relatively to the early morning compared to the two profiles obtained for the city (not possible to perform measurements in the tunnel on a 24h basis). Furthermore, the side effects of the financial recession are depicted through the avoidance of the tunnel due to the tolls.

Page 8, L32 - Page 9, L1: I disagree here again! I do not see that profiles fit nicely. Instead, there are a lot of significant differences. Also, what do the authors consider "common NMHCs"?

**Reviewer Remark:**

See my Reviewer Remark above

*Authors Reply*:

See our answer above.

10   Page 8, L1-3: Why should there be higher traffic related butane fraction due to evaporation in ambient air than in the tunnel? Even more surprising, as the tunnel measurements were taken in May, which presumably has warmer temperatures than wintertime. Also, when butanes should be related to evaporation why does propane, another prominent tracer for evaporation,show pretty similar values in the tunnel measurements compared to the "Morning Peak Athens" data?

**Reviewer Remark:**

15   The remarks made by the authors in the comments for P8, L31-32 were: "Moreover higher values of ethane, propane and butanes that are depicted in the morning hours at the urban sites relatively to the tunnel measurements, reflect the increased numbers of LPG powered vehicles in Athens and natural gas-powered buses". I deduce again, that the tunnel measurements were not representative for the GAA. For me, it does not make much sense to include the tunnel data then.

20   *Authors Reply*:

As it is already mentioned, the tunnel measurements and discussion are now transferred to the supplement. However, the comparison with the street canyon measurement and the tunnel measurements remain in the main manuscript, a fact also justified in a previous comment (for Page 8, L31-32).

Page 8, L12-14: The definition of the background concentration appears odd. How can the minimum value between 12:00-17:00LT be representative for the nighttime period 18:00-05:00LT? Both are pretty long periods (5 and 11 hours, respectively). From Figs. 4 and 5 we learnt that the NMHC concentration critically depends on wind speed and wind direction. How can the authors make sure that such changes in wind speed and/or wind direction would neither occur during

30   the daytime reference period nor during the nighttime period?

**Reviewer Remark:**

The authors did not address the impact on wind direction. Background NMHC concentrations can critically change with wind direction, in particular at coastal locations. How can one minimum NMHC value observed during mid-day and

potentially under one distinct wind direction be representative for all potential wind direction conditions at nighttime from 18:00- 05:00 LT? Also, the authors neglect daytime photochemical degradation, albeit limited, but still present in those months.

*Authors Reply*:

Regarding the photochemistry, with regard to OH radicals, lifetimes in the range of 21h for propene to several days for the rest of NMHCs (e.g. 2.7d, 10d and 21d for ethylene, n-butane and propane, isoprene is excluded) occur under a typical daytime (12h) winter level of $5x10^5$ molecule $cm^{-3}$ OH radicals. By considering that the changes of the local emission sources of traffic and heating spanning in the range of a few hours during morning and evening, the role of photochemical degradation could be omitted due to the dominance of the emissions. Considering the long lifetime of the NMHCs and the decreased photochemistry on winter-time the wind speed is of more importance than wind direction. The morning profile at Thissio station has been determined for smog conditions, i.e. low wind speed all directions included, reflecting local sources not related to NMHCs transported from different directions.

Page 8, L16-19: Are these differences statistically significant?

**Reviewer Remark:**

The statement made by the authors should be added in the manuscript.

*Authors Reply*:

The statement is added in the manuscript (page 11, line 34 – page 12, line 2) as follows: '*When these two profiles are compared (Fig. 11), the statistically significant at p<0.01 difference is obvious, with a smaller contribution from i-pentane (traffic source contributor) during December*'.

Page 10, L13: What were those "different meteorological profiles"?

**Reviewer Remark:**

Still, the term "profile" does not make sense, as long as it is not defined (for instance, there are vertical temperature profiles).

*Authors Reply*:

The term profile was accordingly replaced by the most appropriate term '*conditions*' in page 12, line 31.

Page 10, L14-16: This is not supported by the data presented in the paper!

**Reviewer Remark:**

Still, the tunnel measurements would not support this as discussed earlier.

*Authors Reply*:

Any conclusions concerning the tunnel measurements are removed from the manuscript. Please refer to our answer previously for comments Page 8, L31-32 and Page 8, L1-3.

Table 1: Remove the term "worldwide" in the table caption, as Table 1 shows a few selected data from the Mediterranean/European area at the most. What does the second sentence of the table caption refer to? What quantities are compared in this table: means or medians or ....? What do the authors mean by "sampling" frequency: sampling duration or measurement cycle? There is no information given for "sampling frequency" for Baudic et al., Salameh et al., and Durana et al.. Why are the results for the summer 2012 and winter 2013 Athens campaigns reported by Kaltsonoudis et al (2016) not listed in this table? At least, results for isoprene, benzene, and toluene would be comparable.

**Reviewer Remark:**

The authors should state explicitly that mean values are shown in the table.

*AuthorsReply*:

We disagree with the remark. Based on the Table 1 caption, the use of the mean values was already stated: '*Comparison of NMHCs mean levels between this study and already published works in Athens, Greek and other Mediterranean or European sites. Information about the analyzing or sampling techniques and data resolution are included when available. The number of measurements[a] for each compound determined on the current samples is included below the table*'. In order to be consistent, we also add the term 'mean' in the manuscript (page 6, line 7): '*In Table 1, the mean measurements of this study are compared with those reported in the literature for Athens in the past and other selected areas*'.

Figure 10, figure caption:

I disagree that such a long time period (18:00-05:00 LT; 11 hours!) can be considered a nighttime "peak". Error bars should be included. I do not see that the values shown in the figure add up to 100%.

**Reviewer Remark:**

I did not argue the term "nighttime", I argued the term "peak", which does not make much sense for a time period of 11 hours.

*Authors Reply*:

The period 18:00-05:00 LT corresponds to the night-time enhancement of the observed levels based on Figure 8, whereas the night-time peak is encountered around midnight. Thus, we agree with the reviewer about the 'peak' and we correct accordingly the caption of figure 11 (first version Figure 10) by replacing the term 'night peak' by the more appropriate '*night-time enhancement period*'.

[revised manuscript text omitted]

---

## Author Response (AR3)

Reference article: acp-2017-936

Title: Non Methane Hydrocarbons variability in Athens during winter-time: The role of traffic and heating

5 ## Answers to co-editor

**We would like to thank the co-editor for his comments and we provide the necessary clarifications. Changes on the manuscript (discussed on the authors reply) are indicated with italics. Furthermore, the marked typos are corrected on the manuscript and Table 3 (depicted as figure before) is replaced by the**
10 **corresponding table format.**

**Co-editor report**

I think that the paper shows now enough novel aspects to be a valuable publication for ACP. Measurements in Athens are still scarce especially in the winter season, pollution sources have changed over time (increase of
15 wood burning). Measurement of compounds not yet observed in this region are reported in the paper. Comparisons with previous work have been extended, including also comparison of emission or enhancement ratios.

The author argue, that PMF analysis is premature at this point, because results would probably complicate analysis of the major pollution sources (traffic and heating) they focus on. I found this argument acceptable.
20 The importance of evaporative sources is adequately stressed. They seem to be less important during winter time due to low temperatures, but still are significant.

Tunnel results are now better described, and their importance downsized (now in additional information).

I have some few clarifications to ask the authors. Also some typos are marked in the joint text.

**Page 6 lines 10 - 16: location and built environment (kerbside versus urban background),**

This could suggest that one of the sites is a kerbside, and the other an urban background one. Then a comparison of past and present NMHC levels would not make sense, but I guess this is not the case, as authors state that both sites are nearby. Still it could be possible that one site is more impacted by traffic than another, or that traffic
30 locally changed over time. Please clarify.

**Authors reply:** We agree with the remark. Since both sites are close (both are pedestrian zones) the statement about the kerbside versus urban background has been removed and the corrected sentence at P6 L12-15 is: "*Apart changes in emission sources and source strength during the last twenty years, differences in sampling period (summer versus winter), analytical resolution.. ".*

**Page 6 line 16 : NOx conversion factor**

NMHC to NOx enhancement ratios document the decrease in traffic NMHC emissions. A detail question : Molecular weight of NOx equal is assumed to 31.6. This value is based on an assumed NO2/NOx distribution, but which has probably also have changed during past 20 years. This probably should be taken into account (unless initial measurements were on a molecular density basis and have been translated to mass using the same factor). It is understood that the decrease in ratios still would be present, even if adjusting this factor.

**Authors reply:** The measurements reported in Kourtidis et al. (1999) were on ppb, thus in their work they converted the ppb/ppb ratios of NMHC-to-NOx in w/w (weight/weight) and for that they assumed the molecular weight of NOx equal to 31.6. In order to compare the ratios of Kourtidis et al. (1999) with our ratios (that are in ppb/ppb), we simply re-converted the reported w/w ratios of NMHC-to-NOx into ppb/ppb, by following the reverse calculations described in Kourtidis et al., (1999). Since the ratio of NO2/NOx could be changed with time, we didn't use the molecular weight of 31.6 for NOx for our ratios. We clarify this in two parts in the manuscript:

20     (1)    In P6 L16 of the manuscript we clarify as follows "*…to past measurements, enhancement ratios (ppb/ppb) are calculated…"*

    (2)    In Table 2 the reference for the 3[rd] column is modified as follows: "*The NMHC-to-NOx enhancement ratios of Kourtidis et al. given in w/w (weight/weight) were converted in ppb/ppb by dividing them with the ratio of the molecular weight of the NMHC to the molecular weight of NOx (equal to 31.6 according to Kourtidis et al., 1999)*''

**Page 7, lines 7 – 9 : comparison to Paris data**

"NMHC levels are also compared with those obtained in Paris, one of the European megacities, with more than 10 million inhabitants with relatively mild winters and warm summers. Again, the observed levels in Athens are significantly higher compared to those reported for Paris (Baudic et al., 2016)."

Please put a point after the sentence. And then give some more details, for example concentration differences for classes of compounds or species seen as tracers. The only qualitative comparison is not very informative.

**Authors reply:** To better present the quantitative differences between the current measurements at Athens and the other cities, not only Paris, we decided to add information about the observed levels. The text on P7 L01-11 is adjusted as follows: '*Beirut, located in the Eastern Mediterranean basin (approximately 200 Km SE of Greece, 230 m above sea level), has a population of 2000000 inhabitants and a typical Mediterranean climate with mild winter and hot summer (Salameh et al., 2015). On the contrary Bilbao is an urban and industrial city with 400000 inhabitants in northern Spain, located along a river delta in SE–NW direction, with two mountain ranges in parallel to the river (Ibarra-Berastegi et al., 2008). Due to their location, both cities experience intense sea breeze cycles. The NMHCs levels observed in Athens are higher by a factor of approximately two for ethylene, propene, acetylene and pentanes compared to these two cities and up to 3.5 for isopentane at Bilbao. Exceptions are propane, butanes and toluene for Beirut and n-butane, benzene and toluene for Bilbao, which are quite comparable to Athens. NMHC levels are also compared with those obtained in Paris, one of the European megacities, with more than 10 million inhabitants with relatively mild winters and warm summers. Again, the observed levels in Athens are significantly higher (almost 2 to 8 times) compared to those reported for Paris (Baudic et al., 2016), with the most important differences concern acetylene and i-pentane (factor of 8.4 and 6.7 higher in Athens respectively, Table 1).*

**Reply to remarks on the manuscript**

Page 8: "The influence of the N to SE sector to the enhanced NMHCs levels is probably related to the northern suburbs of GAA, that are characterized by increased number of fireplaces, while the higher living standards of their inhabitants allow the combined use of heating oil in central heating systems and wood in fireplaces and/or woodstoves. "

Just to be sure:

Does the argument of higher living standards apply to population in the northern or southern sector ?

**Authors reply:** The higher living standards apply to population of the northern suburbs (sector). To better clarify it we modify the sentence as follow: '*
[revised manuscript text omitted]